# Priors in Time: Missing Inductive Biases for Language Model Interpretability

**Ekdeep Singh Lubana**[1*]**, Can Rager**[2*]**, Sai Sumedh R. Hindupur**[3*]**,**
**Valérie Costa**[4]**, Greta Tuckute**[5]**, Oam Patel**[3]**, Sonia Krishna Murthy**[5]**, Thomas Fel**[5]**,**
**Daniel Wurgaft**[1,6]**, Eric J. Bigelow**[1,7]**, Johnny Lin**[8]**, Demba Ba**[3,5]**,**
**Martin Wattenberg**[3]**, Fernanda Viegas**[3]**, Melanie Weber**[3]**, Aaron Mueller**[9]

[1]Goodfire AI, [2]Independent, [3]SEAS, Harvard University, [4]EPFL
[5]Kempner Institute at Harvard University, [6]Department of Psychology, Stanford University,
[7]Department of Psychology, Harvard University, [8]Decode Research, [9]Boston University,
[*]Co-first authors

## Abstract

A central aim of interpretability tools applied to language models is to recover meaningful concepts from model activations. Existing feature extraction methods focus on single tokens regardless of the context, implicitly assuming independence (and therefore stationarity). This leaves open whether they can capture the rich temporal and context-sensitive structure in the activations of language models (LMs). Adopting a Bayesian view, we demonstrate that standard Sparse Autoencoders (SAEs) impose priors that assume independence of concepts across time. We then show that LM representations exhibit rich temporal dynamics, including systematic growth in conceptual dimensionality, context-dependent correlations, and pronounced non-stationarity, in direct conflict with the priors of SAEs. This mismatch casts doubt on existing SAEs' ability to reflect temporal structures of interest in the data. We introduce a novel SAE architecture—Temporal SAE—with a temporal inductive bias that decomposes representations at a given time into two parts: a predictable component, which can be inferred from the context, and a residual component, which captures novel information unexplained by the context. Experiments on LLM activations with Temporal SAE demonstrate its ability to correctly parse garden path sentences, identify event boundaries, and more broadly delineate abstract, slow-moving information from novel, fast-moving information, while existing SAEs show significant pitfalls in all the above tasks. Our results underscore the need for inductive biases that match the data in designing robust interpretability tools.

## 1 Introduction

Given the success of Language Models (LMs) (Bubeck et al., 2023; Deepmind, 2025), there is growing interest in understanding how such models incrementally update over sequences of tokens to exhibit complex behaviors (Murthy et al., 2025; Lindsey, 2025; Lindsey et al., 2025; Lepori et al., 2025; Bigelow et al., 2025; Tuckute et al., 2024; Klindt et al., 2025). Interpretability research aims to make such analyses tractable, offering tools for hypothesis design, testing, and intervention based on evaluation of intermediate activations (Geiger et al., 2025; Sharkey et al., 2025; Bereska and Gavves, 2024). Often, such work builds on hypothesized computational models of how concepts are encoded in a neural network's representations, e.g., the linear representation hypothesis (LRH) (Elhage et al., 2022; Arora et al., 2018), correspondingly motivating tools such as sparse autoencoders (SAEs) (Gao et al., 2024; Cunningham et al., 2023) for unsupervised extraction of a dictionary of vectors that (ideally) mediate human-interpretable concepts (Mueller et al., 2025).

A central challenge in "bottom-up" approaches to interpretability, like SAEs, is the mismatch between the assumptions of their underlying implementational account and the precise behavior or computation they intend to explain (Jonas and Kording, 2017; Geiger et al., 2025; Costa et al., 2025) (see Fig. 1). For instance, since LRH posits that different concepts correspond to directions in activation space

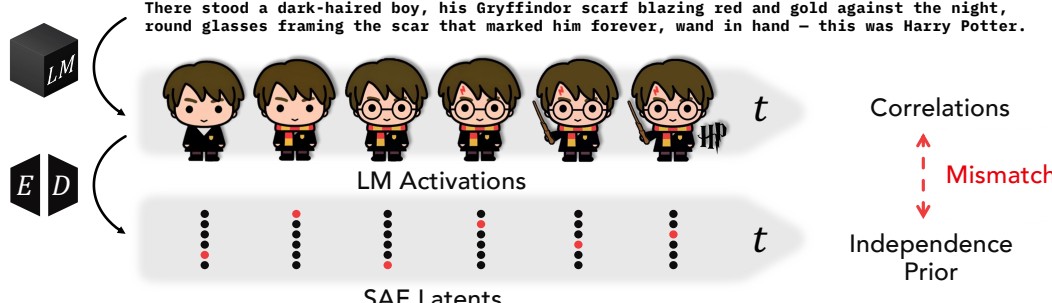

Figure 1: **The mismatch between SAE assumptions and temporal structure of language**. An illustrative sentence describing attributes of Harry Potter is shown. When passed into a language model (LM), it leads to activations $x_t$ that include concepts within them (possibly entangled): note the presence of large numbers of shared attributes over time, which manifest as correlations across time of activations. Sparse Autoencoders (SAEs) implicitly have an independence (i.i.d.) prior across time $t$ over their latents and thereby over concepts, which clashes with the true structure of language.

that can be independently manipulated, it implicitly claims the data distribution can be factorized into independently varying latent variables (Allen, 2024). This mismatch between the structure of the data distribution and strong priors codified in LRH can lead to misleading or pathological explanations when using SAEs to understand neural networks (Chanin et al., 2025; Bricken et al., 2023; Hindupur et al., 2025). This raises a set of critical questions for using SAEs to interpret models trained on sequential data like language. Specifically, since language exhibits rich temporal structure at *multiple scales* (Marslen-Wilson and Tyler, 1980; Thompson, 1999)—e.g., sentences contain dependencies that link words across time (Gibson et al., 2000; McElree et al., 2003), upcoming words can be anticipated from context (Hale, 2001; Levy, 2008), and discourse imposes structure over longer timescales through phenomena like event boundaries (Zacks et al., 2007; Baldassano et al., 2017)—one can ask *what assumptions about temporal structures do SAEs make? How do these assumptions align with the actual temporal structure present in a LM's activations?*

**This work.** Building on the Bayesian interpretation of sparse coding (Olshausen and Field, 1996; 1997)—the framework that motivates SAEs—we rephrase the optimization objective of SAEs as a MAP (maximum a posteriori) estimation problem. This allows us to make explicit prior assumptions about temporal structure embedded in SAEs, showing they implicitly assume concepts are *uncorrelated across time* and the number of concepts necessary to explain an activation is *time-invariant*—that is, the information present at each token position is independent of information at other positions and uniformly distributed. As we empirically show, these assumptions stand in stark contrast to the actual temporal structure present in language and language model representations, and can result in empirically observed pathologies in SAEs, such as feature splitting (Bricken et al., 2023; Chanin et al., 2025; Bussmann et al., 2025).

These results then motivate us to draw a broader parallel between SAEs and computational neuroscience approaches for understanding neural data. Specifically, population-level analyses of neural recordings have revealed that representations often lie on structured manifolds (Khona and Fiete, 2022; Nogueira et al., 2023; Sohn et al., 2019), challenging the reductionist assumption in sparse coding that computations occur via independently firing, monosemantic features (Eichenbaum, 2018; Saxena and Cunningham, 2019; Barack and Krakauer, 2021). This motivated a paradigm shift towards more structured analysis protocols—methods designed around the generative process of the behavior one aims to explain (Schneider et al., 2023; Chen et al., 2018). Motivated by this and similar findings of intricate geometrical structure in neural network representations (Fel et al., 2025; Gurnee et al., 2025; Modell et al., 2025), we propose **Temporal SAEs**, a new protocol for interpreting language model activations that incorporates explicit inductive biases about temporal structure. Our approach decomposes activations at each timestep into two orthogonal components: a *predictable component*, obtained by projecting current representations onto past context using a learned attention mechanism, and a *novel component*, representing residual information orthogonal to the predictable component. That is, we assume the *novel* component—not the total representation—is uncorrelated over time. This allows correlations between total codes and hence enables our method to capture the temporal

structure of LM activations. Overall, we argue **interpretability methods should be driven by the behavior one is trying to explain**.

## 2 PRELIMINARIES

**Notations.** Let bold, lowercase letters represent vectors (e.g., $z$). Subscripts on vectors denote different samples (e.g., $z_i$), while superscripts denote the index within the vector, leading to a scalar (e.g., $z^k$). We denote model activations by $x \in \mathbb{R}^n$, SAE latents (sparse code) by $z \in \mathbb{R}^M$, and the dictionary by $D \in \mathbb{R}^{n \times M}$ ($M$ is the dictionary size).

**Sparse Coding.** Sparse dictionary learning (Olshausen and Field, 1996; 1997) expresses data as a sparse linear combination of dictionary elements, where both the weights and dictionary are learned from data. Intuitively, the dictionary behaves as a data-adaptive overcomplete basis; i.e., it typically has more elements that the dimension of ambient space. The optimization problem involved in this framework is $\arg\min_{D,z} \frac{1}{N} \sum_{i=1}^{N} \|x_i - Dz_i\|_2^2 + \lambda \mathcal{R}(z_i)$, where $\mathcal{R}(\cdot)$, typically chosen to be the $\ell_1$-norm, is a sparsity-inducing regularizer. Sparsity assists in picking the fewest most relevant dictionary atoms to explain a given data point.

**Sparse Autoencoders (SAEs).** SAEs (Shu et al., 2025) aim to disentangle (Bengio et al., 2013; Higgins et al., 2018; Olah, 2023) neural network activations into human-interpretable concepts (Cunningham et al., 2023; Bricken et al., 2023). Specifically, SAEs transform their inputs (i.e., neural network activations) into a latent representation which is encouraged to be sparse. As shown by Hindupur et al. (2025), this is achieved by solving the sparse coding problem using a specific parametric form for the sparse codes:

$$\arg\min_{D,z} \frac{1}{T} \sum_{i=1}^{T} \|x_i - Dz_i\|_2^2 + \lambda \mathcal{R}(z_i), \quad \text{s.t.} \quad z_k = f_{\texttt{SAE}}(x_k) \ \forall k, \ \tilde{g}(z_1, \ldots, z_T) = 0, \quad (1)$$

where $\mathcal{R}(\cdot)$ is the regularizer (typically the $L_1$ norm), $f_{\texttt{SAE}}$ is the SAE encoder architecture, and $\tilde{g}(\cdot)$ captures SAE-specific sparsity constraints on $z$. $f_{\texttt{SAE}}$ is typically a single hidden layer as in the ReLU SAE (Bricken et al., 2023; Cunningham et al., 2023), TopK SAE (Gao et al., 2024; Makhzani and Frey, 2013), JumpReLU SAE (Rajamanoharan et al., 2024) and BatchTopK SAE (Bussmann et al., 2024), though recent work has also explored alternative architectures inspired by sparse coding algorithms to capture specific structures, e.g., hierarchies (Muchane et al., 2025; Costa et al., 2025).

## 3 TEMPORAL STRUCTURE IN LANGUAGE MODEL ACTIVATIONS

To contextualize the prior assumptions made by SAEs about temporal structure in LMs' activations, we first perform an empirical characterization of such temporal structure in pretrained LMs. Specifically, since LMs are trained to generate coherent text by learning the distribution of natural language, one can expect their representations capture the rich phenomenology of its sequential structure (Elman, 1990); indeed, recent work has in fact found LM representations to be predictive of human neural recordings during language comprehension (Hosseini et al., 2024; Schrimpf et al., 2021; Tuckute et al., 2024; Hong et al., 2024; Georgiou et al., 2023). Motivated by this, we perform two experiments relevant to our discussion (see Fig. 2): (i) *measuring intrinsic dimensionality*—an approximation of the number of concepts necessary to explain the data, which can be expected to increase in a monotonic manner with time (Zhong et al., 2024; Can, 2025; Barak and Tsodyks, 2014; Meister et al., 2021)—and (ii) *signal nonstationarity*—which assesses whether model activations reflect the contextual relations between phrases of a passage (Zacks et al., 2007).

**Increasing intrinsic dimensionality.** Fig. 2 (a,e) show the dimensionality of the underlying manifold structure (intrinsic dimension) in model activations. We estimate the intrinsic dimensionality at a fixed position across a set of sequences with the U-statistic (App. Sec. J.2). For language model activations, this metric increases steadily with sequence position. On the other hand, a stationary surrogate of the data (see App. Sec. J.3) shows nearly constant intrinsic dimension over time. This indicates that model activations get 'denser', i.e., they possess more information over time. Correspondingly, the number of concepts needed to explain them varies with context.

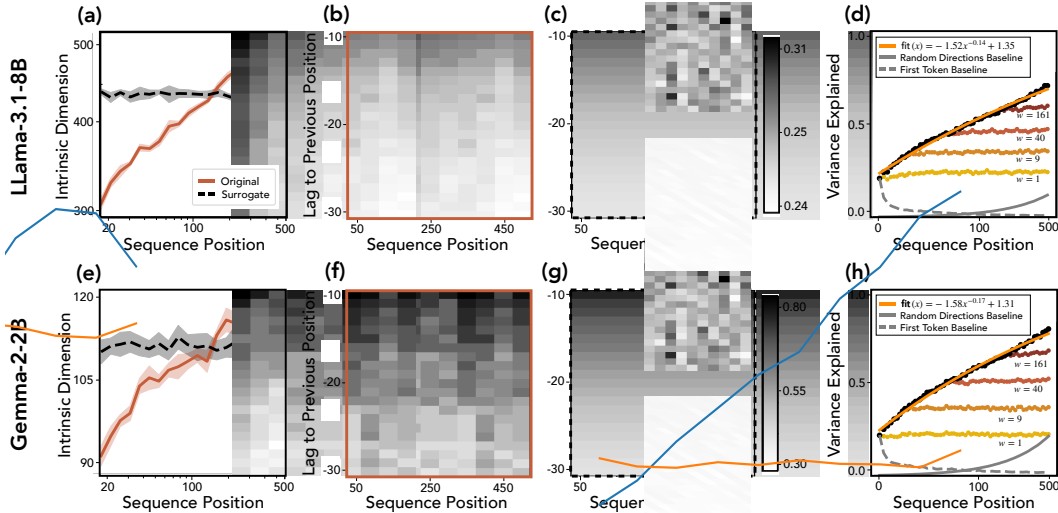

Figure 2: **Temporal structure of LLM activations reveals nonstationarity.** We use Pile samples (Monology, 2021) to analyze temporal structure from activations of two pretrained LMs, comparing it to a surrogate signal that is stationary in nature (see App. J.3). (**a, e**) Intrinsic dimension of model activations and stationary surrogate. (**b, f**) Autocorrelations $A(\boldsymbol{x}_t, \boldsymbol{x}_{t-\tau})$ as a function of sequence position ($t$) and lag ($\tau$). (**c, g**) Autocorrelation of the stationary surrogate. (**d, h**) Variance explained by projecting current representation $\boldsymbol{x}_t$ onto past context window $\{\boldsymbol{x}_{t-1}, \ldots, \boldsymbol{x}_{t-w}\}$ with different sizes $w$, along with a baseline. Results consistently show representations getting 'denser' over time and being significantly more structured than a stationary surrogate.

**Non-stationarity: Context explains bulk of signal variance.** Subplots (b), (f) show the autocorrelation of model activations (App. Sec. J.1), which is noticeably different at different sequence positions (x-axis), while the stationary surrogate, as expected, shows nearly position-invariant autocorrelation values (subplots (c), (g)). This finding is a clear signature of time-dependent correlation structure, and therefore of non-stationarity. We quantify the similarity of a representation with its context in subplots (d), (h). Specifically, we project representations of token $\boldsymbol{x}_t$ at a given time $t$ onto the subspace spanned by preceeding representations in the context $\{\boldsymbol{x}_{<t}\}$. These subplots show that up to $80\%$ variance of $\boldsymbol{x}_t$ is explained by a context of 500 tokens, further highlighting strong cross-temporal correlations. Significant variance in the representation at time $t$, $\boldsymbol{x}_t$, can be predicted (expressed) using representations from the past context.

## 4 TEMPORAL PRIORS OF SPARSE AUTOENCODERS

We now state the prior assumptions made by existing SAEs regards sequential structure in an input, contrasting these assumptions with the empirical results shown in Sec. 3. Specifically, building on the arguments used by Olshausen and Field (1997) to formalize the problem of sparse coding, we note that the SAE training objective (Eq. 1) can be interpreted from a Bayesian lens: minimize the negative log posterior $\arg\min_{\{\boldsymbol{z}_t\}} - \log P(\boldsymbol{z}_1, \ldots, \boldsymbol{z}_T \mid \boldsymbol{x}_1, \ldots, \boldsymbol{x}_T)$ of the data, which, by Bayes' rule, can be written as the sum of log likelihood (MSE) and log prior (the regularizer $\mathcal{R}$). From this lens, SAEs' prior assumptions on sequential structure in LM activations can be described as follows.

**Proposition 4.1** (Independence prior over time)**.** *Consider the SAE maximum aposteriori (MAP) objective from Eq. 1. Since the sparsity constraints are additive over time, this objective has an independent and identically distributed (i.i.d.) prior over time:*

$$P(\boldsymbol{z}_1, \ldots, \boldsymbol{z}_T) \propto \prod_{t=1}^{T} \exp\left(-\lambda\mathcal{R}(\boldsymbol{z}_i) - \tilde{\lambda}\tilde{g}(\boldsymbol{z}_i)\right) = \prod_i P(\boldsymbol{z}_i). \quad (2)$$

A more precise version of the claim for specific SAE architectures is provided in Appendix I.1. Intuitively, the claim above says that SAEs assume an independence of latents, and hence the concepts underlying the generative process of language, over time. This directly conflicts with the rich contextual structure of LM activations we empirically observed in Fig. 2b–d, f–h. Crucially, this

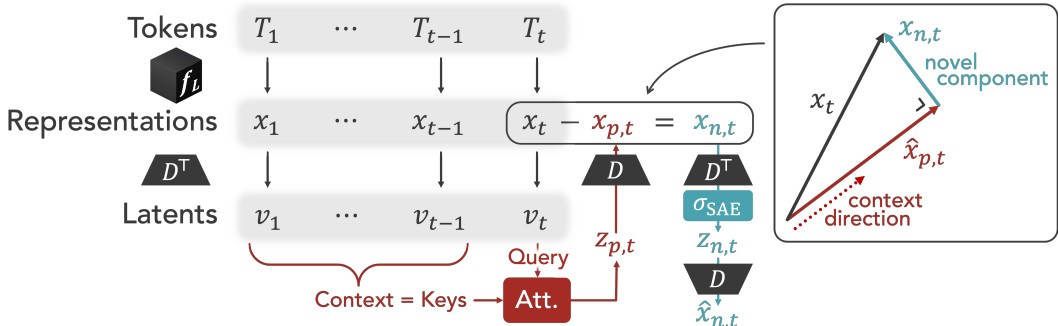

Figure 3: **Schematic of Temporal SAEs.** Temporal SAEs decompose activations $x_t$ into two components: a predictable component, obtained by projecting $x_t$ onto a context direction (derived from the past $x_{<t}$ using attention), and a sparse, novel component orthogonal to the predictable component that captures new information seen at time $t$.

also implies that SAEs assume the sparsity of latent codes necessary to explain model activations to be time-invariant, as stated formally in the corollary below.

**Corollary 4.1.1** (Assumptions of time-invariant sparsity). *As a consequence of the i.i.d. priors over time from Prop. 4.1, standard SAEs assume that sparsity of representations emerges from a fixed distribution independently over time (i.i.d.), i.e.,* $P(\|z_1\|_0, \ldots, \|z_T\|_0) = \prod_t P(\|z_t\|_0)$.

SAEs thus assume that sparsity—which, in their underlying generative model of activations corresponds to the number of concepts necessary for explaining the data (Bricken et al., 2023; Elhage et al., 2022)—remains approximately constant over time. This again does not align with the increasing dimensionality of representations observed in LM activations (see Fig. 2a,e). Correspondingly, if enough concepts aggregate over context such that a model's activations become 'denser' than the assumed sparsity budget, the assumption of time-invariant sparsity implies SAEs can fail to capture the temporal structure inherent in language—as is arguably already observed empirically with phenomena like feature splitting (Chanin et al., 2025; Bussmann et al., 2025; Bricken et al., 2023; Shu et al., 2025). We empirically validate this claim in Sec. 6.

## 5 TEMPORAL SAEs: EXPLICITLY MODELING TEMPORAL PRIORS

As stated in Sec. 2, sparse coding, a framework designed in computational neuroscience to understand neural representations in biological brains (Olshausen and Field, 1996; 1997), inspired SAEs as a framework for interpreting artificial neural networks (Bricken et al., 2023; Cunningham et al., 2023). In fact, the parallels between these communities can be made deeper: motivated by observations of intricate geometry of neural representations derived out of multi-dimensional population analyses (Khona and Fiete, 2022; Nogueira et al., 2023; Sohn et al., 2019), there were calls in computational neuroscience to discard the limiting reduction assumed in sparse coding that computations occur via a set of independently firing, monosemantic features (Eichenbaum, 2018; Saxena and Cunningham, 2019; Barack and Krakauer, 2021; Seung, 1996; Chung and Abbott, 2021)—similar to our arguments in Sec. 3, 4 (and results that follow in Sec. 6). Correspondingly, a need for more structured protocols was suggested (Eichenbaum, 2018; Barack and Krakauer, 2021), leading to methods that were motivated by the generative process of the behavior one is trying to explain (Schneider et al., 2023; Chen et al., 2018; Chen, 2019; Wiskott and Sejnowski, 2002; Linderman et al., 2017). We argue a similar paradigm shift is needed in language model interpretability: given that we train models to learn the distribution of highly structured data, we ought to embrace the fact that neural network activations can exhibit intricate geometrical organization. In what follows, as an attempt to qualify our arguments, we propose *one* such approach that focuses on the temporal structure of LM activations.

**Temporal SAEs.** In computational neuroscience, when analyzing data from dynamical domains (e.g., audio, language, or video), a commonly made assumption is that there is contextual information present in the recent history that informs the next state—this part of the signal is deemed *predictable* (Chen et al., 2025; Millidge et al., 2024), *slow-changing* (Berkes and Wiskott, 2005),

*invariant* (Olshausen and Cadieu, 2007), or *dense* (Tasissa et al., 2022). Meanwhile, the remaining signal corresponds to new bits of information added by the observed state at the next timestep—this part can be deemed *novel / surprising*, *fast-changing*, *variant*, or *sparse* with respect to the context. We argue LM activations are amenable to a similar generative model. Specifically, our observations in Sec. 3 show that activations $x_t$ at time $t$ are strongly correlated with the context and can be decomposed into two such parts. We thus propose the following generative model of LM activations:

$$x_t = x_{p,t} + x_{n,t}, \quad \text{where} \quad x_{p,t} = Dz_{p,t} \text{ and } x_{n,t} = Dz_{n,t}. \tag{3}$$

In the above, $x_{p,t}$ denotes a *predictable* component of the signal that captures the correlations of $x_t$ with past data $\{x_{<t}\}$, while $x_{n,t}$ denotes a *novel* component that represents new information added by the current token $x_t$. To obtain $z_{p,t}$, we project $x_t$ onto $\{x_{<t}\}$ to explain the predictable variance in $x_t$ as a convex combination of past data. Specifically, we use a self-attention layer $f$ on top of a single ReLU layer, yielding $z_{p,t} = f(\{x_1, \ldots, x_{t-1}\}, x_t)$. Meanwhile, $z_{n,t} = \tilde{f}(x_t, z_{p,t}) = \sigma(D^T(x_t - Dz_{p,t}))$ captures the residual component of the code, which is not correlated with the past (note that $\sigma$ is the nonlinearity). We use a standard SAE encoder (either TopK or BatchTopK) to instantiate $\sigma$, applying it to $x_t - Dz_{p,t}$ to derive $z_{n,t}$. See Fig. 3 for an overall schematic of the encoding process. The learning objective in Temporal SAEs follows.

$$\underset{D,z}{\arg\min} \frac{1}{T} \sum_{i=1}^{T} \|x_i - D(z_{p,i} + z_{n,i})\|_2^2 + \lambda \mathcal{R}(z_{n,i}), \tag{4}$$

$$\text{s.t. } z_{p,k} = f_{\texttt{SAE}}(\{x_{<k}\}, x_k), \ z_{n,k} = \tilde{f}_{\texttt{SAE}}(x_k, z_{p,k}), \ z_k = z_{p,k} + z_{n,k} \ \forall k.$$

Relating to Sec. 4, we note the prior assumption in Temporal SAEs is that the residual $z_{n,t} = z_t - z_{p,t}$, which captures the novel information in $x_t$ remaining after removing the projections onto the past context, is *i.i.d.* over time. This prior allows temporal correlations between codes $z$, and thereby allows correlations between concepts across time, instead of assuming them to be temporally independent.

**Sanity Checking Temporal SAEs.** Before analyzing how different approaches represent the temporal structure of language, we demonstrate that Temporal SAEs performs on par with SAEs on standard metrics such as reconstruction error. Specifically, we train a Temporal SAE and standard SAEs (ReLU, TopK, Batch-TopK) on 1B token activations extracted from Gemma-2-2B (Team et al., 2024) from the Pile-Uncopyrighted dataset (Monology, 2021). We also analyze a baseline of the prediction only module from Temporal SAEs, reported as 'Pred. only', which can be expected to underperform since predicting the next-token representation is likely to be more difficult than reconstructing it. Results are provided in Tab. 1, 2 and show competitive performance between all protocols, except Pred. only. One can also assess which part of a fully trained Temporal SAE is more salient in defining its performance, i.e., does the estimated predictive part $\hat{x}_{p,t}$ contribute more to the reconstruction $\hat{x}$ or does the estimated novel part $\hat{x}_{n,t}$. Results are reported in Tab. 3. We see that the error vectors, i.e., $x - \hat{x}_p$ and $x - \hat{x}_n$, are approximately orthogonal, suggesting the modules capture separate bits of information from the input—this is inline with results by Costa et al. (2025), who show optimizing to reconstruct residuals (as we do with the novel code) *can* lead to orthogonal codes at different stages of an SAE. Furthermore, we find that a bulk of the reconstructed signal $\hat{x}_t$ (in the sense of norm) is captured by the predictive code—in fact, the percentage contribution of the predictive code is $\sim 80\%$, in line with numbers observed in Fig. 2d. However, analyzing the reconstruction performance, we see the

Table 1: Temporal SAEs achieves NMSE similar to standard SAEs across domains (Simple Stories, Webtext, Code).

|  | ReLU | TopK | BTopK | Pred. Only | Temporal |
|---|---|---|---|---|---|
| Story | 0.20 | 0.155 | 0.152 | 0.34 | 0.139 |
| Web | 0.19 | 0.144 | 0.139 | 0.36 | 0.139 |
| Code | 0.20 | 0.154 | 0.149 | 0.38 | 0.152 |

Table 2: Temporal SAEs explains similar amount of signal variance as standard SAEs.

|  | ReLU | TopK | BTopK | Pred. Only | Temporal |
|---|---|---|---|---|---|
| Story | 0.60 | 0.71 | 0.72 | 0.29 | 0.73 |
| Web | 0.69 | 0.78 | 0.79 | 0.40 | 0.79 |
| Code | 0.65 | 0.75 | 0.75 | 0.33 | 0.75 |

Table 3: predictive and novel codes explain different parts of the input signal across domains.

|  | Sim. | % Norm | | NMSE | | Var. Expl. | |
|---|---|---|---|---|---|---|---|
|  |  | Pred. | Novel | Pred. | Novel | Pred. | Novel |
| Story | -0.02 | 76.2 | 23.5 | 0.53 | 4.03 | 0.11 | 0.64 |
| Web | -0.02 | 80.5 | 19.5 | 0.49 | 4.28 | 0.17 | 0.66 |
| Code | -0.02 | 74.2 | 26.0 | 0.57 | 3.84 | 0.14 | 0.65 |

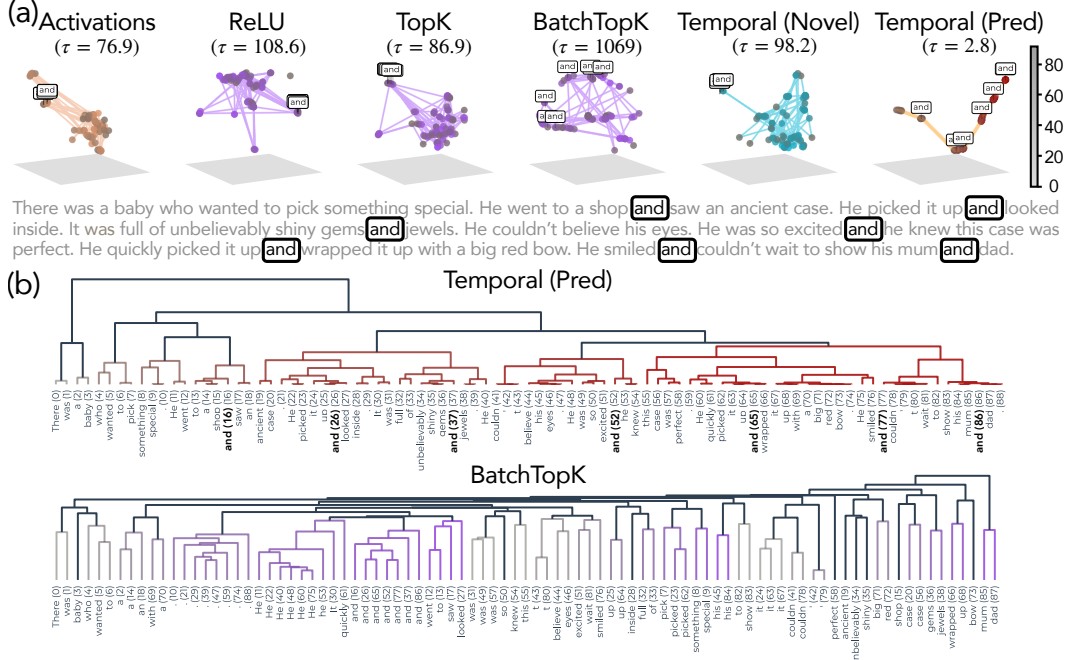

Figure 4: **Temporal SAEs unroll stories, decomposing into events.** We consider model activations from a story and compute pairwise similarity of codes extracted from different interpretability protocols. (a) We see predictive codes from Temporal SAE organizes in hierarchical block structures that seem to align with (sub)event boundaries in the analyzed story, while the novel code primarily emphasizes sudden changes in the narrative; meanwhile, standard SAEs show a mixture of the two structures, with a stronger similarity to the structure exhibited by the novel codes. (b) We confirm the alignment of predictive codes with event boundaries by running an off-the-shelf hierarchical clustering algorithm, finding the token clusters indeed correspond to (sub)events occurring in the story as the narrative proceeds. Running this process on SAEs, we find this process yields temporally incoherent clusters that are primarily defined by lexical information.

predictive component primarily contributes to achieving a good NMSE, while the novel component is more responsible for explaining the input signal variance. These results align with the generative model assumed in Temporal SAEs (Eq. 3). Specifically, NMSE captures the average reconstruction, and hence a slower moving, contextual signal can expect to dominate its calculation. Meanwhile, variance assesses changes per dimension and timestamp in the signal, which better matches the inductive bias imposed on the novel part.

## 6    CAPTURING TEMPORAL STRUCTURE WITH TEMPORAL SAE

We now evaluate the ability of different SAEs to capture temporal dynamics in language model representations. We first analyze a narrative setting where, locally in time, one can expect a lot of correlated structure as an event transpires, with strict event boundaries delineating events from each other. As we show, TemporalSAEs yield a clear delineation of the slow-moving local information from the fast-changing boundaries, while standard SAEs generally ignore the slow-moving information to optimize for the faster changes. The narrative evaluation assesses how different SAE codes represent local/global semantic information. We now investigate local/global syntactic information by analyzing SAE codes extracted from garden path sentences. Here, we find that unlike other SAEs, the use of a predictive module in TemporalSAEs yields codes that relate tokens from garden path sentences in a manner that would align with the ultimately correct (rather than garden-path) parses—an ability that language models are in fact known to possess, but that standard SAEs seem to not explain (Li et al., 2024; Hanna and Mueller, 2024).

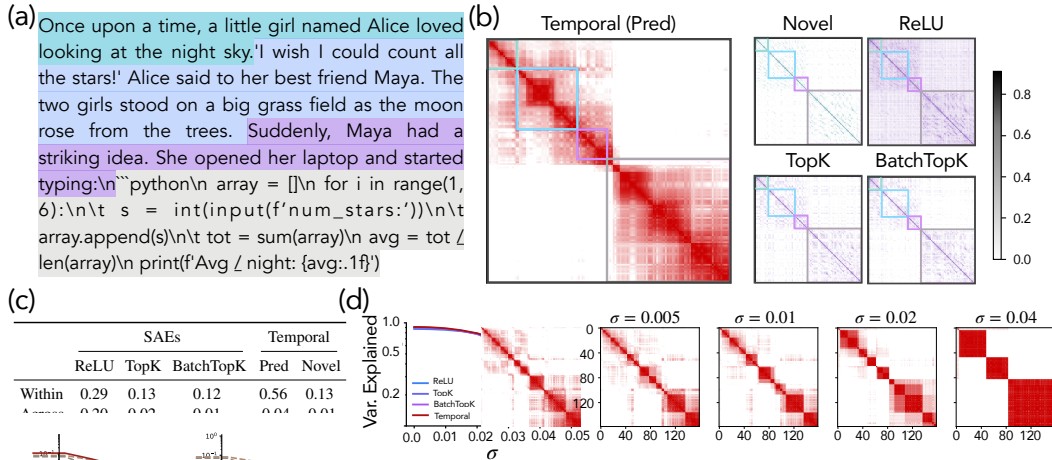

Figure 6: **Predictive codes decompose stories into events.** (a) We consider model representations from a synthetic story with well-defined event boundaries. (b) Computing pairwise cosine similarity of latent codes extracted using different protocols, we see the predictive code of Temporal SAE organizes in hierarchical block structures that seem to align with (sub)event boundaries in the analyzed story. (c) We confirm the alignment of predictive codes with event boundaries by computing average pairwise similarity of token latent codes for tokens that span the same event ('within') versus not ('across'). Results clearly show high within-event similarity scores for the predictive code. (d) The results above are further corroborated by running a noising process on the latent codes: we add Gaussian noise of scale $\sigma$ to the input before computing latent codes, defining the similarity maps and computing explained variance of un-noised data. This process elicits coarser grained clusters from the similarity maps for the predictive code, suggesting the multi-scale temporal structure of stories is reflected in predictive codes.

## 6.1    THE GEOMETRY OF STORIES: A NARRATIVE-DRIVEN DOMAIN

**UMAP of Latent Codes Suggests Models Temporally Straighten Activations.**    We consider the TinyStories datasets (Eldan and Li, 2023) for its relatively straightforward narrative structures, and qualitatively analyze the geometry of latent codes extracted from model activations when processing these stories. Visualizing the latent codes in a low-dimensional basis via a 3D UMAP projection (McInnes et al., 2018), we see SAEs yield a highly irregular and unstructured geometry (see Fig. 4a). Calculating Tortuosity (Bullitt et al., 2003), a measure of how aligned local arcs are with respect to the global structure of a curve, we see very high values emerge for SAEs' latent codes geometry, suggesting sudden changes in the local similarity as a story unravels. To further understand the results above, we highlight a specific token ('and') from the story, the UMAP analysis shows that standard SAEs generally just cluster tokens by lexical identity. This is further corroborated by running a hierarchical clustering algorithm on the latent codes (SciPy, 2025), finding temporally incoherent, but lexically related clusters (see Fig. 4b).

**Quantifying Similarity to Slow vs. Fast Moving Signals.** To further quantify the straightening claim, we compute the Fourier transform of the model activations and divide the frequency spectrum into two halves at a critical frequency $f_c$ such that the energy (i.e., sum of squared value of phase information) in the frequencies below $f_c$ equals that of the remaining ones. We call the first split "slow part" of a sequence, and latter the "fast part". We then compute the correlation matrix defined by the slow and fast parts, compute their spectrum, and

Table 4: **Similarity between latent codes and model activations.** Predictive codes from Temporal SAEs align better with slow-changing part of activations; novel codes and standard SAEs' codes align with fast-changing part.

|  | SAEs | | | Temporal | |
|---|---|---|---|---|---|
|  | ReLU | TopK | BatchTopK | Novel | Pred |
| Slow | 0.37 | 0.35 | 0.35 | 0.19 | 0.75 |
| Fast | 0.54 | 0.54 | 0.54 | 0.75 | 0.18 |

Figure 5: **Kernel spectrum for latent codes and model representations**. Kernels defined using novel code from Temporal SAE and standard SAEs both align well with the fast-changing part of model representations; meanwhile, only the predictive code shows strong similarity to the slow changing part.

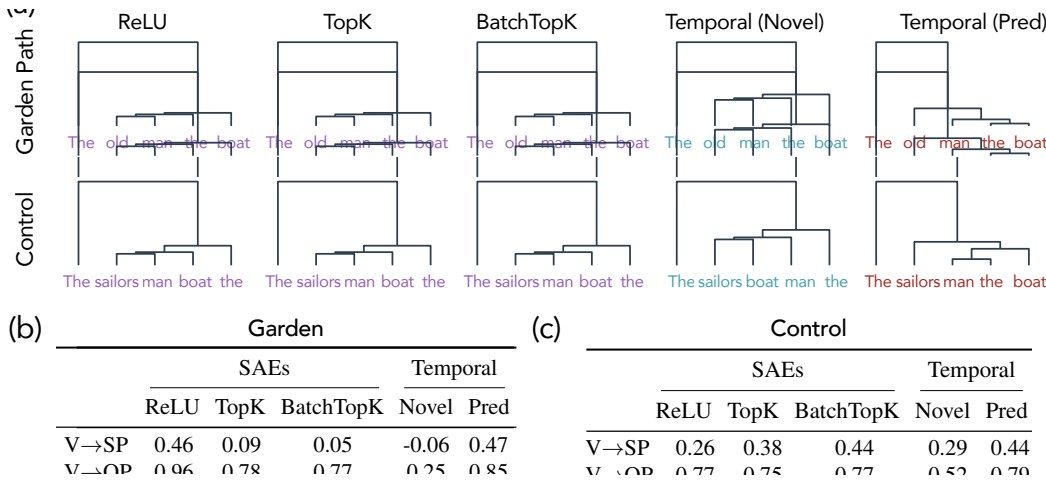

Figure 7: **Hierarchically clustering SAE codes for garden path sentences.** (a) Pairwise similarity maps of the predictive code from Temporal SAE link long-distance heads and dependents that define the ultimately correct parse in garden path sentences, while standard SAEs (e.g., BatchTopK) emphasize only local, transient relations, falling for the misleading cues. (b, c) Comparing the cosine similarity of average latent code extracted from the subject phrase (SP), verb phrase (V), and object phrase (OP), we see across ambiguous garden path sentences and unambiguous control variants thereof, only the predictive component of Temporal Features Analyzers shows consistent similarity scores (as expected if the SP and V ambiguity is reflect in the latent codes).

analyze how similar these spectrums are to the ones defined using different SAEs' codes. Results are shown in Fig. 5. We clearly see the spectrum extracted from the predictive component of Temporal SAE approximates that of the slow part of the representation, while the novel component's spectrum is similar to that of the fast moving part. Meanwhile, spectra of SAEs only exhibits similarity to the fast part, suggesting they do not capture longer range dependencies necessary for interpreting narrative-driven texts like most language domains. To quantify this further, we also measure kernel similarities (CKA) between the kernels respectively defined by the slow moving and fast moving signal to the different SAE codes. Results clearly show Standard SAEs are primarily similar to the fast moving signal.

**Predictive Component Captures Local Event Boundaries.** The results above demonstrate Temporal SAEs' predictive component qualitatively align with event boundaries in a story. To investigate this result more quantitatively, we use GPT-5 to create a synthetic dataset of 50 stories with well-defined event boundaries (see Fig. 6a for an example). We extract latent codes for these stories' tokens, center them by subtracting the mean to remove any globally shared information, and compute the cosine similarity of token to token latent codes. If the latent codes reflect local event structure of a story, the cosine similarity (on average) will be high between token pairs that come from the same event and low (if not zero) between pairs sampled across events; see Fig. 6b for an example similarity map corresponding to the story shown in Fig. 6a. Results are shown in Fig. 6 (c) and corroborate our qualitative findings: we see predictive components from Temporal SAE show substantially higher similarity of codes if tokens are sampled from within an event, while the novel component and SAEs generally show low similarity between two tokens. These results are further supported by the robustness of Temporal SAEs to noise. Specifically, we see that when we add noise to the input data, which, on average, will lead to turning off of latents with small magnitudes (due to the encoding nonlinearity), the temporal structure of the data, if it is present, will be amplified. We see precisely this effect in Fig. 6d: the predictive components' cosine similarity map under Gaussian noised input maps elicits coarser block structures with increasing noise scale; this is reminiscent of percolation or heat diffusion perspectives on graph clustering, wherein noise diffuses only within a connected component and hence community structure is elicited (Von Luxburg, 2007). Correspondingly, we see

Temporal SAEs respond most gracefully to noise: variance explained reduces slower than SAEs', which in fact drops to ~0 at some scale.

## 6.2 GARDEN PATH SENTENCES: AMBIGUITIES RESOLVED VIA TEMPORAL STRUCTURE

Garden-path sentences—e.g., `The old man the boat`—initially cue an incorrect local parse before a later token forces reanalysis. Language models have been shown to be able to correctly parse such ambiguous sentences, offering in fact a predictive account of human per-token surprisals (Li et al., 2024; Hanna and Mueller, 2024; Oh and Schuler, 2023). Interestingly, when using SAE codes to assess whether LLM representations offer a valid parse of the sentence, we find hierarchical clustering of SAE codes yields a parse that is suggestive of the misleading cue; meanwhile, Temporal SAE recovers the correct parse by separating the predictable, slow-moving component of the representation from the novel, fast-changing residual. Specifically, in Fig. 7, we see the predictive codes link long-distance heads and dependents that reflect the correct parse (e.g., `man` as verb), producing coherent similarity structure over the full span, whereas standard SAEs emphasize only transient, local changes and miss these cross-temporal constraints. These results suggest Temporal SAEs encode syntactic structure that unfolds over time when evidence to collapse the correct constituent parse emerges. To make these results more quantitative in nature, we use GPT-5 to synthetically generate a set of 50 garden path sentences where the subject is ambiguous. We create 50 control variants of these sentences such that the controls do not possess ambiguity with respect to typical parse of the sentence constituents: e.g., changing the subject in the sentence `The old man the boat` from `old` to `sailors`, yielding `The sailors man the boat`. We then divide all sentences into their respective subject phrase (SP), verb phrase (V), and object phrase (OP): e.g., `The old` (SP), `man` (V), and `the road` (OP). We compute the average latent code for tokens from these three constituent phrases, under the hypothesis that if the sentence ambiguity is reflected in the latent code, then until the OP shows up, the correct parse of prior words cannot be identified. Correspondingly, all valid parses must be stored in the same representation. This suggests the cosine similarity of latent codes of V and SP tokens should be of a similar order in both the garden path and control sentences; meanwhile, the similarity between V and OP should be much higher than V and SP. Results are reported in Fig. 7 (b,c). We clearly see extreme sensitivity in similarity values of SAE latent codes and the novel component of Temporal SAE, but the predictive component is essentially invariant across sentence type, suggesting it captures the temporal dynamics likely relevant for a LM to parse garden path sentences.

## 7 DISCUSSION

Our findings reinforce the broader lesson that interpretability tools must align their inductive biases with the statistical structure of the data they are applied to. We showed that standard SAEs impose independence priors across time, which are fundamentally misaligned with the nonstationary and context-dependent structure of language model activations. This mismatch explains why existing SAEs tend to underrepresent temporal dependencies, despite capturing other kinds of structure. By contrast, Temporal SAE incorporates empirically observed correlations across time as an inductive bias. Its decomposition of activations into predictable (slow-moving) and novel (fast-changing) components enables the recovery of temporal structure that standard SAEs fail to expose. In particular, we demonstrated that predictable codes align with stable, high-level information, while novel codes isolate transient or surprising information, allowing Temporal SAE to highlight event boundaries and syntactic reanalyses in garden-path sentences. Taken together, these results emphasize a general principle: interpretability methods should not be viewed as neutral feature extractors but as models with their own structural assumptions. When these assumptions mismatch the true data distribution, important aspects of representation may be obscured. Incorporating empirically motivated temporal priors offers one way to close this gap, suggesting that future progress in interpretability will require tailoring methods to the dynamics of the representations under study.

## ACKNOWLEDGMENTS

ESL thanks members of the Mechanisms team at Goodfire AI, particularly Tom McGrath, Owen Lewis, Jack Merullo, and Atticus Geiger for helpful discussions. ESL additionally thanks Jack Lindsey for useful comments that helped formalize the intrinsic dimensionality arguments in Sec. 4, and Raphael Sarfati for suggesting the Tortuosity metric used in Fig. 4. The authors also thank David Bau for a helpful discussion about the generalization ability of SAEs, and David Klindt for several useful comments on an earlier draft of the paper. ESL and SSRH further thank Noor Sajid and members of the CRISP lab at Harvard for helpful conversations. CR is supported by a MATS extension grant. AM's work is partially supported by the National Science Foundation (Grant No. 2530728) and Binational Science Foundation. This work has been made possible in part by a gift from the Chan Zuckerberg Initiative Foundation to establish the Kempner Institute at Harvard University.

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

# A EXPERIMENTAL DETAILS

Below, we discuss broad details of the experimental setup, training protocol, and measures used for evaluations.

## A.1 TRAINING, HARVESTING, AND INFERENCE FOR SAEs

**Compute.** All experiments were performed using 1 H100, both for extracting representations, analyzing them, and training Temporal and standard SAEs. To save training costs, we precache activations on a local server and train in an "online manner", i.e., sampling a new batch of activations every training iteration.

**SAEs' training.** All analyzed SAEs were trained from scratch on 1B precached activations from the Pile-Uncopyrighted dataset (Monology, 2021). While we did try to use existing, off-the-shelf SAEs, we found the results to be wildly inconsistent depending on where we borrowed the SAE from. To enable a consistent and fair evaluation, we thus preferred to train all SAEs from scratch. For Gemma models, activations were extracted from Layer 12; for Llama models, from layer 15 (all 0-indexed).

**Training procedure.** We use Adam optimizer with standard hyperparameters. We initialize training with a warmup of 200 steps to a learning rate of $10^{-3}$, following from thereon to a minimum of $9 \times 10^{-4}$, i.e., the learning rate remains essentially constant throughout training.

**Activations normalization.** Following best practices, we normalize activations to unit *expected* norm (Bricken et al., 2023; Costa et al., 2025). This helps put different SAEs on the same training scale, and especially for ReLU SAEs, makes training substantially easier by reducing the need for tuning regularization strength. On a related note, we emphasize our ReLU SAEs have a slightly higher $L_0$, i.e., are less sparse, than almost all other SAEs analyzed in this work.

## A.2 EXPERIMENTS WITH LLM REPRESENTATIONS: AUTOCORRELATION AND U-STATISTIC

We primarily analyze Gemma-2-2B and Llama-3.1-8B models in this paper. Specifically, we precache 10K activations for the analyzed datasets for our experiments on dynamics on language model representations.

### A.2.1 AUTOCORRELATION

We compute autocorrelation by selecting evenly spaced tokens across the sequence and measuring the cosine similarity between each token and tokens at various lags in the past. Specifically, for tokens at position $t$, we compute similarities to tokens at $t - w$ where lag $w$ ranges from 5 to 20. This creates a heatmap where rows represent lag offsets and columns represent token positions.

For a stationary process, we expect the autocorrelation pattern to remain consistent across time—that is, the relationship between a token and its historical context should be similar regardless of position in the sequence. This would manifest as similar autocorrelation patterns repeating horizontally across token positions. In contrast, for a non-stationary process where representations evolve over time, we expect the autocorrelation patterns to vary systematically across positions, with columns showing different temporal dependency structures as the sequence progresses.

### A.2.2 U-STATISTIC

We measure the effective dimensionality of LLM representations using a U-statistic (an unbiased estimator) based on pairwise cosine similarities. Let $\mathbf{X}_t = [\boldsymbol{x}_t^{(1)}, \ldots, \boldsymbol{x}_t^{(M)}]$ be $M$ samples of normalized activations at time $t$ (from different timeseries). $\mathbf{G}_t = \mathbf{X}_t^T \mathbf{X}_t$ is the Gram matrix. Using the above, a U-statistic of intrinsic dimension which we employ is defined below:

$$\text{U-stat}(t) = \frac{M^2 - M}{\|\mathbf{G}_t\|_F^2 - M} \tag{5}$$

where $\|\mathbf{G}_t\|_F^2$ is the squared Frobenius norm of the Gram matrix. This quantity estimates the effective rank $1/\mathrm{tr}(\mathbf{C}_t^2)$, where $\mathbf{C}_t = \mathbb{E}[\boldsymbol{x}_t \boldsymbol{x}_t^T]$ is the second moment matrix and $\boldsymbol{x}_t$ is the activation vector at time $t$. Under stationarity, U-stat remains constant. When representations evolve over time, U-stat increases systematically as more orthogonal directions become active.

### A.2.3 PROJECTION ANALYSIS

This analysis quantifies how much of the representation $\boldsymbol{x}_t \in \mathbb{R}^D$ at token position $t$ can be reconstructed from its preceding context $\{\boldsymbol{x}_i\}_{i \in W}$, where the context window $W = [t - w, t - 1]$ contains the previous $w$ tokens.

For a population of $B$ sentence samples, we compute the projection of each target representation onto the subspace spanned by its context. Let $\mathbf{X}_W^{(b)} = [\boldsymbol{x}_{t-w}^{(b)}, \ldots, \boldsymbol{x}_{t-1}^{(b)}]^T \in \mathbb{R}^{w \times D}$ denote the matrix of context representations for sample $b$, where each row is a context vector. Prior to projection, we center all representations by subtracting the mean computed over the target positions across samples. The projection of $\boldsymbol{x}_t^{(b)}$ onto the span of the context is:

$$\mathbf{c}_{t,w}^{(b)} = \mathrm{span}(\mathbf{X}_W^{(b)}) \cdot \boldsymbol{x}_t^{(b)} \tag{6}$$

where $\mathrm{span}(\mathbf{X}_W^{(b)})$ is the row space of $\mathbf{X}_W^{(b)}$, that can be computed via SVD. The variance explained by the context is:

$$\mathrm{expvar}(t, w) = \frac{\sum_{d=1}^{D} \mathrm{var}(\mathbf{c}_{t,w,d})}{\sum_{d=1}^{D} \mathrm{var}(\mathbf{x}_{t,d})} \tag{7}$$

where $\mathrm{var}(\mathbf{c}_{t,w,d})$ and $\mathrm{var}(\boldsymbol{x}_{t,d})$ denote the variance of the $d$-th dimension across the $B$ samples. This ratio measures the fraction of total representational variance that can be linearly reconstructed from the preceding context. Values approaching 1 indicate high predictability from context; values near 0 indicate representations largely orthogonal to their context subspace.

### A.2.4 SURROGATE

For the U-statistic and Autocorrelation metrics, we compare LLM activations to surrogate distributions that preserve certain statistical properties while removing temporal structure. We operate on representation vectors $\mathbf{X} \in \mathbb{R}^{B \times T \times d}$, where $B$ denotes batch size, $T$ denotes sequence length, and $d$ denotes the model dimension.

**U-statistic surrogate (Fig. 2 a, e):** For each sequence $i \in \{1, \ldots, B\}$, we construct the surrogate $\tilde{\mathbf{X}}_i$ by applying a random permutation $\pi_i : \{1, \ldots, T\} \to \{1, \ldots, T\}$ to the temporal positions:

$$\tilde{\mathbf{X}}_{i,t,:} = \mathbf{X}_{i,\pi_i(t),:}, \quad \forall t \in \{1, \ldots, T\}$$

This preserves the marginal distribution of activations within each sequence while destroying temporal dependencies.

**Autocorrelation surrogate (Fig. 2 c, g):** Given the similarity matrix $\mathbf{S} \in \mathbb{R}^{T \times T}$ where $S_{ij} = \mathrm{sim}(\mathbf{X}_{\cdot,i,\cdot}, \mathbf{X}_{\cdot,j,\cdot})$, we construct the surrogate similarity matrix $\tilde{\mathbf{S}}$ by replacing each diagonal with its mean:

$$\tilde{S}_{ij} = \bar{S}_k \quad \text{where } k = |i - j|, \quad \bar{S}_k = \frac{1}{T-k} \sum_{t=1}^{T-k} S_{t,t+k}$$

This preserves the average correlation structure at each lag while removing position-specific temporal patterns.

### A.3 ANALYSIS OF STORIES: UMAP PROJECTIONS, DENDROGRAMS, SIMILARITY HEATMAPS, AND EVENT BOUNDARY DETECTION

**Data.** For experiments in Sec. 6.1, we used either stories sampled from the TinyStories dataset (Eldan and Li, 2023) (for Fig. 4, 5) or synthetically sampled stories using GPT-5 (for Fig. 6). For the former, we sampled stories that were up to a 100 tokens long, allowing us to characterize compute

UMAP for both Gemma and Llama models without hitting memory bottlenecks. For consistency, we then used these stories for all experiments. For synthetically defined stories, we defined 3 in-context examples (one of which is shown in Fig. 6a) and prompted GPT-5 to produce 50 stories with similar such suddenly changing events. Stories varied in size, but were generally less than 150 tokens long. Following Georgiou et al. (2023), the generated stories' events were labeled using GPT-5. These labels served as the 'ground truth' event boundaries. We qualitatively analyzed and confirmed the event boundaries overlap with our intuitively expected event structure in the stories.

**Analysis Details.** To isolate the low-dimensional geometry of latent codes, we perform a 3D UMAP analysis using the open-source package (McInnes and Healy, 2025). Similarly, Dendrograms in Fig. 4 are computed using the hierarchical clustering package in SciPy SciPy (2025). Branches are colored based on proximity, which in our case was between 0–1 (since we use cosine similarity as the clustering measure). We put a proximity bound of 0.2 as the distance under which branches are colored within a cluster. For Fourier Analysis, as mentioned in Sec. 6.1, we performed a Fourier transform of the model activations for a given story, isolated its lower frequency components—defined as $0.1\times$ the Nyquist rate, which turns out to possess $\sim$50% of the signal norm—and compute the cosine similarity kernel of the low / high frequency spectra. These kernels are then compared to kernels of latent codes derived using Temporal or standard SAEs.

### A.4 ANALYZING GARDEN PATH SENTENCES: DENDROGRAMS AND PHRASE SIMILARITY

**Data.** We used GPT-5 to sample 50 garden path sentences and control variants thereof (1 corresponding to each sentence). Sentences were 20–40 tokens long and had a structure such that the observation of the object phrase resolved ambiguity. The precise ambiguity structure was were varied in type, i.e., the ambiguity could be resolved by altering the subject (e.g., `old` $\rightarrow$ `sailors` in `The old man the road`) or by other mechanisms such as adding punctuations to elicit a pause (e.g., adding a comma, such as `The old train the young fight` $\rightarrow$ `The old train, the young fight`). Control variants had a mixture of such resolutions applied.

**Analysis.** For assessing whether tokens in verb phrase relate more with the subject phrase, as would be expect by the typical parse in our used garden path sentences, or with the object phrase, as would be necessary for the correct parse, we computed Dendrograms using the hierarchical clustering package in SciPy (SciPy, 2025). The similarity between subject phrase (SP), verb phrase (V), and object phrase (OP) was computed by taking the set of tokens $T_P$ that belong to a phrase $P \in \{\text{SP, V, OP}\}$, and computing the average latent code: $\bar{c}(P) = \frac{1}{|P|} \sum_{t \in T_P} c_t$, where $c_t$ denotes the latent code extracted for token $t$ using either SAEs or the predictive or novel component of Temporal SAEs. This protocol is similar to popularly used strategies for computing sentence embeddings (see, e.g., work using BERT embeddings (Koroteev, 2021)). Cosine similarity between these phrase-averaged garden path / control sentences yields the tables shown in Fig. 7.

# B    FURTHER EVALUATION RESULTS

## B.1    U-STATISTIC ACROSS DOMAINS FOR GEMMA-2-2B

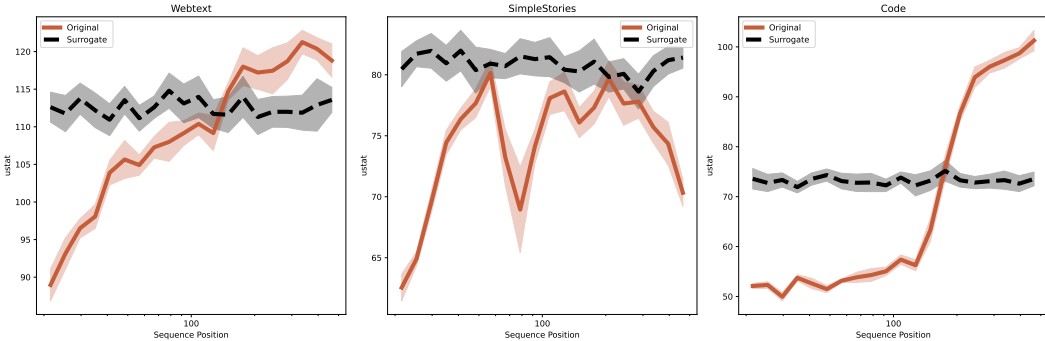

Figure 8: U-Statistic across domains for LLM activations, surrogate

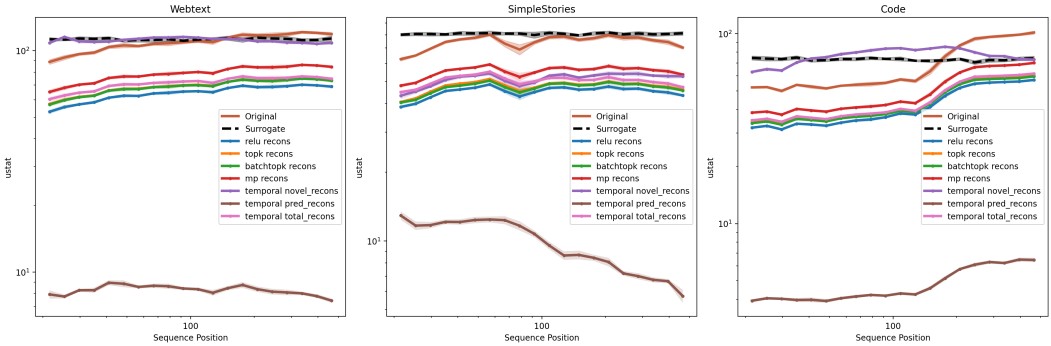

Figure 9: U-Statistic across domains for SAE reconstructions

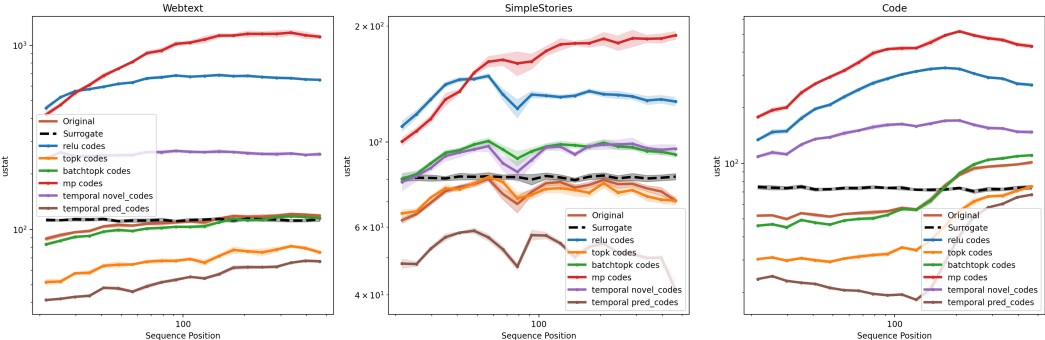

Figure 10: U-Statistic across domains for SAE codes

## B.2    AUTOCORRELATION ACROSS DOMAINS FOR GEMMA-2-2B

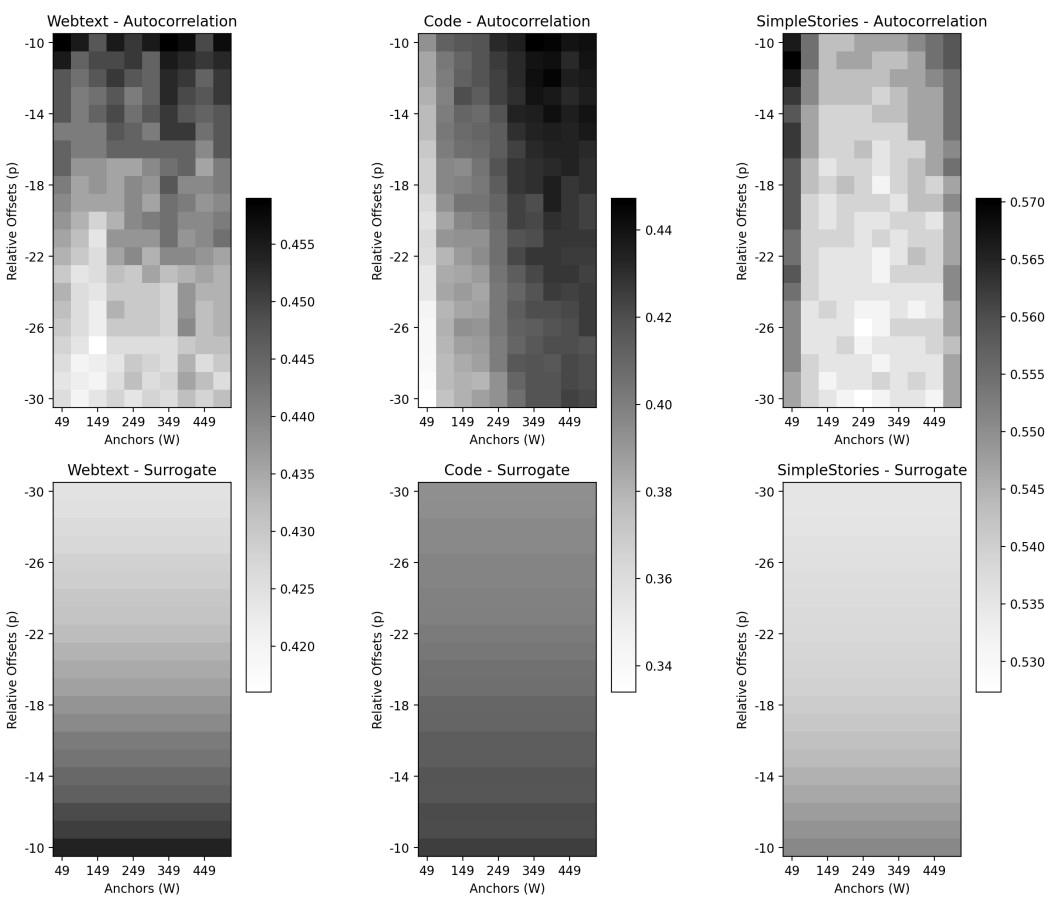

Figure 11: Autocorrelation of language model activations and a stationary surrogate across webtext, simple stories and code domains.

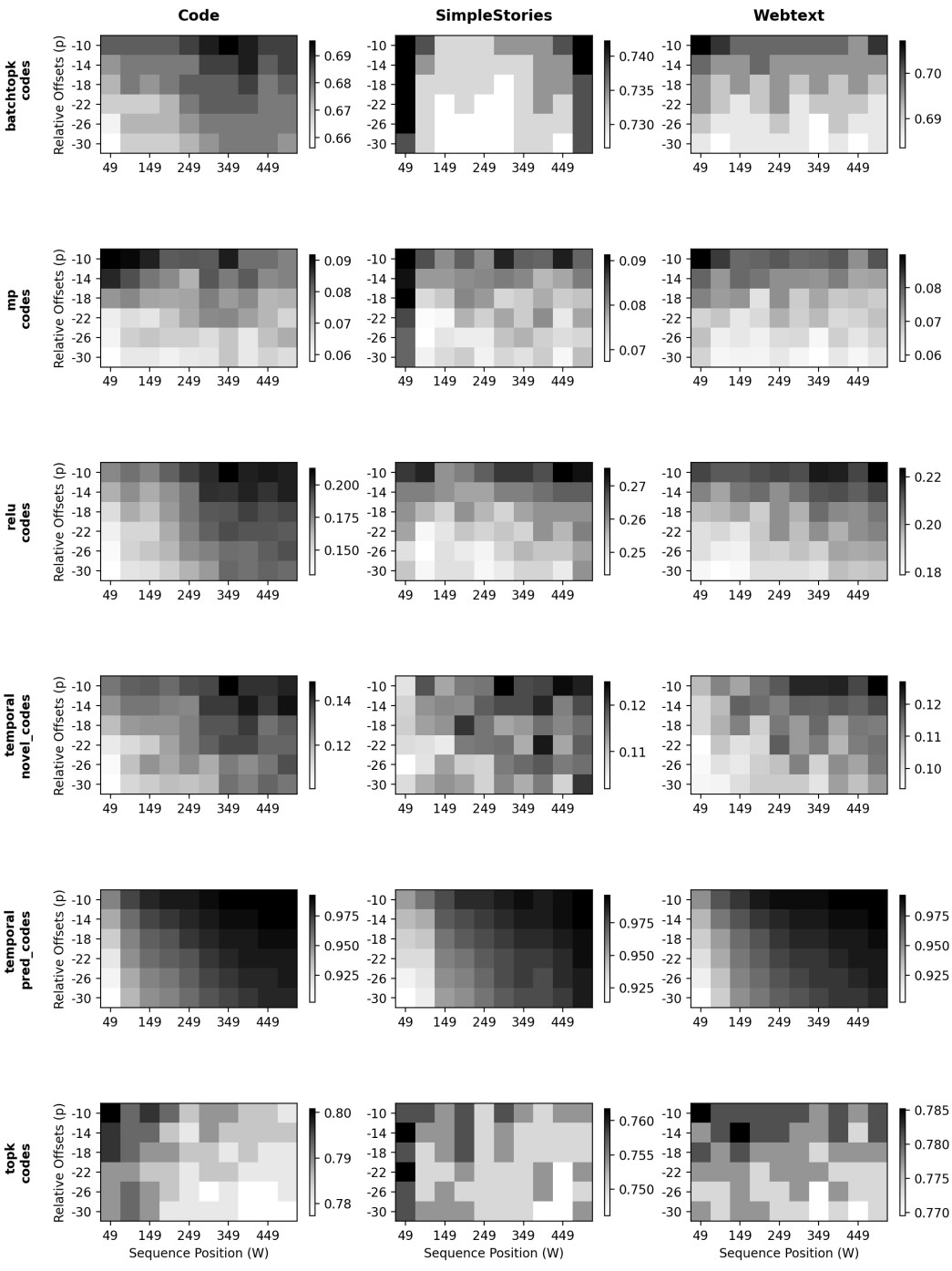

Figure 12: Autocorrelation of SAE latent activations across webtext, simple stories and code domains.

# C  RECONSTRUCTION PERFORMANCE OF SAES

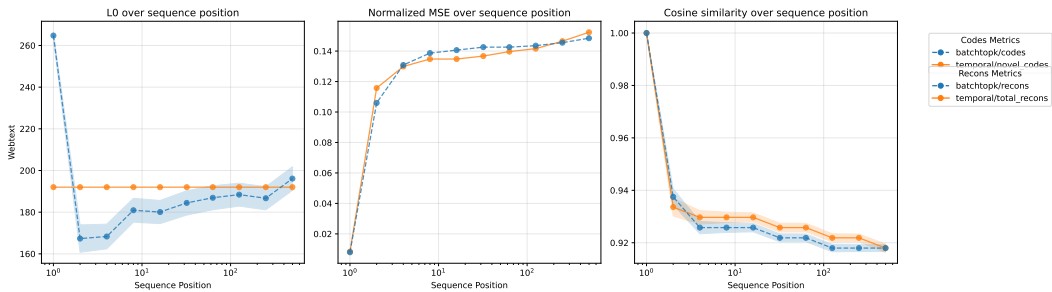

Figure 13: Attn SAE vs BatchTopK

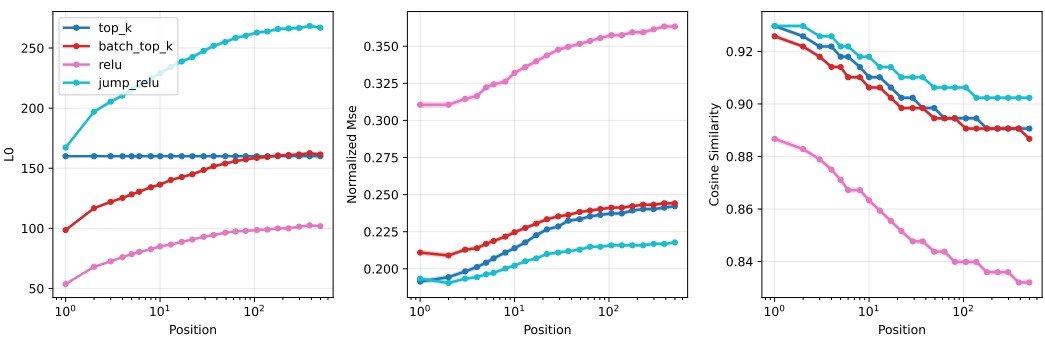

Figure 14: Reconstruction performance of LLama-3.1-8B resid post Layer 16 (1/2 forward pass) SAEs on webtext.

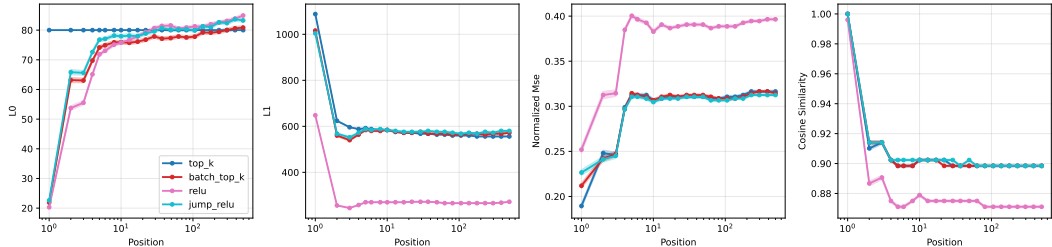

Figure 15: Reconstruction performance of Gemma-2-2B resid post Layer 12 (1/2 forward pass) SAEs on webtext.

# D CONSTANT SPARSITY ARGUMENT: RANK AND U-STATISTIC

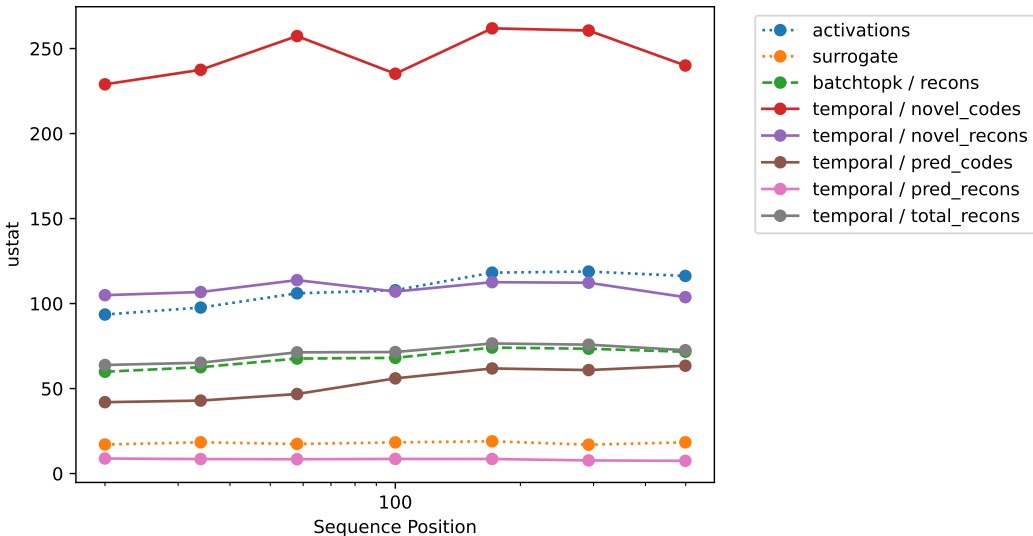

Figure 16: Gemma-2-2b: LLM activations vs Attention SAEs vs BatchTopK

## D.1 LLAMA-3.1-8B

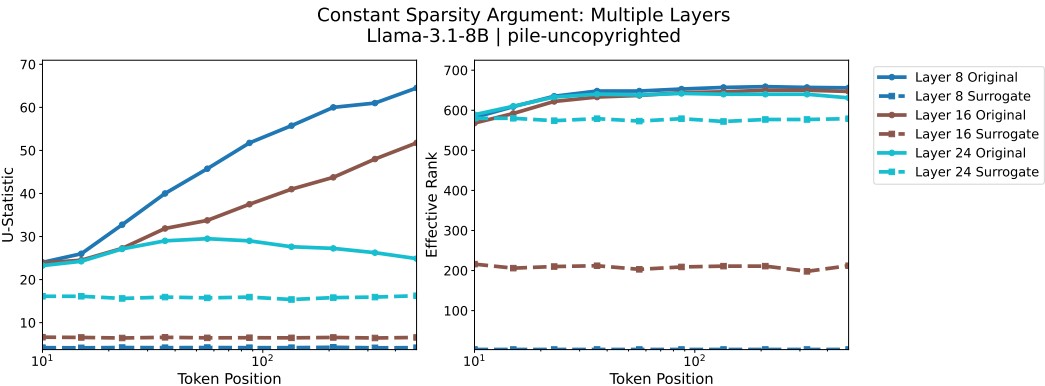

Figure 17: Rank and U-statistic, Llama-3.1-8B llm activations and surrogate, Multiple Layers

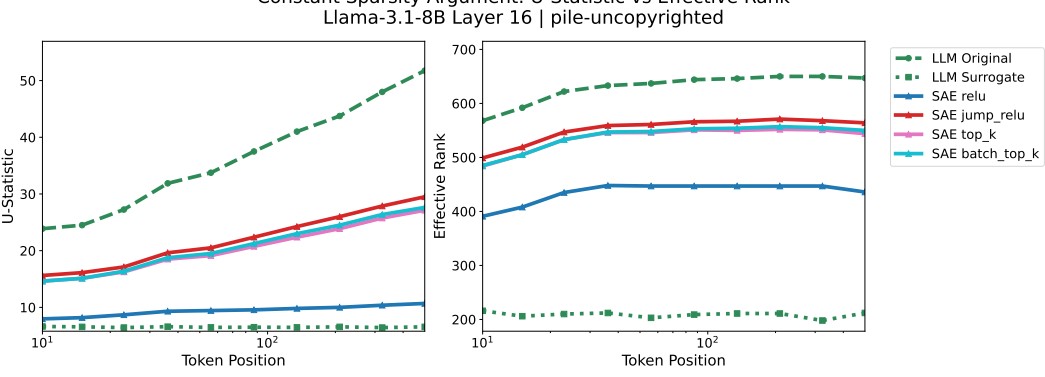

Figure 18: Rank vs U-statistic, Llama-3.1-8B, webtext

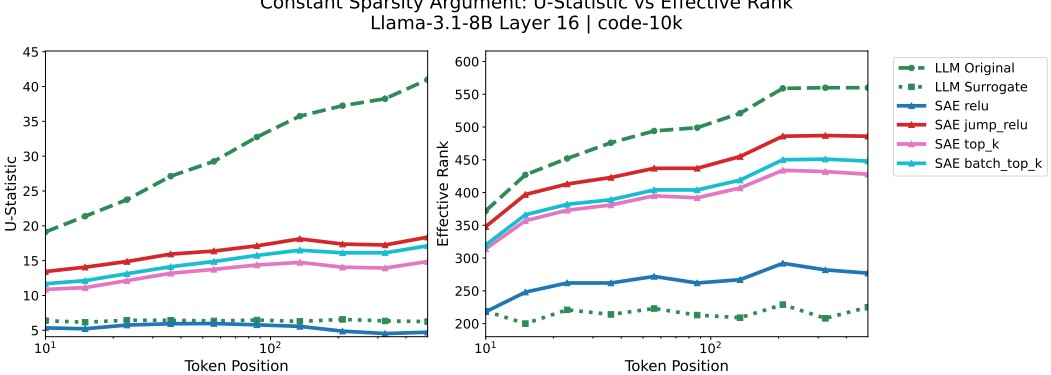

Figure 19: Rank vs U-statistic, llama-3.1-8B, simple stories

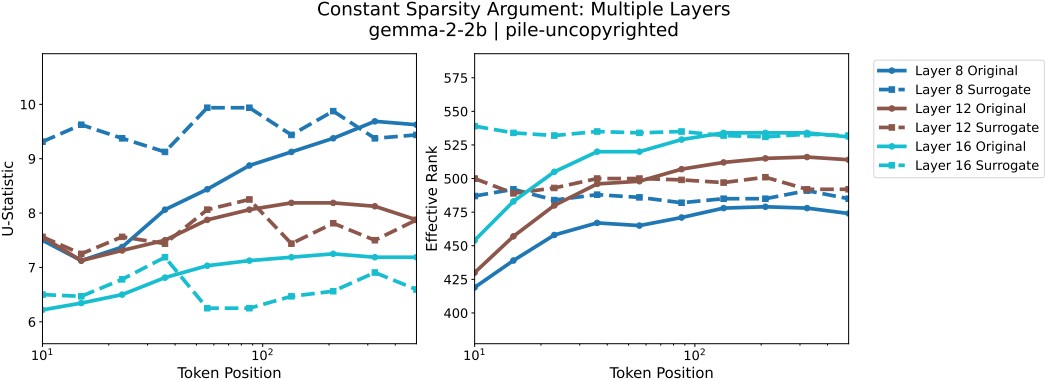

Figure 20: Rank vs U-statistic, llama-3.1-8B, Code

## D.2   GEMMA-2-2B

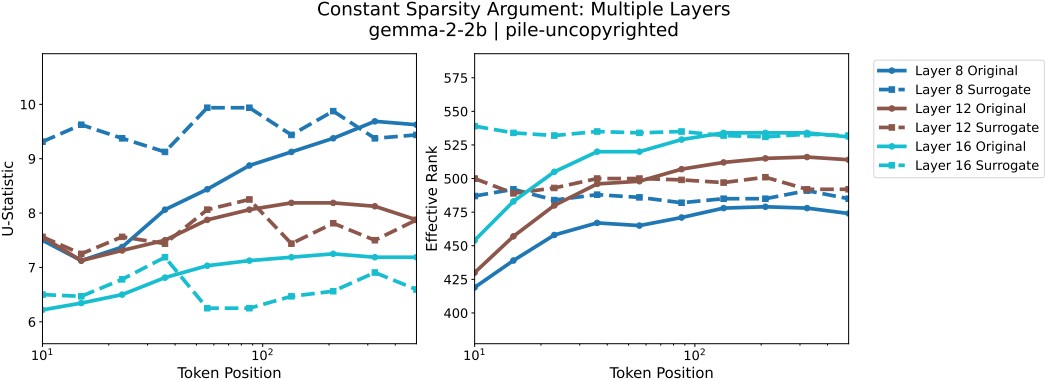

Figure 21: Rank and U-statistic, Gemma-2-2b llm activations and surrogate, Multiple Layers

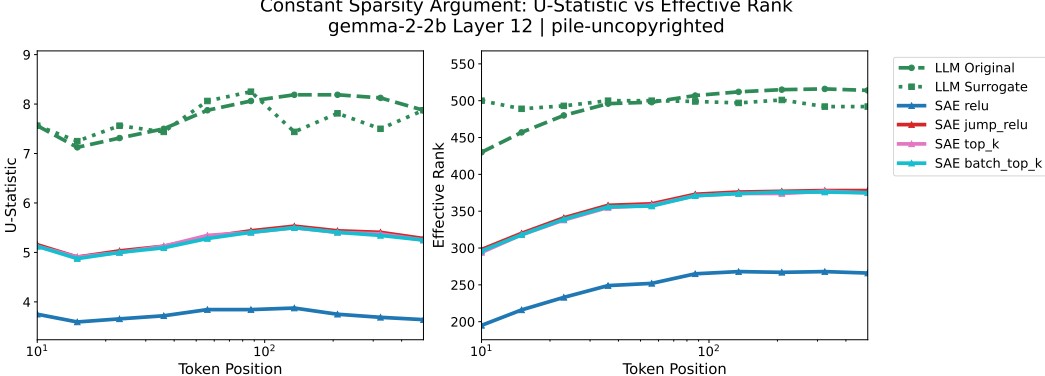

Figure 22: Rank vs U-statistic, Gemma-2-2b, webtext

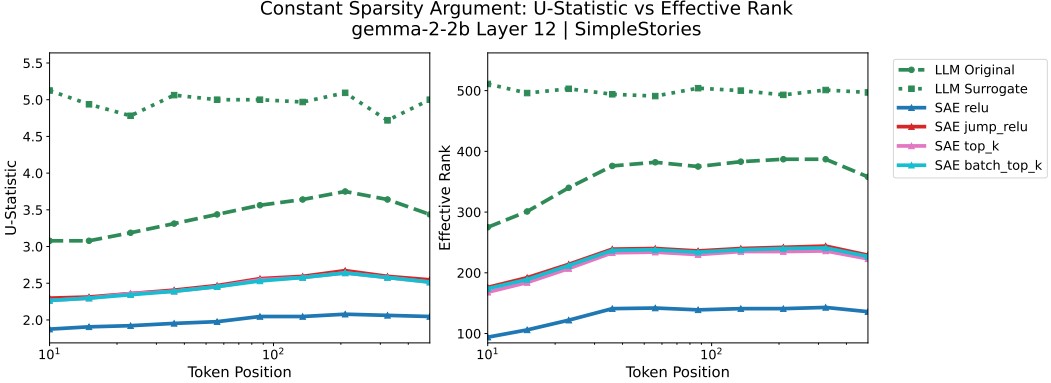

Figure 23: Rank vs U-statistic, Gemma-2-2b, simplestories

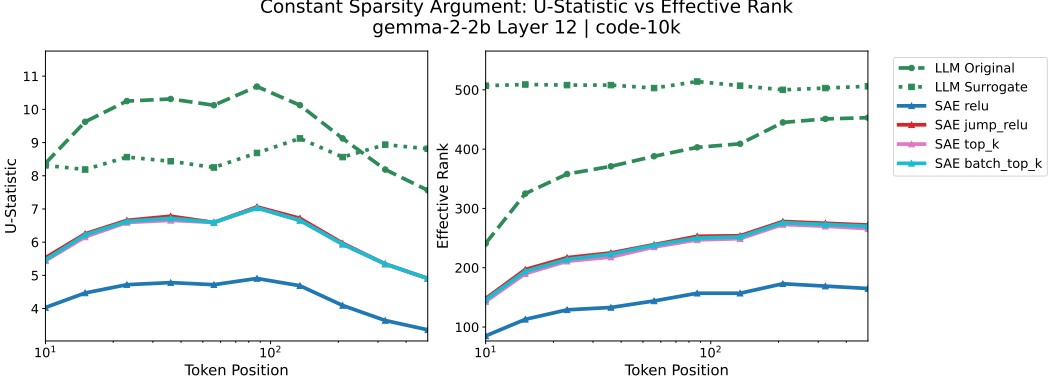

Figure 24: Rank vs U-statistic, Gemma-2-2b, code

# E    FURTHER INVESTIGATION OF TEMPORAL SAEs

In the results above, we showed across three domains that Temporal SAEs elicits intricate in-context geometry inline with our expectations based on experiments with LM activations in Sec. 3. Next, to make a case for following such more structured approaches for interpretability, we add to the results above to show that Temporal SAEs (i) offers a new way of interpreting temporally structured domains (e.g., user–model chats) via the context-dependent predictive component, and (ii) information that can be identified via SAEs continues to remain available within the novel component of Temporal SAEs.

## E.1    GEOMETRY OF LATENT CODES IN AN OOD DIALOGUE DOMAIN

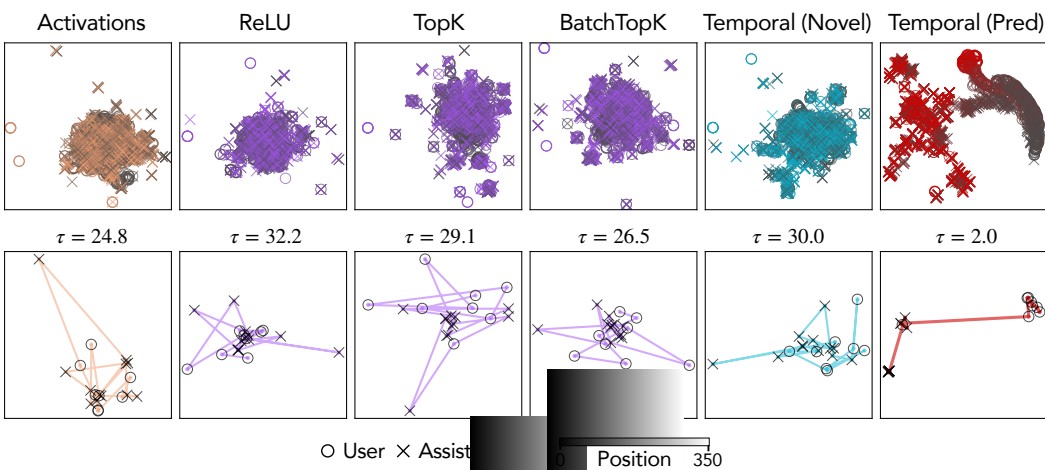

Figure 25: **Population and trajectory geometry in OOD dialogue.** UMAPs of raw activations and SAE codes on Ultrachat (first 350 tokens per conversation). **Top:** population embeddings colored by sequence position (gray→color) and marked by role (user=○, assistant=×); only the *predictive* code cleanly separates speaker roles while preserving a smooth temporal gradient. **Bottom:** a single conversation overlaid on the same UMAP; the predictive code follows a smooth near-geodesic with low tortuosity $\tau$, in contrast to jagged paths from standard SAEs/novel codes.

We analyze a multi-turn chat domain to test whether Temporal SAEs continues to reveal slow-moving structure. We emphasize the activations in this experiment are sampled from a Gemma-2-2B-IT model, although all analyzed interpretability protocols were trained on the base (non-instruction tuned) model. Thus, in a sense, the experiments in this section also gauge the ability of different protocols to generalize out-of-distribution (OOD)—a property SAEs have been argued to partially possess when transferring between base and the corresponding instruction-tuned models (Kissane et al., 2024). Specifically, we use the Ultrachat dataset, where we take the first 350 tokens from 1,000 conversations (mostly single-turn within this window, with up to four turns). For each token, we compute a shared 2D UMAP embedding from (i) raw model activations and (ii) latent codes from ReLU, TopK, and BatchTopK SAEs, as well as the novel and predictive components of the Temporal SAE. To visualize population structure clearly, we display only 100 conversations at a time to avoid clutter (Fig. 25), though fitting is applied to all 1,000.

At the population level, the predictive component of Temporal SAEs is the only representation that cleanly organizes the data along *two* interpretable axes: (1) a *speaker-role separation*, with user and assistant tokens forming distinct yet smoothly adjoining manifolds, and (2) a *temporal gradient*, with positions flowing continuously from early to late tokens along the manifold. By contrast, standard SAEs and the novel code show diffuse clouds with weaker role separability and no coherent positional gradient (Fig. 10, top row). To assess within-sequence geometry, we overlay a single sequence's trajectory onto the population embedding (Fig. 10, bottom row). Predictive codes trace a *smooth trajectory* that remains locally coherent within each role segment and transitions gracefully at speaker switches. In contrast, standard SAEs and the novel code produce jagged, zig–zagging paths. We again quantify this with *tortuosity* $\tau$ (Bullitt et al., 2003) (lower is straighter): predictive codes achieve $\tau \approx 2.0$, whereas SAEs and the novel code yield $\tau \approx 25$–$30$, indicating substantially more local

turning (Fig. 10, panel headers). This straightening mirrors the story-domain results (Sec. 6.1), now in an OOD conversational chat regime.

## E.2 Automatic Interpretability of Novel Codes

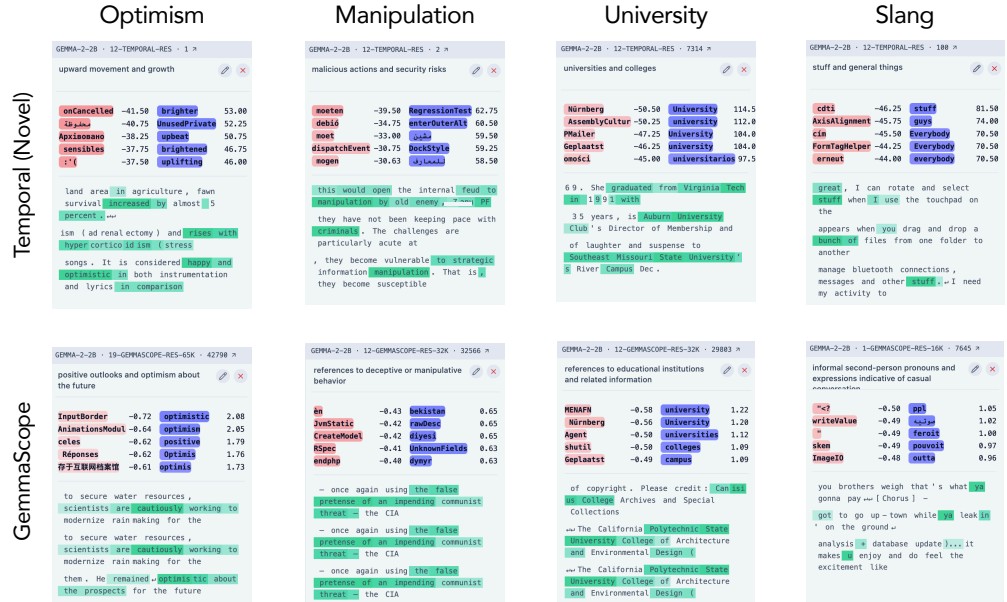

Figure 26: **Interpretability of novel codes.** We qualitatively compare the interpretability of novel codes extracted using Temporal SAEs with latent codes extracted using GemmaScope SAEs across four diverse categories. Example feature cards are shown across four categories, produced with the same activation-triggered examples plus direct logit attribution workflow used for standard SAEs (Lieberum et al., 2024).

A central benefit of SAEs is that their latents admit automatic interpretability pipelines ("autointerp") that attach human-readable semantics to features via activation-triggered examples, token/phrase saliency, and lightweight dashboards (e.g., GemmaScope / Neuronpedia). Our Temporal SAE adds a new, temporally aware predictive axis, but crucially *it does not sacrifice* these established interpretability affordances: the novel code remains compatible with standard SAE autointerp workflows. In other words, Temporal SAEs augment the toolbox rather than replacing it.

To support this point, we apply a conventional autointerp loop to the novel codes: (i) collect high-activation snippets and nearest-neighbor contexts for each latent; (ii) compute direct logit attribution to inspect which vocabulary items a latent tends to support or suppress; and (iii) render compact, browsable "feature cards" that aggregate examples and scores (as in GemmaScope / Neuronpedia). While we do not perform an exhaustive causal-intervention study here, nothing in the pipeline is specific to standard SAEs; the same probes and dashboards operate unchanged on the novel codes, and we leave a systematic intervention suite to future work. Qualitative inspection confirms that the novel codes recover the kinds of monosemantic, language-level features typically reported for SAEs. Figure 11 (p. 13) shows representative examples across diverse categories—Optimism, Manipulation, University, and Slang—where the novel latents consistently activate on intuitively relevant cue phrases and contexts, and where their attribution profiles align with the expected lexical sets.

Overall, when standard SAEs produce readable feature cards, the Temporal SAE's novel stream does as well. This complements our earlier results: the predictive stream captures slow, contextual structure (events, roles, long-range constraints), whereas the novel stream concentrates the fast, stimulus-driven information that matches inferences possible via existing SAEs, including autointerp pipelines.

# F    EMERGENT SEPARATION OF PREDICTIVE AND NOVEL PARTS

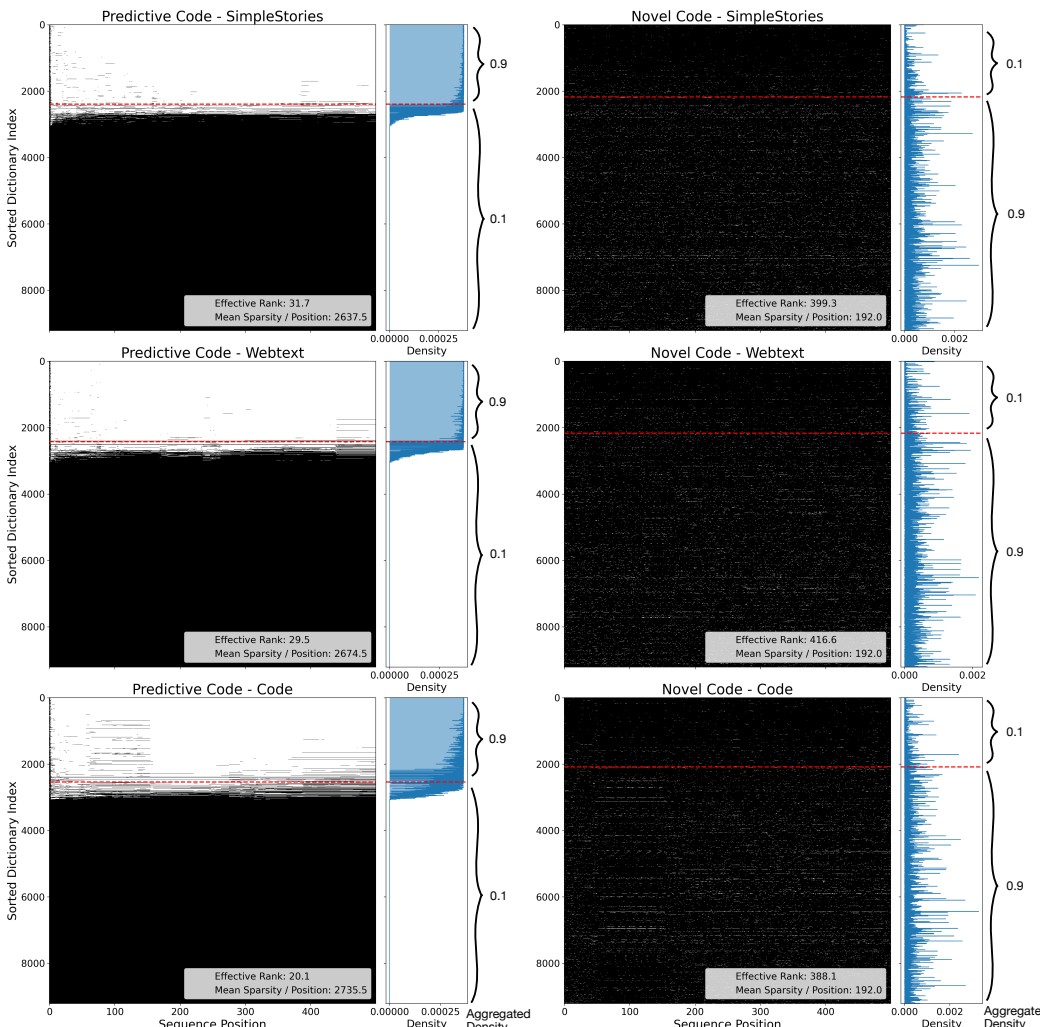

Figure 27: **Separation of Predictive and Novel Dictionaries.** Predictive and novel codes for sequences drawn from three datasets—SimpleStories (top), Webtext (middle), and Code (bottom)—are binarized to zero (black) and non-zero (white). The dictionary elements are sorted by the sum of non-zero activations in the predictive code across the sequence. The ordering obtained from the predictive code is also applied to the novel code. The red dashed line marks the separation of 90% of non-zero activation counts; only 10% are overlapping into the other subset. Additionally, the effective rank of the predictive codes is two orders of magnitudes lower than the average number of non-zero dictionary elements (Mean L0).

The Temporal SAEs studied in the main paper share a dictionary for both predictive and novel codes. We now investigate if there is any shared structure in the two dictionaries. In particular, given the predictive and novel codes play a different role, it is plausible the SAE learns to approximately split the dictionary into two parts: one responsible for computing the predictive code and the other for novel one. Specifically, we investigate a Temporal SAE trained on Gemma-2-2B Layer 12 residual stream activations. Results are shown in Fig. 27 and show a separation of dictionary elements for sequences drawn from three datasets: SimpleStories, Webtext, and Code. Specifically, we find ∼2K dictionary elements participate in defining the predictive code, while the remaining ∼8K primarily participate in defining the novel code. However, the absolute count is merely indicative of the L0 sparsity of a code, which may not reflect how many *directions* are actually used to define the code—estimating latter requires computing the rank of the code (in fact, we note that rank-sparsity is a well-known alternative to L0 sparsity in dictionary learning literature (Elad, 2010)). As we show

in Fig. 27, the effective rank of the predictive codes ($\sim$20–30) is substantially lower than both the effective rank ($\sim$390–410) and absolute L0 sparsity of novel codes.

Overall, the posthoc analysis above shows that predictive and novel codes largely use separate subsets of dictionary elements. This emergent disentanglement of the two components motivates one to preemptively split the dictionary into two components—one responsible for the predictive code and other for novel one. Our preliminary experiments show a Temporal SAE trained with this split-dictionary architecture is similarly performant as the tied dictionary one, resulting in predictive / novel codes with effective ranks $\sim$20–30 / $\sim$390–420, and a slightly better overall loss. It is possible optimizing hyperparameters (e.g., having different expansion factors for the two components) can make this architecture more performant than the tied dictionary one, but we leave a further characterization to future work.

## G  THE GEOMETRY OF STORIES: A NARRATIVE-DRIVEN DOMAIN

### G.1  HIERARCHICAL CLUSTERING OF CODES FROM STORY TOKENS

#### G.1.1  STORY 1

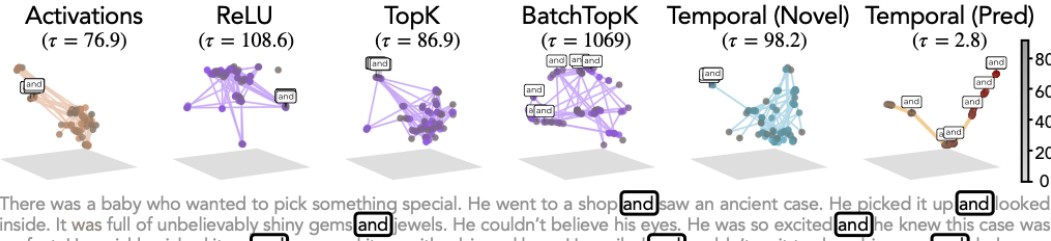

Figure 28: **Geometry.** Repeating results of Fig. 4, we again find smooth trajectories in UMAP projections for Temporal SAEs.

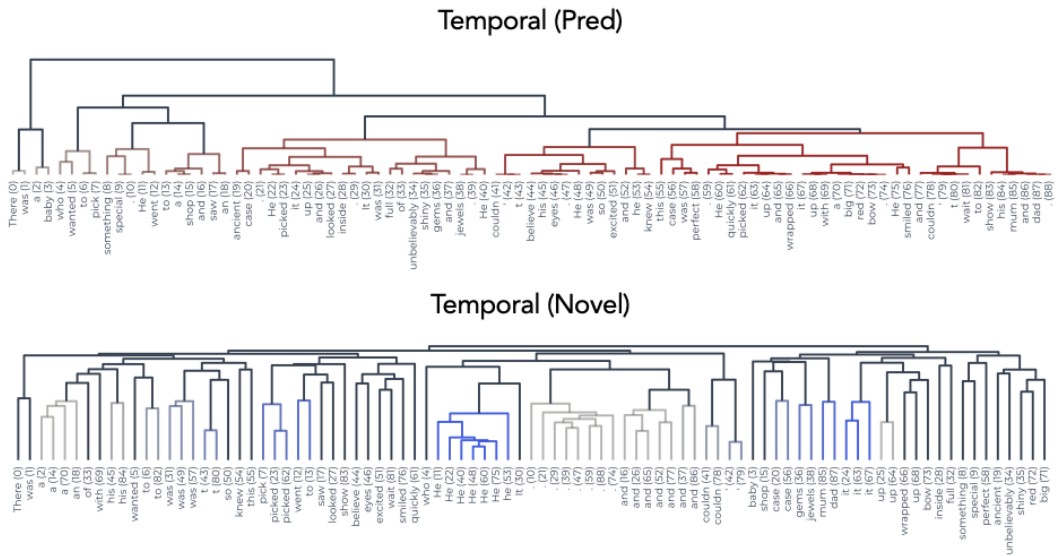

Figure 29: **Dendrograms of Predictive and Novel Components from Temporal SAEs.** Reproduction of the results from Fig. 4b, but including the novel component's dendrogram as well.

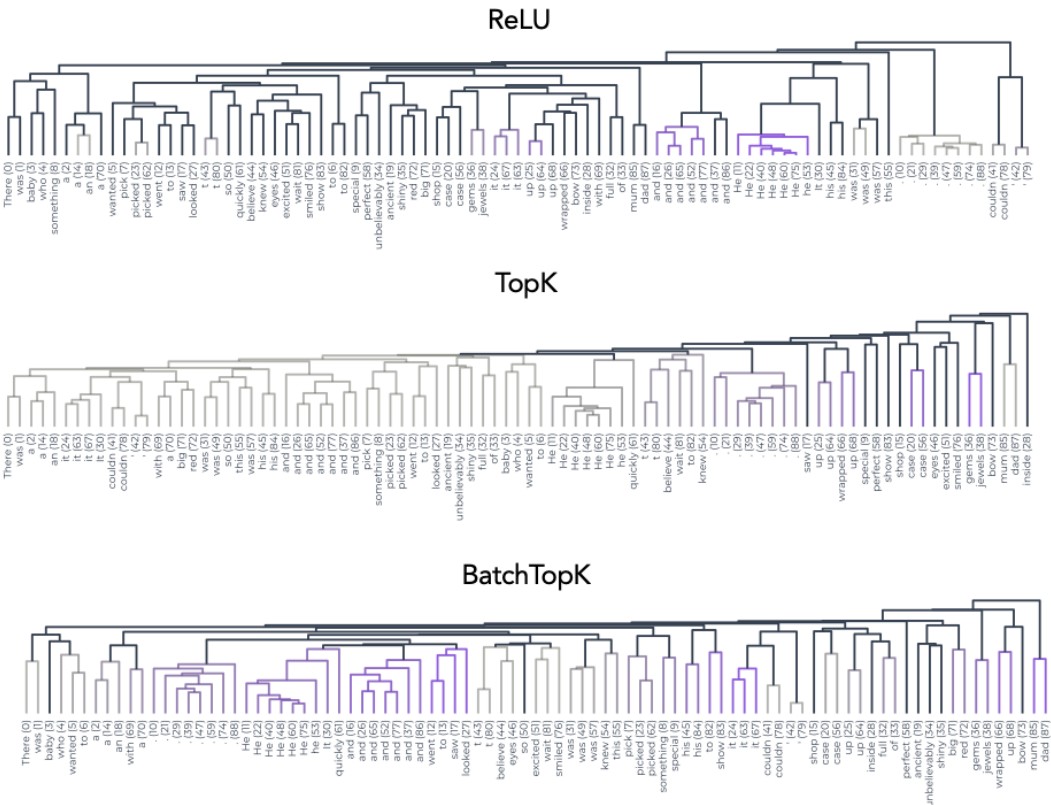

Figure 30: **Dendrograms of Standard SAEs' Latents Codes.** Reproduction of the results from Fig. 4b, but for latent codes extracted using standard SAEs.

## G.1.2 STORY 2

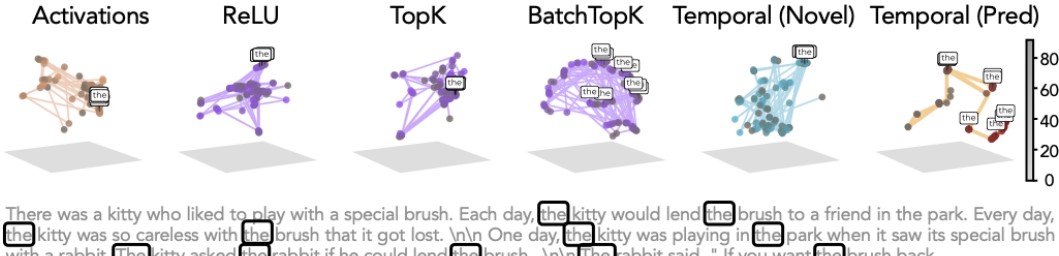

Figure 31: **Geometry.** Repeating results of Fig. 4 on a different story, we again find smooth trajectories in UMAP projections for Temporal SAEs.

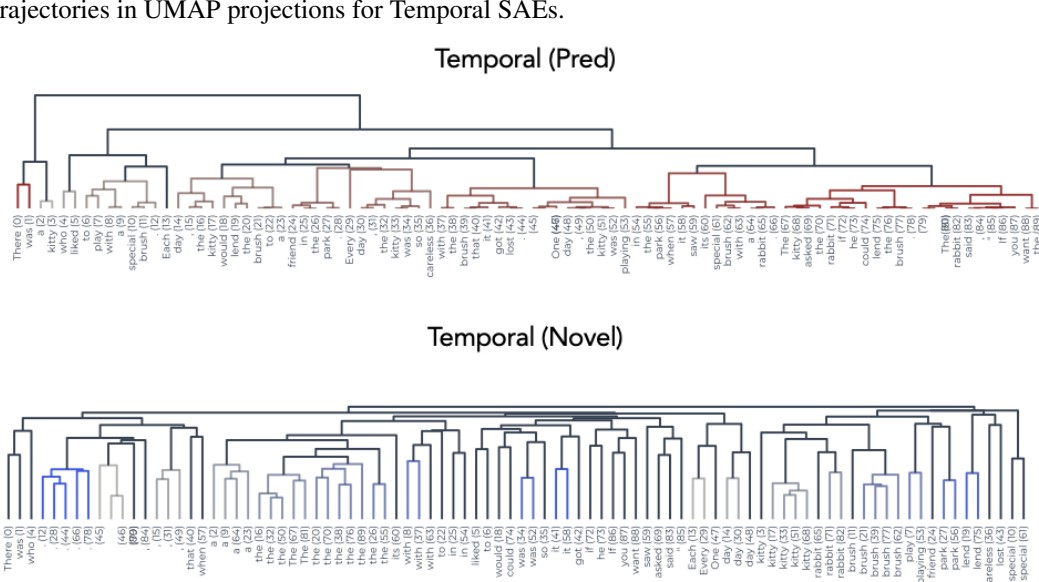

Figure 32: **Dendrograms of Predictive and Novel Components from Temporal SAEs.** Reproduction of the results from Fig. 4b, but on a different story and with both the predictive and novel component.

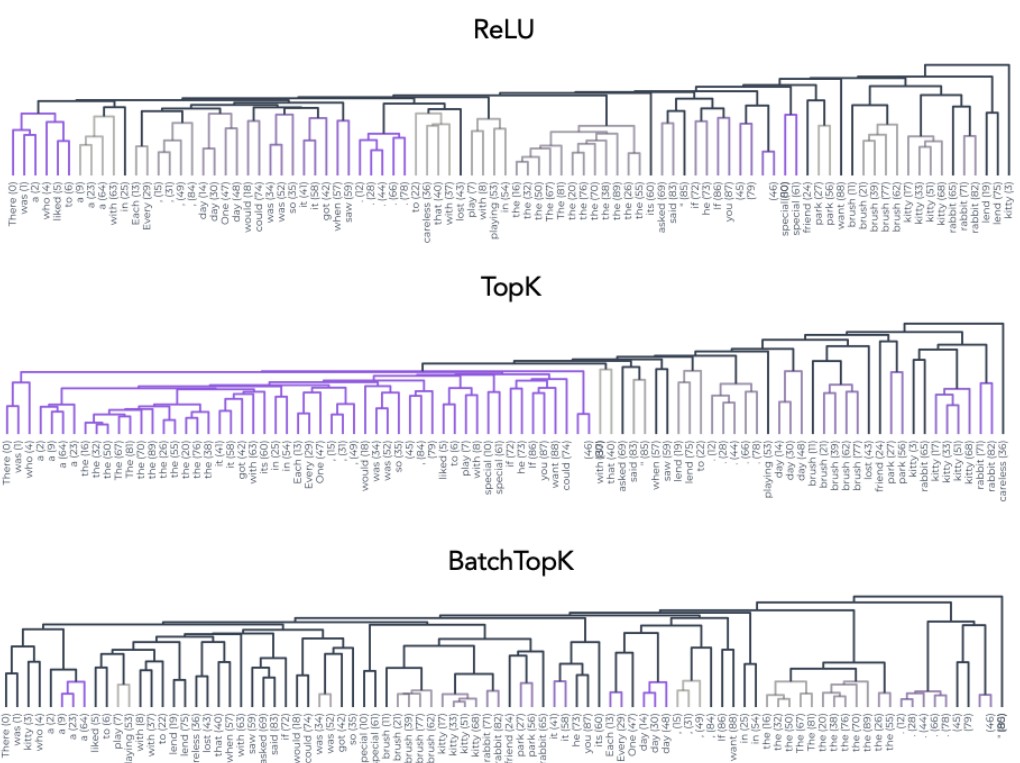

Figure 33: **Dendrograms of Standard SAEs' Latents Codes.** Reproduction of the results from Fig. 4b, but for latent codes extracted using standard SAEs on a different story.

### G.1.3 STORY 3

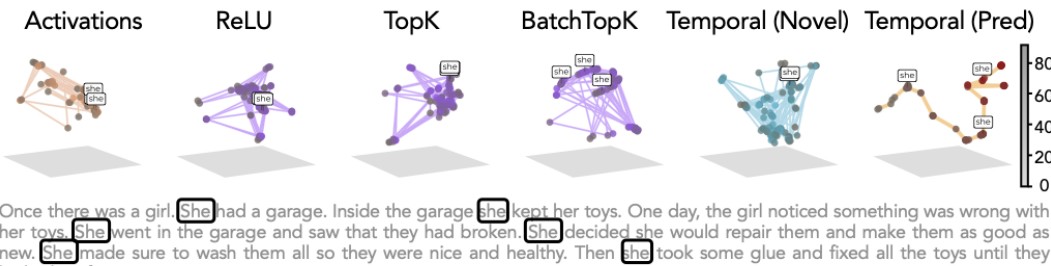

Figure 34: **Geometry.** Repeating results of Fig. 4 on a different story, we again find smooth trajectories in UMAP projections for Temporal SAEs.

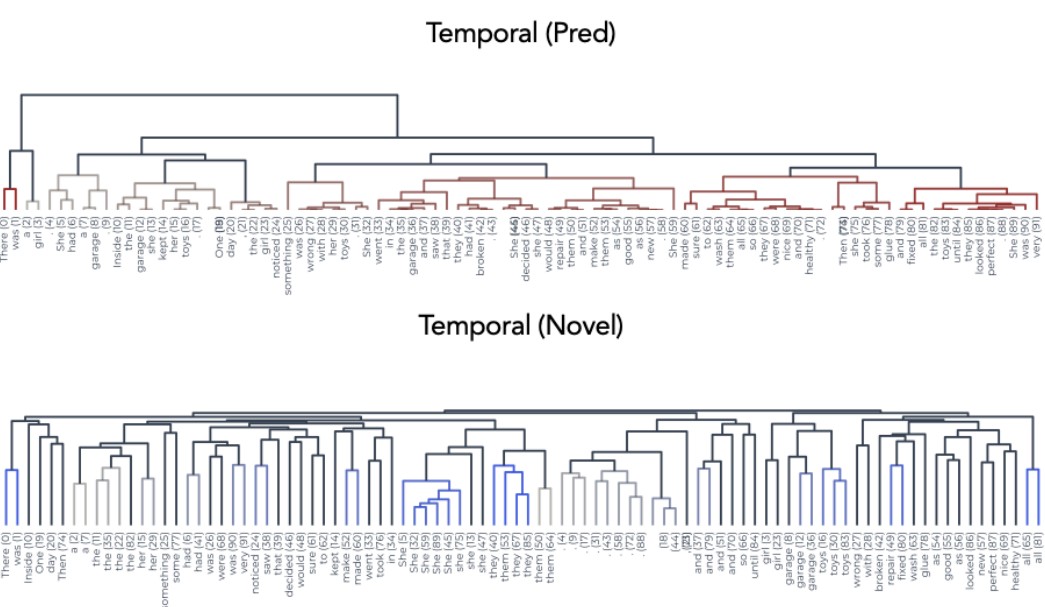

Figure 35: **Dendrograms of Predictive and Novel Components from Temporal SAEs.** Reproduction of the results from Fig. 4b, but on a different story and with both the predictive and novel component.

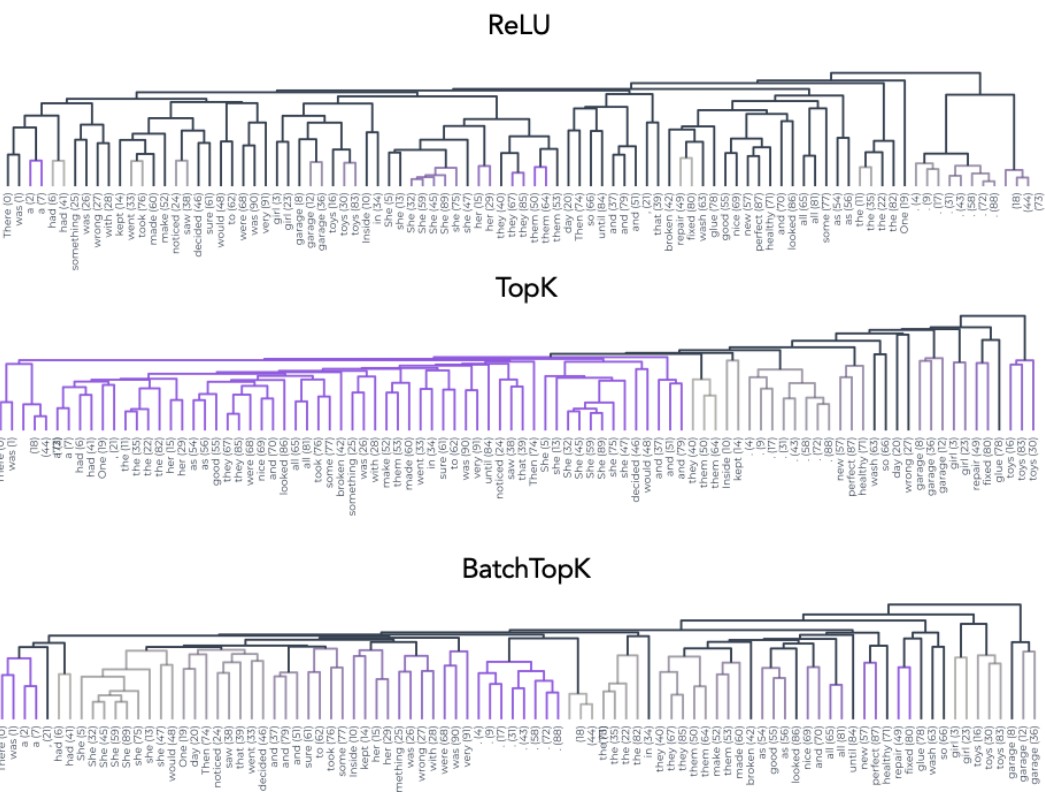

Figure 36: **Dendrograms of Standard SAEs' Latents Codes.** Reproduction of the results from Fig. 4b, but for latent codes extracted using standard SAEs on a different story.

### G.2 CODE: ANALYZING ANOTHER DOMAIN

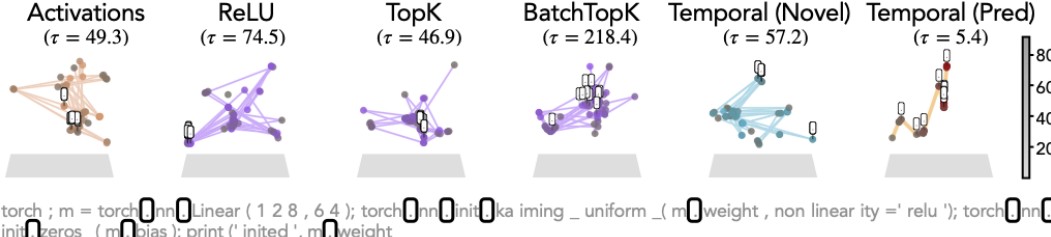

Figure 37: **Code.** Reproducing the results of Fig. 4, but on a different domain, i.e., code. We again see a temporally disentangled, smoothly running trajectory for latent codes extracted using the predictive component of Temporal SAEs.

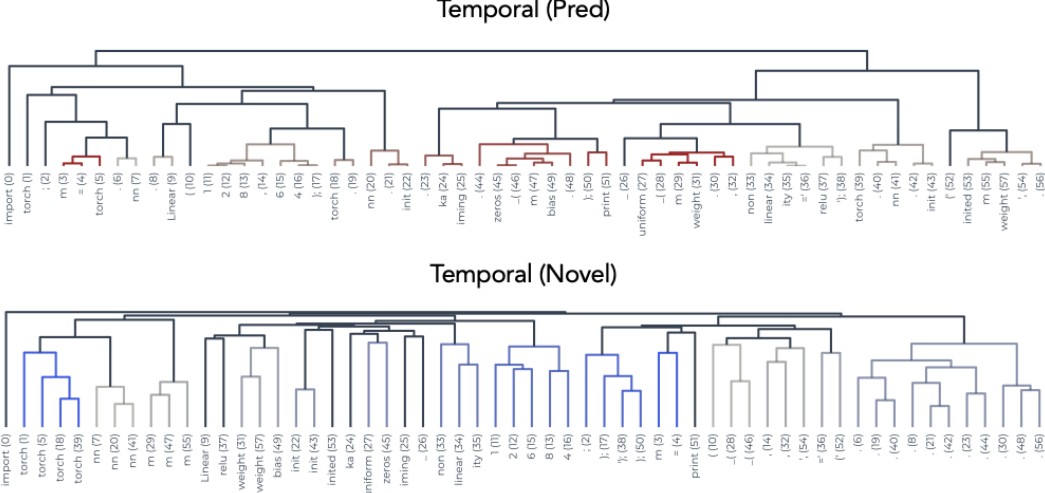

Figure 38: **Dendrograms of Predictive and Novel Components from Temporal SAEs.** Reproduction of the results from Fig. 4b, but on a different domain shows similar results as with narrative-driven text (stories) for both the predictive and novel component.

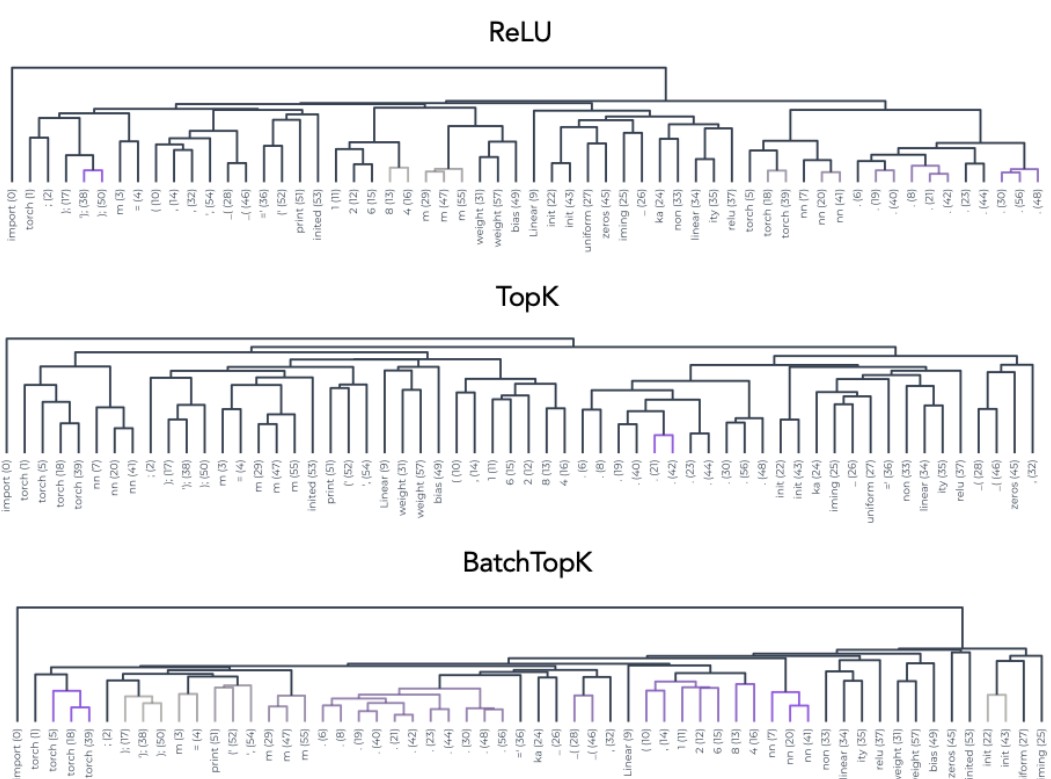

Figure 39: **Dendrograms of Standard SAE Latent Codes.** Reproduction of the results from Fig. 4b, but on a different domain shows similar results as with narrative-driven text (stories) for standard SAEs.

## G.3 FURTHER RESULTS: SIMILARITY MAPS ON STORIES

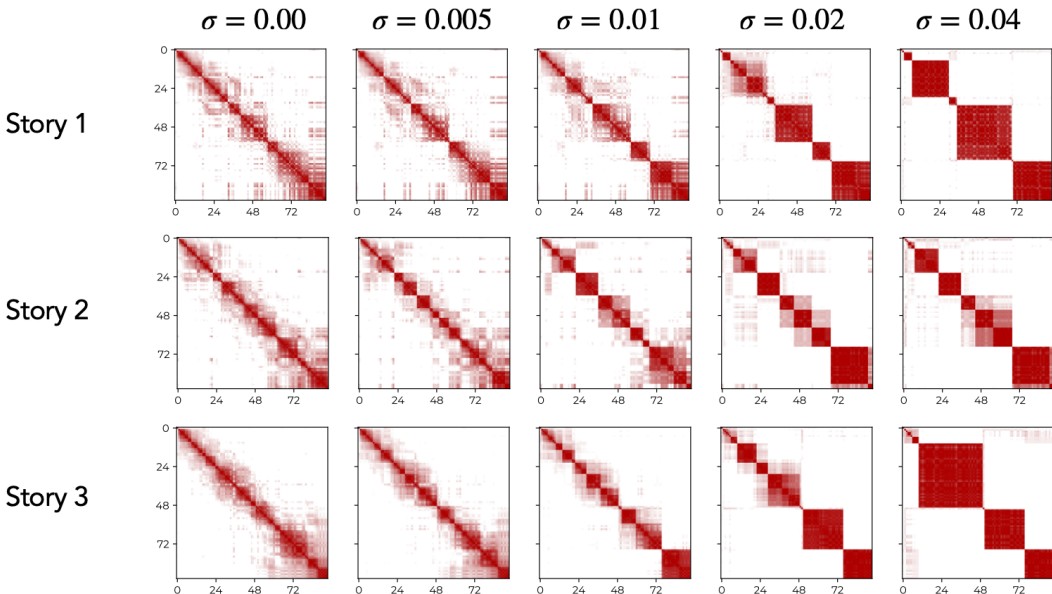

Figure 40: **Temporal (Pred) Similarity Maps Elicit Multi-Scale Structure with Noise.** Repeating the analysis shown in Fig. 6, we find the coarsening of temporal blocks is a robust result that continues to hold for different inputs.

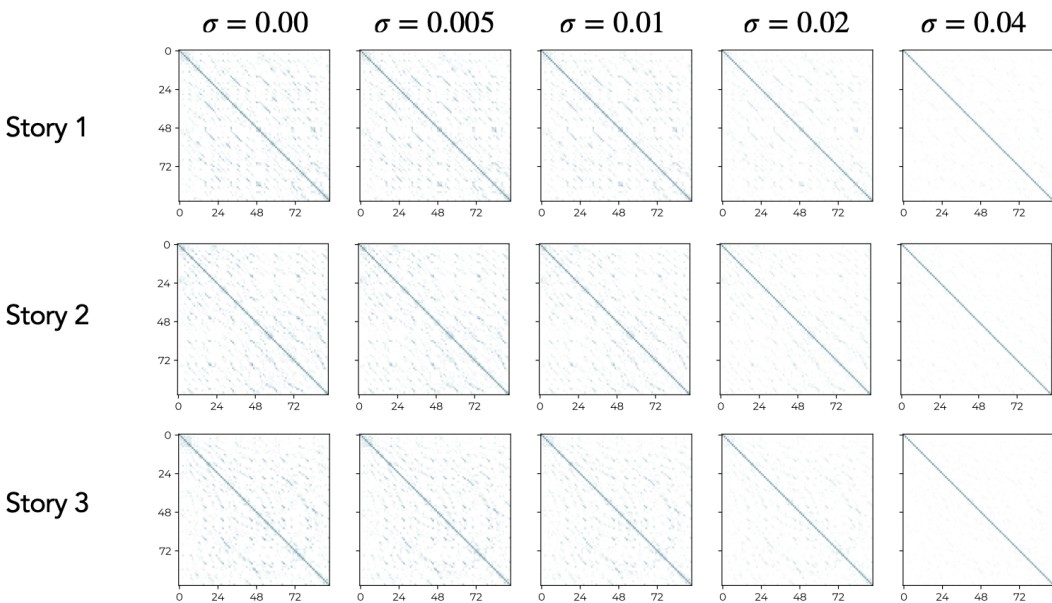

Figure 41: **Temporal (Novel) Similarity Maps under Noise.** Repeating the analysis shown in Fig. 6, we find the novel component is only able to capture minimal local similarities, which when analyzed via dendrograms, show clustering based on lexical information.

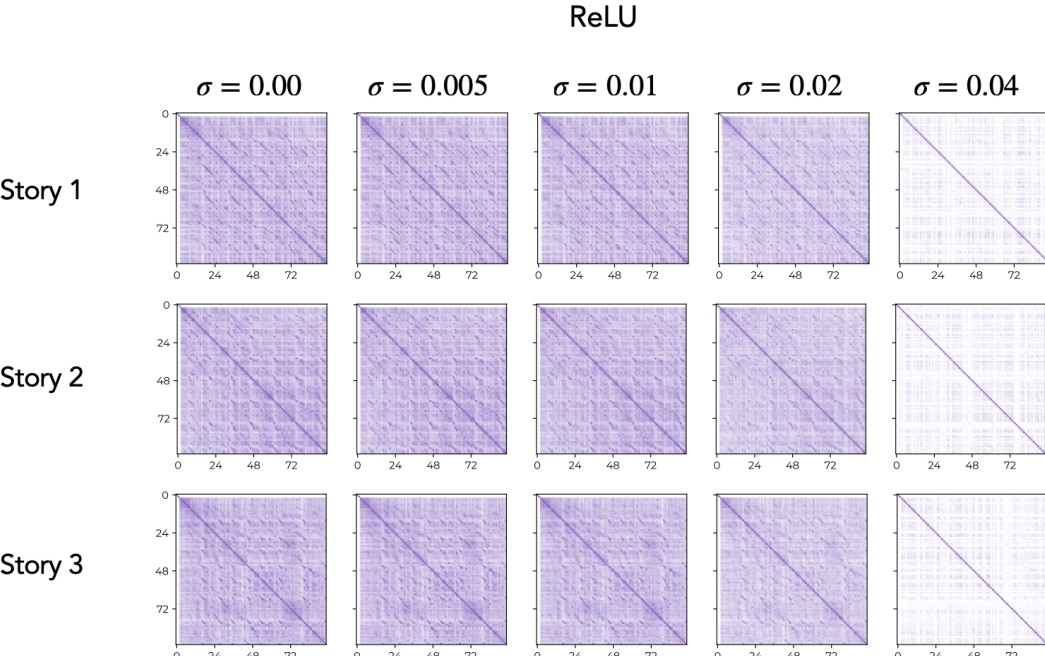

Figure 42: **ReLU Codes' Similarity Maps under Noise.** Repeating the analysis shown in Fig. 6 on ReLU SAEs, we find the ReLU latent code has high similarity across the board, suggesting lack of meaningful temporal information. This similarity is entirely removed when noise scale increases too much, which, as shown in Fig. 6c, corresponds to the point that variance explained by ReLU SAEs drops to $\sim$0.

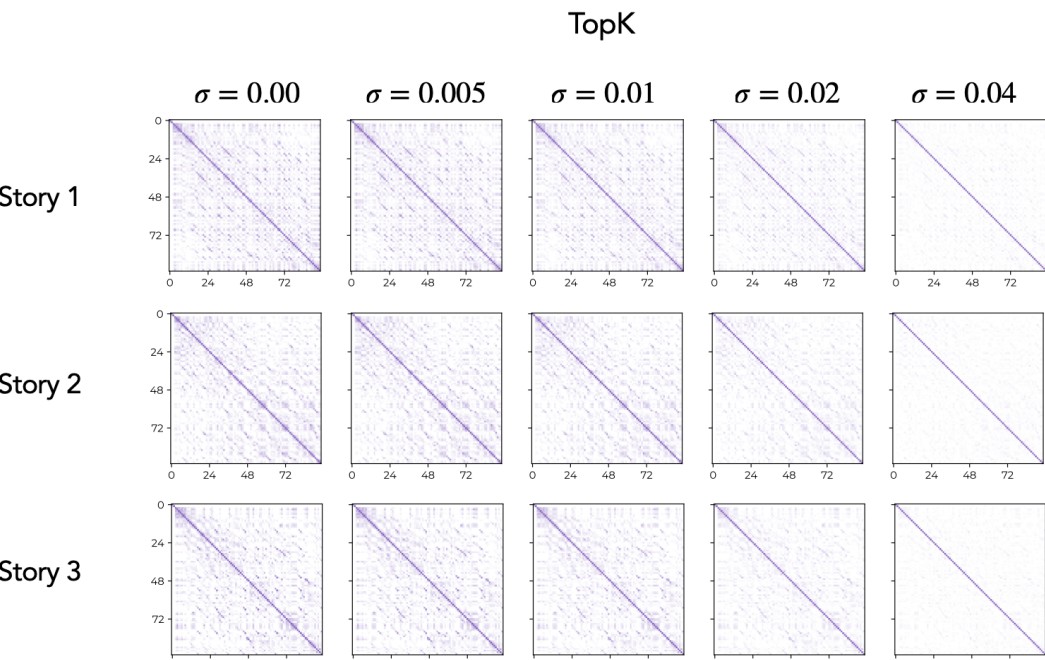

Figure 43: **TopK Codes' Similarity Maps under Noise.** Repeating the analysis shown in Fig. 6, we find TopK SAE's latent codes are only able to capture minimal local similarities, which when analyzed via dendrograms, show clustering based on lexical information. Akin to ReLU SAEs, we see this similarity map approximately turns into an identity matrix when the noise scale increases too much, which, as shown in Fig. 6c, corresponds to the point that variance explained by TopK SAEs drops to ~0.

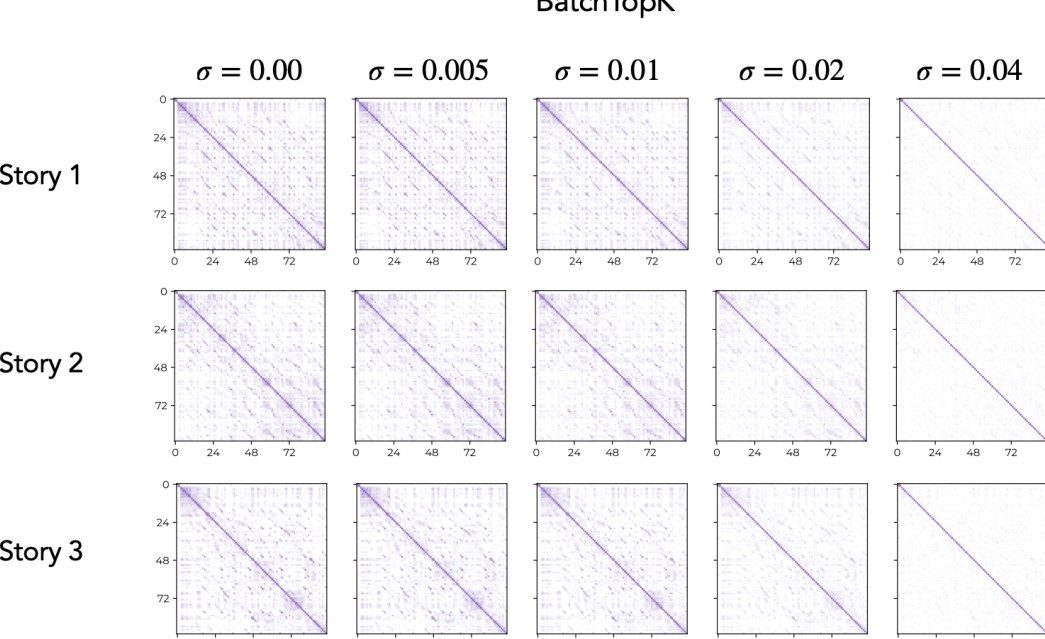

Figure 44: **BatchTopK Codes' Similarity Maps under Noise.** Repeating the analysis shown in Fig. 6, we find BatchTopK SAE's latent codes are only able to capture minimal local similarities, which when analyzed via dendrograms, show clustering based on lexical information. Akin to ReLU SAEs, we see this similarity map approximately turns into an identity matrix when the noise scale increases too much, which, as shown in Fig. 6c, corresponds to the point that variance explained by BatchTopK SAEs drops to ∼0.

## G.4 FURTHER RESULTS: DENDROGRAMS AND EVALUATIONS ON GARDEN PATH SENTENCES

### G.4.1 DENDROGRAMS

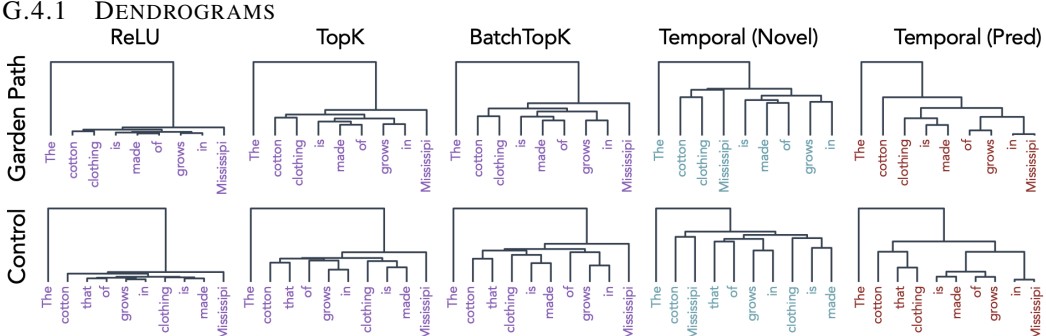

Figure 45: **Example Sentence 1.** Repeating the results of Fig. 7a on a different garden path sentence.

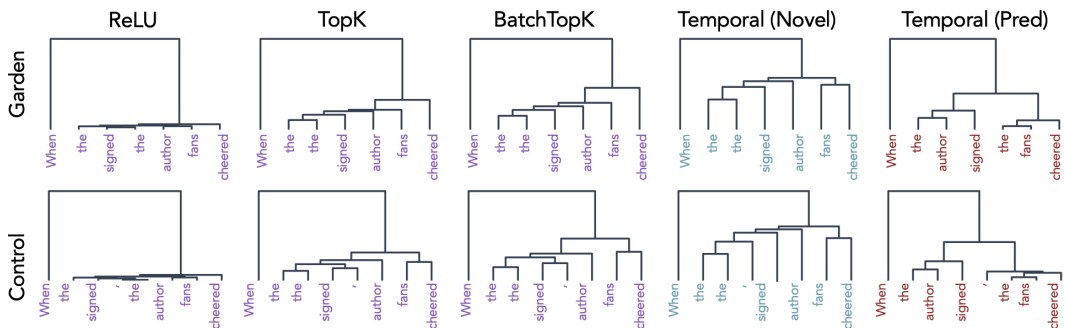

Figure 46: **Example Sentence 2.** Repeating the results of Fig. 7a on a different garden path sentence.

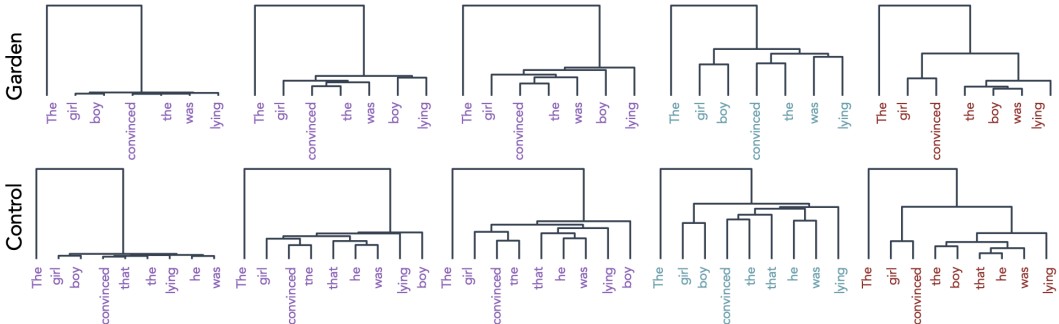

Figure 47: **Example Sentence 3.** Repeating the results of Fig. 7a on a different garden path sentence.

## H    FURTHER RESULTS: TEMPORAL SAEs ON LLAMA-3.1-8B

We replicate a subset of the experiments evaluating Temporal SAEs on Llama-3.1-8B. The training protocol remains the same as that of Gemma-2-2B model: train on 1B token activations with similar normalization schema, but activations are harvested now from Layer 15 (i.e., at ∼50% of model depth). We note that the trained SAEs are not finished training, and hence the following results are solely meant to be an impression of whether the qualitative trends observed with Gemma models generalize to another model class.

### H.1    GEOMETRY, DENDROGRAMS, AND SPECTRA

#### H.1.1    STORY 1

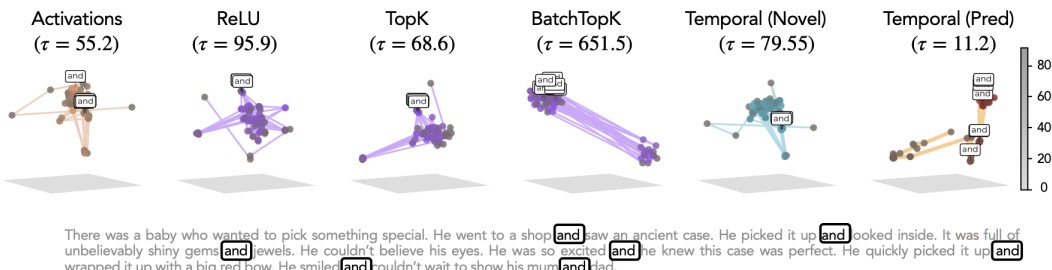

Figure 48: **Geometry.** Repeating results of Fig. 4, we again find smooth trajectories in UMAP projections for Temporal SAEs.

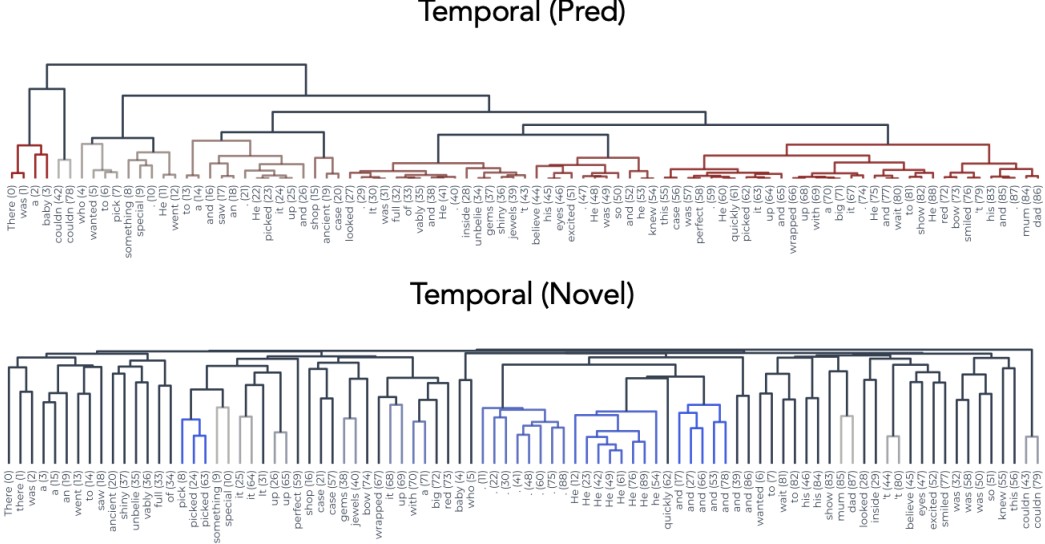

Figure 49: **Dendrograms of Predictive and Novel Components from Temporal SAEs.** Reproduction of the results from Fig. 4b, but including the novel component's dendrogram as well.

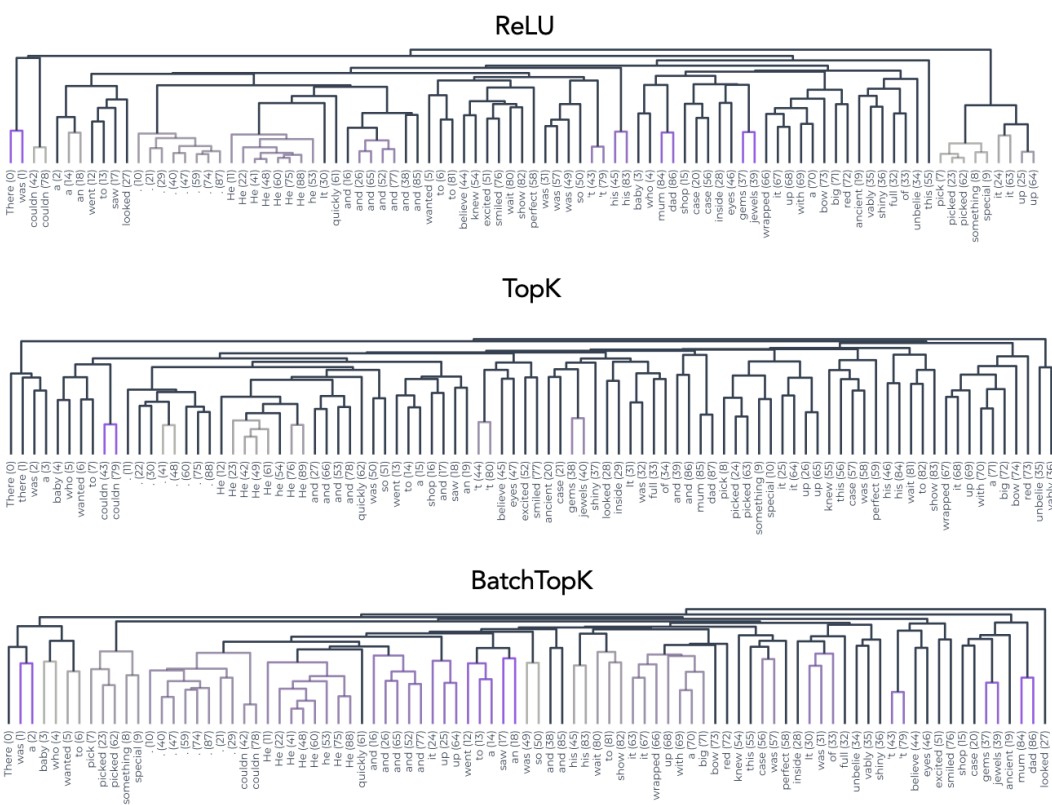

Figure 50: **Dendrograms of Standard SAEs' Latents Codes.** Reproduction of the results from Fig. 4b, but for latent codes extracted using standard SAEs.

### H.1.2 STORY 2

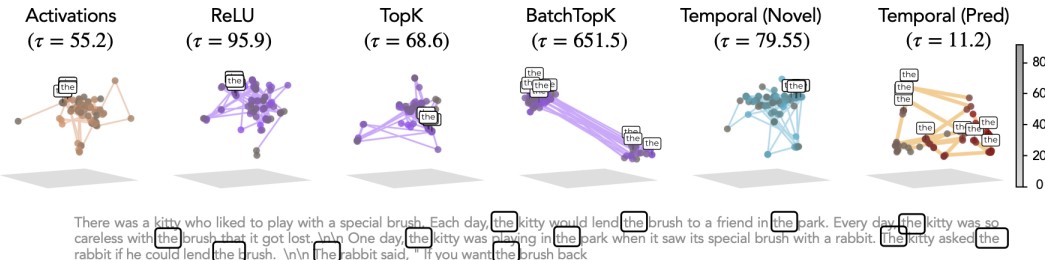

Figure 51: **Geometry.** Repeating results of Fig. 4, we again find smooth trajectories in UMAP projections for Temporal SAEs.

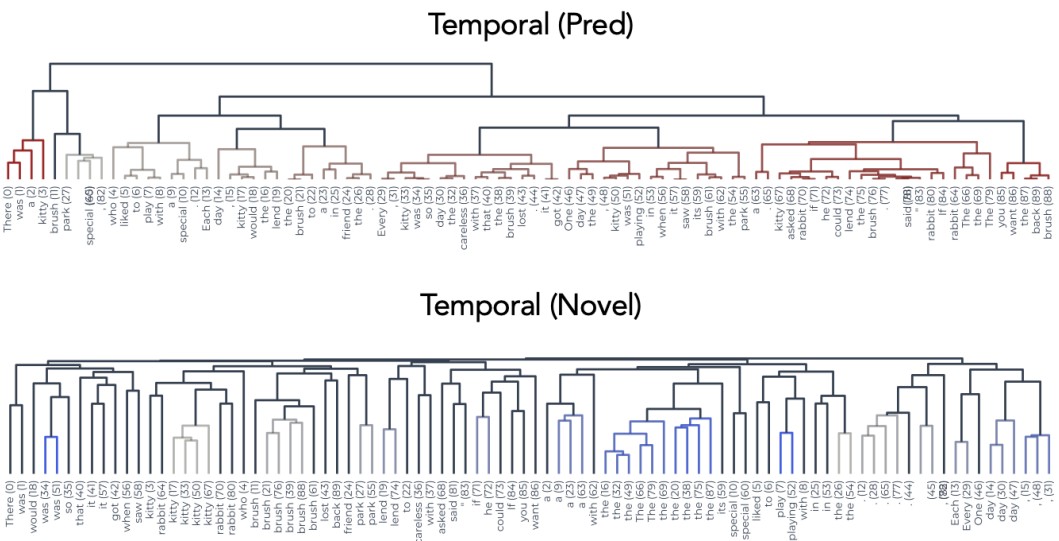

Figure 52: **Dendrograms of Predictive and Novel Components from Temporal SAEs.** Reproduction of the results from Fig. 4b, but including the novel component's dendrogram as well.

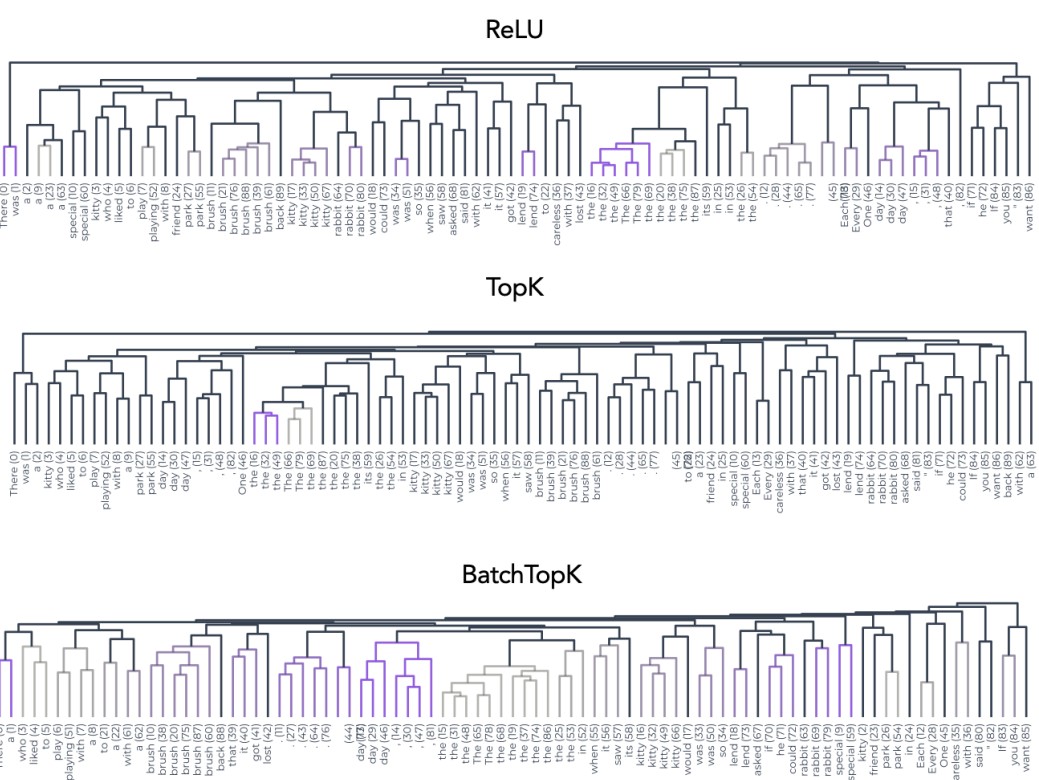

Figure 53: **Dendrograms of Standard SAEs' Latents Codes.** Reproduction of the results from Fig. 4b, but for latent codes extracted using standard SAEs.

### H.1.3 STORY 3

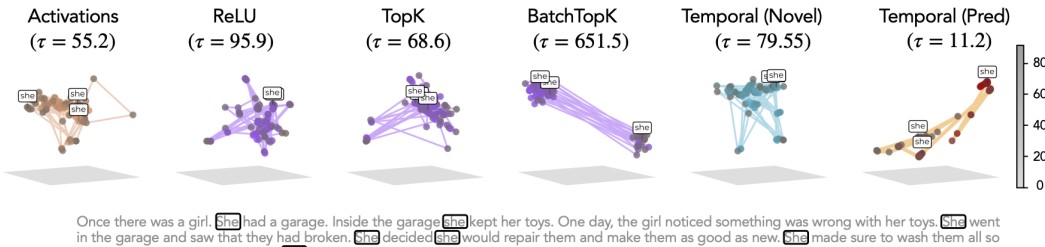

Once there was a girl. She had a garage. Inside the garage she kept her toys. One day, the girl noticed something was wrong with her toys. She went in the garage and saw that they had broken. She decided she would repair them and make them as good as new. She made sure to wash them all so they were nice and healthy. Then she took some glue and fixed all the toys until they looked perfect.

Figure 54: **Geometry.** Repeating results of Fig. 4, we again find smooth trajectories in UMAP projections for Temporal SAEs.

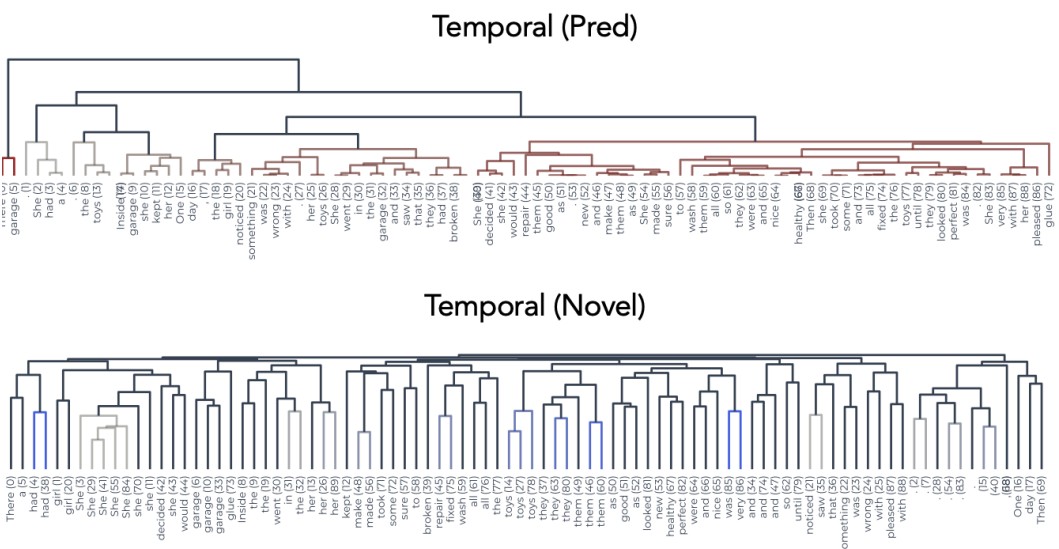

Figure 55: **Dendrograms of Predictive and Novel Components from Temporal SAEs.** Reproduction of the results from Fig. 4b, but including the novel component's dendrogram as well.

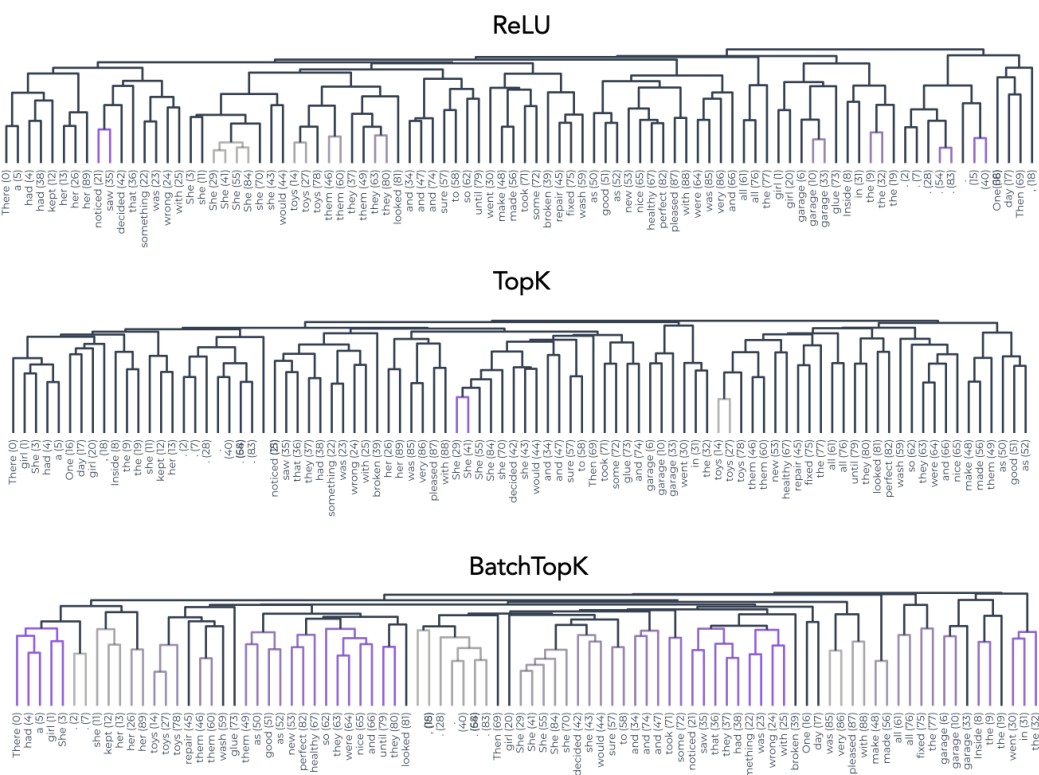

Figure 56: **Dendrograms of Standard SAEs' Latents Codes.** Reproduction of the results from Fig. 4b, but for latent codes extracted using standard SAEs.

### H.1.4 COMPARING EIGENSPECTRUM WITH SLOW VS. FAST FEATURES

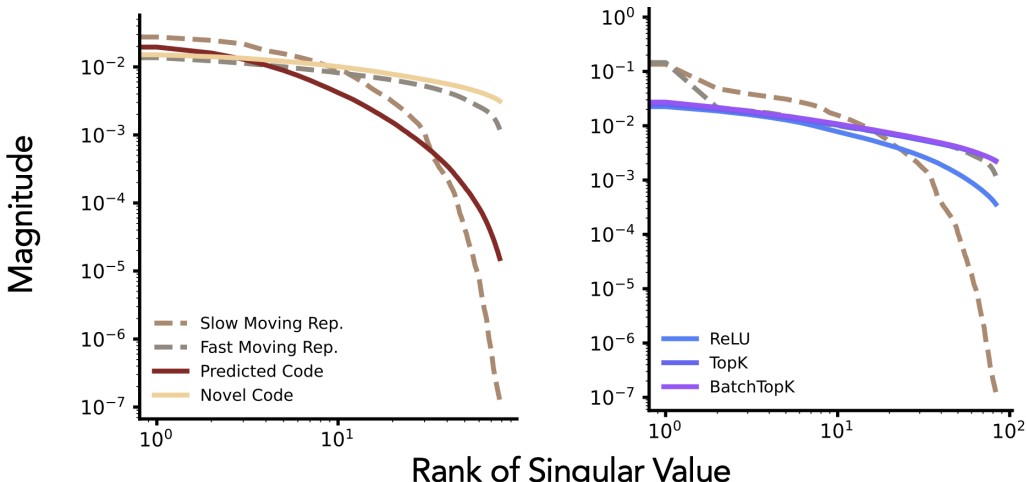

Figure 57: **Kernel spectrum for latent codes and model representations**. Kernels defined using novel code from Temporal SAEs and standard SAEs both align well with the fast-changing part of model representations; meanwhile, only the predictive code shows strong similarity to the slow changing part.

## H.2 EVENT BOUNDARIES AND NOISE STABILITY

### H.2.1 EVENT BOUNDARIES

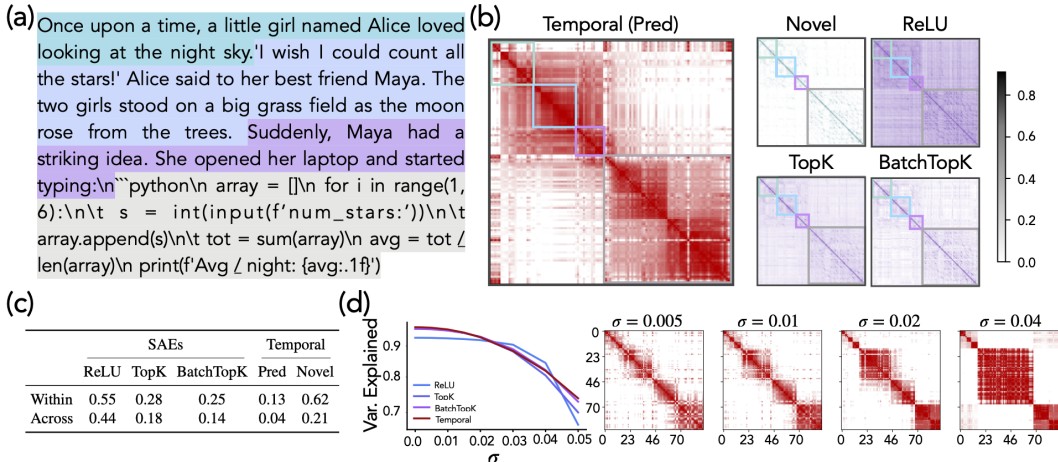

Figure 58: **Analysis of Event Boundaries.** Reproducing results from Fig. 6 on Llama-3.1-8B, we see similar behavior as the Gemma analysis done previously.

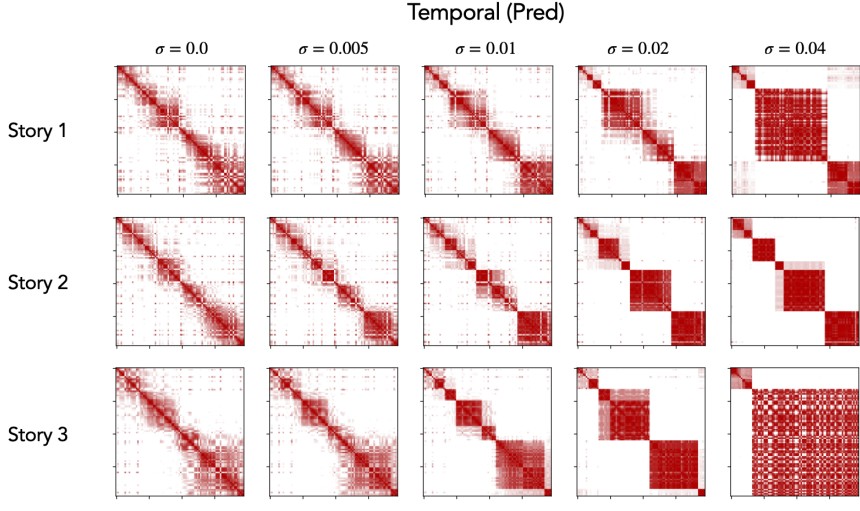

Figure 59: **Temporal (Pred) Similarity Maps Elicit Multi-Scale Structure with Noise.** Repeating the analysis shown in Fig. 6, we find the coarsening of temporal blocks is a robust result that continues to hold for different inputs.

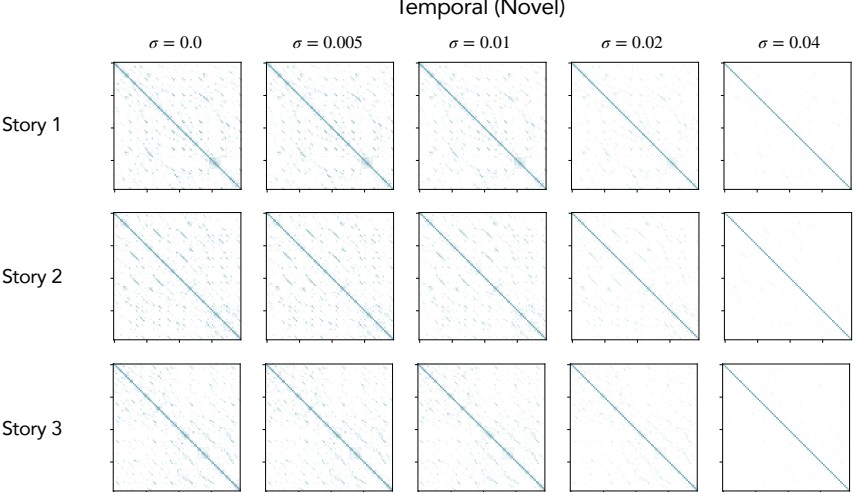

Figure 60: **Temporal (Novel) Similarity Maps under Noise.** Repeating the analysis shown in Fig. 6, we find the novel component is only able to capture minimal local similarities, which when analyzed via dendrograms, show clustering based on lexical information.

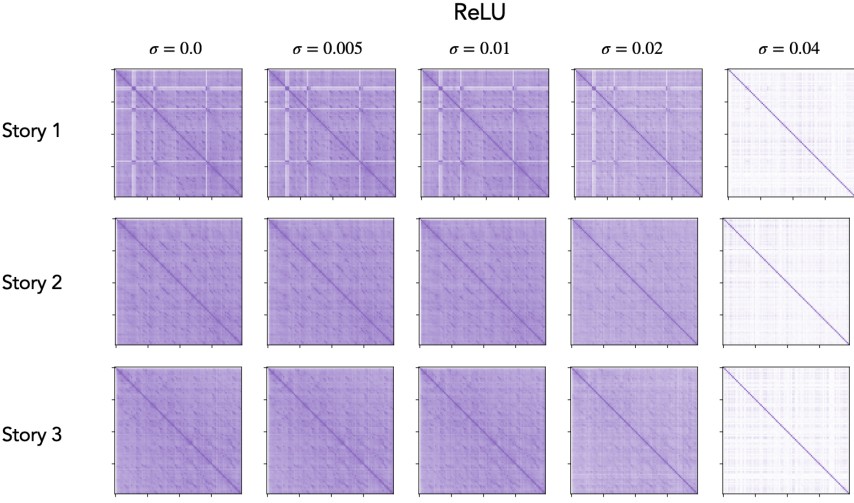

Figure 61: **ReLU Codes' Similarity Maps under Noise.** Repeating the analysis shown in Fig. 6 on ReLU SAEs, we find the ReLU latent code has high similarity across the board, suggesting lack of meaningful temporal information.

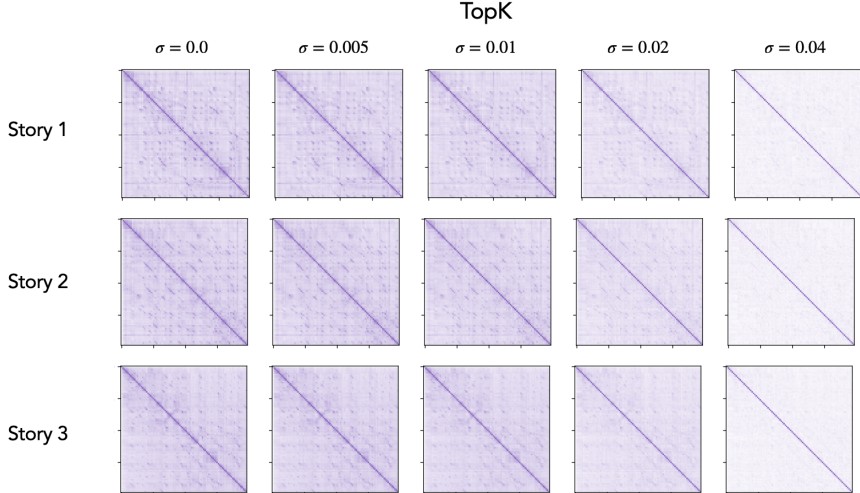

Figure 62: **TopK Codes' Similarity Maps under Noise.** Repeating the analysis shown in Fig. 6, we find TopK SAE's latent codes are only able to capture minimal local similarities, which when analyzed via dendrograms, show clustering based on lexical information.

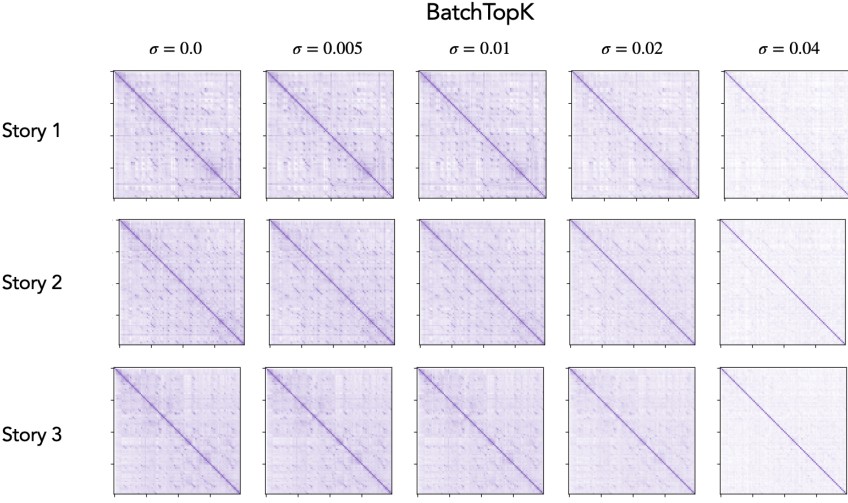

Figure 63: **BatchTopK Codes' Similarity Maps under Noise.** Repeating the analysis shown in Fig. 6, we find BatchTopK SAE's latent codes are only able to capture minimal local similarities, which when analyzed via dendrograms, show clustering based on lexical information.

# I    FURTHER THEORY RESULTS

## I.1    PRIORS ON THE SPARSE CODE FOR VARIOUS SAEs

We restate and prove the proposition on independence priors of SAEs over time (Proposition 4.1) below.

**Proposition I.1** (Independence priors over time). *Consider the SAE maximum aposteriori (MAP) objective for ReLU, JumpReLU, TopK and BatchTopK SAEs. The sparsity constraints for these SAEs are additive over time, resulting in:*

$$
\arg\min_{\boldsymbol{D},\boldsymbol{z}} \frac{1}{T}\sum_{i=1}^{T} \|\boldsymbol{x}_i - \boldsymbol{D}\boldsymbol{z}_i\|_2^2 + \lambda\mathcal{R}(\boldsymbol{z}_i),
$$

$$
\text{s.t. } \boldsymbol{z}_k = f_{\text{SAE}}(\boldsymbol{x}_k)\ \forall k,\ \tilde{g}(\boldsymbol{z}_1,\ldots,\boldsymbol{z}_T) = \sum_i \tilde{g}(\boldsymbol{z}_i) = 0.
$$

(8)

*This MAP objective has an independent and identically distributed (i.i.d.) prior over time i.e.,*

$$
P(\boldsymbol{z}_1,\ldots,\boldsymbol{z}_T) \propto \prod_{t=1}^{T} \exp\left(-\lambda\mathcal{R}(\boldsymbol{z}_i) - \tilde{\lambda}\tilde{g}(\boldsymbol{z}_i)\right) = \prod_i P(\boldsymbol{z}_i),
$$

*Proof.* The sparsity constraints and sparsity-promoting regularizers for the SAEs under study are specified in the table below.

| SAE | Regularizer $\mathcal{R}(\boldsymbol{z}_i)$ | Sparsity Constraint $\tilde{g}(\boldsymbol{z}_1,\ldots,\boldsymbol{z}_T) = 0$ |
|---|---|---|
| ReLU | $\|\boldsymbol{z}_i\|_1$ | 0 |
| JumpReLU | $\|\boldsymbol{z}_i\|_0$ | 0 |
| TopK | 0 | $\sum_{i=1}^{T}(\|\boldsymbol{z}_i\|_0 - K)^2$ |
| BatchTopK | 0 | $\frac{1}{T}\sum_{i=1}^{T}\|\boldsymbol{z}_i\|_0 - K$ |

Table 5: Sparsity constraints and regularizers for SAEs

Note that TopK imposes a pointwise hard sparsity constraint, which has been restated using sum-of-squares above for convenience. While BatchTopK imposes the fixed mean sparsity for each mini batch, we take the batch to capture the entire timeseries in the above formulation. The above table shows us that the sparsity constraint is additive over time in all cases:

$$
\tilde{g}(\boldsymbol{z}_1,\ldots,\boldsymbol{z}_T) = \sum_{i=1}^{T} \tilde{g}(\boldsymbol{z}_i) = 0,
$$

(9)

$$
\text{where } \tilde{g}(\boldsymbol{z}_i) = \begin{cases} (\|\boldsymbol{z}_i\|_0 - K)^2 & \text{TopK} \\ \frac{1}{T}(\|\boldsymbol{z}_i\|_0 - K) & \text{BatchTopK} \\ 0 & \text{ReLU, JumpReLU} \end{cases}
$$

(10)

Recall that SAEs solve the following constrained optimization problem (restated from Eq. 1).

$$
\arg\min_{\boldsymbol{D},\boldsymbol{z}} \frac{1}{N}\sum_{i=1}^{T} \|\boldsymbol{x}_i - \boldsymbol{D}\boldsymbol{z}_i\|_2^2 + \lambda\mathcal{R}(\boldsymbol{z}_i),
$$

$$
\text{s.t. } \boldsymbol{z}_k = f_{\text{SAE}}(\boldsymbol{x}_k)\ \forall k,\ \tilde{g}(\boldsymbol{z}_1,\ldots,\boldsymbol{z}_T) = 0 .
$$

(11)

We rewrite the above problem using Lagrange multipliers on the sparsity constraints, and further simplify using the above result on constraints being additive over time, as:

$$
\arg\min_{\boldsymbol{D},\boldsymbol{z}} \frac{1}{N}\sum_{i=1}^{T} \|\boldsymbol{x}_i - \boldsymbol{D}\boldsymbol{z}_i\|_2^2 + \lambda\mathcal{R}(\boldsymbol{z}_i) + \tilde{\lambda}\tilde{g}(\|\boldsymbol{z}_i\|_0),
$$

$$
\text{s.t. } \boldsymbol{z}_k = f_{\text{SAE}}(\boldsymbol{x}_k)\ \forall k.
$$

(12)

Note that we don't use Lagrange multipliers on the SAE architecture constraint $\boldsymbol{z} = f_{\text{SAE}}(\boldsymbol{x})$ since we only care about $\boldsymbol{z}$-specific constraints (which don't include the data $\boldsymbol{x}$) for the prior.

**Bayesian Interpretation.** The objective function above (sans the SAE architecture constraint) can be thought of as minimizing the negative log posterior, which is proportional to log prior added to log likelihood:

$$-\log P(\boldsymbol{z}_1, \ldots, \boldsymbol{z}_T \mid \boldsymbol{x}_1, \ldots, \boldsymbol{x}_T) \propto \underbrace{\frac{1}{N} \sum_{i=1}^{T} \|\boldsymbol{x}_i - \boldsymbol{D}\boldsymbol{z}_i\|_2^2}_{-\log P(\boldsymbol{x}_1, \ldots, \boldsymbol{x}_T \mid \boldsymbol{z}_1, \ldots, \boldsymbol{z}_T)} + \underbrace{\frac{1}{N} \sum_{i=1}^{T} \lambda \mathcal{R}(\boldsymbol{z}_i) + \tilde{\lambda}\tilde{g}(\|\boldsymbol{z}_i\|_0)}_{-\log P(\boldsymbol{z}_1, \ldots, \boldsymbol{z}_T)}$$

$$\tag{13}$$

The prior over latents $\boldsymbol{z}$ is:

$$\log P(\boldsymbol{z}_1, \ldots, \boldsymbol{z}_T) = -\frac{1}{N} \sum_{i=1}^{T} \lambda \mathcal{R}(\boldsymbol{z}_i) + \tilde{\lambda}\tilde{g}(\|\boldsymbol{z}_i\|_0), \tag{14}$$

$$\implies P(\boldsymbol{z}_1, \ldots, \boldsymbol{z}_T) = \prod_{i=1}^{T} \exp\left(-\lambda \mathcal{R}(\boldsymbol{z}_i) + \tilde{\lambda}\tilde{g}(\|\boldsymbol{z}_i\|_0)\right) = \prod_i P(\boldsymbol{z}_i). \tag{15}$$

Therefore, the prior is multiplicative over time, implying independence, and the distribution at each time $t$ has the same form, implying that the prior is independent and identically distributed (i.i.d.). This completes the proof.

$$\square$$

### I.2 PRIORS OVER CONCEPTS AND GENERATIVE PRIORS OVER TIME

We can further think of the priors of each SAE over concepts as well as over time in a generative fashion. In some cases, this mainfests as a hierarchical latent variable model $n \to S \to \boldsymbol{z}$, where $n = \|\boldsymbol{z}\|_0$ is the sparsity, $S = \text{supp}(\boldsymbol{z}) = \{k : z^k > 0\}$ is the support, and $\boldsymbol{z}$ is the SAE latent code.

**Proposition I.2** (SAE Priors on Sparse Code). *Let $S_t = \text{supp}(\boldsymbol{z}_t) = \{k : z_t^k > 0\}$ be the set of active latents in the sparse code $\boldsymbol{z}$ at time $t$, and $n_t = |S_t|$ be the cardinality of $S_t$ (the number of active latents). Each SAE imposes a prior distribution on the sparse code $\boldsymbol{z}$, arising from its sparsity penalty $\mathcal{R}(\boldsymbol{z})$ or implicit conditions imposed on the sparse code. These conditions are highlighted in Table 6.*

Table 6: Priors over concept interactions and dynamics for various SAEs

| $f_{\text{SAE}}, \mathcal{R}(\boldsymbol{z})$ | Across-Concept Prior (interaction) | Across-time Prior (dynamics) |
|---|---|---|
| ReLU, $L_1$-norm | $z_t^1, \ldots, z_t^M \overset{\text{i.i.d.}}{\sim} \text{Laplace}(0, \cdot)$ | $\boldsymbol{z}_1, \ldots, \boldsymbol{z}_t \overset{\text{i.i.d.}}{\sim} P_{\boldsymbol{z}}$ |
| TopK | $z_t^{i_1}, \ldots, z_t^{i_K} \mid S_t \overset{\text{i.i.d.}}{\sim} U(0, \cdot) \; \forall i. \in S_t,$ $S_t \sim U([M]^K)$ | $(\boldsymbol{z}_1, S_1), \ldots, (\boldsymbol{z}_t, S_t) \overset{\text{i.i.d.}}{\sim} P_S P_{\boldsymbol{z}\mid S},$ $S_1, \ldots, S_t \overset{\text{i.i.d.}}{\sim} U([M]^K)$ |
| JumpReLU, $L_0$-norm | $z_t^{i_1}, \ldots, z_t^{i_{n_t}} \mid S_t \overset{\text{i.i.d.}}{\sim} U(0, \cdot) \; \forall i. \in S_t,$ $S_t \mid n_t \sim U([M]^{n_t})$ | $(\boldsymbol{z}_1, S_1, n_1), \ldots, (\boldsymbol{z}_t, S_t, n_t) \overset{\text{i.i.d.}}{\sim} P_n P_{S\mid n} P(\boldsymbol{z} \mid S),$ $n_1, \ldots, n_t \overset{\text{i.i.d.}}{\sim} P_n$ |
| BatchTopK | $z_t^{i_1}, \ldots, z_t^{i_{n_t}} \mid S_t \overset{\text{i.i.d.}}{\sim} U(0, \cdot) \; \forall i. \in S_t,$ $S_t \mid n_t \sim U([M]^{n_t})$ | $(\boldsymbol{z}_1, S_1, n_1), \ldots, (\boldsymbol{z}_t, S_t, n_t) \overset{\text{i.i.d.}}{\sim} P_n P_{S\mid n} P(\boldsymbol{z} \mid S),$ $n_1, \cdots, n_t \overset{\text{i.i.d.}}{\sim} P_n, \; \mathbb{E}[n_t] = K$ |

### I.2.1 ReLU SAE

The vanilla ReLU SAE (Bricken et al. (2023), Cunningham et al. (2023)) is trained with the $L_1$-norm penalty:

$$\mathcal{R}(\boldsymbol{z}) = \|\boldsymbol{z}\|_1. \tag{16}$$

The prior over $\boldsymbol{z}$ for the above case is:

$$\log P(\boldsymbol{z}_1, \ldots, \boldsymbol{z}_N) \propto -\sum_{i=1}^{N} \sum_{k=1}^{M} |z_i^k|, \tag{17}$$

$$\implies P(\boldsymbol{z}_1, \ldots, \boldsymbol{z}_N) \propto \prod_{i=1}^{N} \left( \prod_{k=1}^{M} \exp -\nu |z_i^k| \right). \tag{18}$$

This joint distribution implies that for each sample $i$, different indices $k$ are sampled i.i.d. from the same distribution:

$$z_i^1, \ldots, z_i^M \overset{\text{i.i.d.}}{\sim} \text{Laplace}(0, 1/\nu), \tag{19}$$

and different samples are all independently sampled from the same product Laplace distribution:

$$\boldsymbol{z}_1, \ldots \boldsymbol{z}_N \overset{\text{i.i.d.}}{\sim} \text{Laplace}^M(0, 1/\nu). \tag{20}$$

This concludes the proof for priors of ReLU SAE trained with $L_1$ norm sparsity penalty. $\qquad\square$

### I.2.2 TOPK SAE

The TopK SAE (Makhzani and Frey (2013), Gao et al. (2024)) directly controls the sparsity of the representation $\boldsymbol{z}$ by fixing it at $\|\boldsymbol{z}\|_0 = K$, instead of imposing an explicit sparsity penalty $\mathcal{R}(\boldsymbol{z})$ in the loss function. The objective function for TopK SAE is:

$$\underset{\boldsymbol{D}, \boldsymbol{z}}{\arg\min} \sum_{i=1}^{N} \frac{1}{N} \|\boldsymbol{x}_i - \boldsymbol{D}\boldsymbol{z}_i\|_2^2, \tag{21}$$

$$\text{s.t. } \forall j, \boldsymbol{z}_j = f_{TopK}(\boldsymbol{x}_j), \ \|\boldsymbol{z}_j\|_0 = K. \tag{22}$$

Since the fixed sparsity is a hard constraint that depends on $\boldsymbol{z}$ alone (and not the data $\boldsymbol{x}$), it can further be simplified as a sum-of-squares constraint: $\sum_j (\|\boldsymbol{z}_j\|_0 - K)^2 = 0$. We can use Lagrange multipliers to reformulate it as an effective prior:

$$\underset{\boldsymbol{D}, \boldsymbol{z}, \{\lambda_i\}}{\arg\min} \sum_{i=1}^{N} \frac{1}{N} \left( \|\boldsymbol{x}_i - \boldsymbol{D}\boldsymbol{z}_i\|_2^2 + \lambda \left( |\|\boldsymbol{z}_i\|_0 - K| \right)^2 \right), \tag{23}$$

$$\text{s.t. } \forall j, \boldsymbol{z}_j = f_{TopK}(\boldsymbol{x}_j). \tag{24}$$

The prior over $\boldsymbol{z}$ for the above (effective) regularizer is:

$$\log P(\boldsymbol{z}_1, \ldots, \boldsymbol{z}_N) \propto -\sum_{i=1}^{N} \lambda \left( \left( \|\boldsymbol{z}_i\|_0 - K \right)^2 \right) \tag{25}$$

$$\implies P(\boldsymbol{z}_1, \ldots, \boldsymbol{z}_N) \propto \prod_{i=1}^{N} \exp\left( -\lambda \left( \|\boldsymbol{z}_i\|_0 - K \right)^2 \right) \tag{26}$$

Note that the above prior is finite for finite values of $\lambda$, but the overall objective optimizes over $\lambda$, resulting in a *hard* prior peaked at $\|\boldsymbol{z}_i\|_0 = K$ for each sample $i$.

The factorization over samples $i$ implies mutual independence of $\boldsymbol{z}_1, \ldots, \boldsymbol{z}_n$: $P(\boldsymbol{z}_1, \ldots, \boldsymbol{z}_n) = \prod_{i=1}^{N} P(\boldsymbol{z}_i)$.

As defined in Theorem I.2 (and restated here for convenience), let $S_i = \text{supp}(\boldsymbol{z}_i) = \{k : z_i^k > 0\}$, $n_i = |S_i| = \|\boldsymbol{z}_i\|_0$ denote the active indices and their number (sparsity) respectively.

For individual samples $\boldsymbol{z}_i$, if we condition on the set of active indices $S_i$, the sparsity gets fixed since $\|\boldsymbol{z}_i\|_0 = |S_i| = n_i$, and the distribution becomes constant:

$$P(\boldsymbol{z}_i \mid S_i) = C \tag{27}$$

$$\implies z_i^\mu \mid S_i \sim \begin{cases} U(0, \kappa) & \mu \in S_i \\ \delta_0 & \mu \notin S_i \end{cases}, \text{ and} \tag{28}$$

$$z_i^{\mu_1}, \ldots, z_i^{\mu_{|S_i|}} \mid S_i \overset{\text{i.i.d.}}{\sim} U(0, \kappa) \text{ for } \mu. \in S_i \tag{29}$$

where $C, \kappa$ are appropriate constants.

Since $\{\boldsymbol{z}_i\}_i$s are mutually independent, any measurable function of each is also independent. The indices of nonzero entries of $\boldsymbol{z}_j$, i.e., $S_j$ is a measurable function since it is a map $S : \mathbb{R}_+^M \to 2^M$

which is discrete valued, and pre images of each value—a set of nonzero indices—are measurable since they equal the cartesian products of the measurable sets $\{z = 0\}, \{z > 0\}$ over all indices. Hence, $S_1, \ldots, S_n$ are also independent.

Since $S_i = g(\boldsymbol{z}_i)$ and the distribution of $\boldsymbol{z}_i$ depends only on $n_i = \|\boldsymbol{z}_i\|_0$ (Eq. 25), the distribution of $S_i$ will also depend only on $n_i$, becoming uniform when conditioned on $n_i$. In TopK SAE, $n_i = K$ is a constant. Therefore, each $S_i \sim U([M]^K)$, and together with independence argued above,

$$S_1, \ldots, S_N \overset{\text{i.i.d.}}{\sim} U([M]^K) \tag{30}$$

This completes the proof for the priors of TopK SAE. $\qquad\square$

### I.2.3 BATCHTOPK SAE

BatchTopK SAE (Bussmann et al. (2024)) is a modification of the TopK SAE. Instead of fixing sparsity like TopK, BatchTopK allows variable sparsity per input while fixing the mean sparsity over a batch at $K$. The objective function for BatchTopK SAE can equivalently be written as:

$$\underset{\boldsymbol{D}, \boldsymbol{z}}{\arg\min} \sum_{i=1}^{N} \frac{1}{N} \|\boldsymbol{x}_i - \boldsymbol{D}\boldsymbol{z}_i\|_2^2, \tag{31}$$

$$\text{s.t. } \forall j, \boldsymbol{z}_j = f_{TopK}(\boldsymbol{x}_j), \ \frac{1}{N} \sum_{j=1}^{N} \|\boldsymbol{z}_j\|_0 = K. \tag{32}$$

While BatchTopK imposes a mean sparsity per batch, for simplicity, we use the batch size to match the size of the entire dataset (WLOG). Smaller batch sizes can easily be incorporated by adding separate constraints, each over the entire batch (only leads to a change in constants—lagrange multipliers—in the analysis).

Following similar analysis as for TopK SAE (App. I.2.2), we can derive an equivalent prior over $\boldsymbol{z}$ for BatchTopK SAE:

$$P(\boldsymbol{z}_1, \ldots, \boldsymbol{z}_N) \propto \prod_{i=1}^{N} \exp\left(-\lambda \big|\|\boldsymbol{z}_i\|_0 - K\big|\right) \tag{33}$$

The sparse codes for different samples $\{\boldsymbol{z}_i\}_i$ are thus sampled i.i.d. from a distribution that only depends on the sparsity penalty. While this prior looks very similar to the prior of TopK SAE, the difference is that in TopK, the fixed sparsity constraint is imposed per sample, leading to a different Lagrange multiplier $\lambda_i$ per sample to optimize over, while in BatchTopK, we have a common multiplier $\lambda$ over all examples in a batch (with multiple batches, we will have one multiplier per batch), which is then optimized over to ensure that average sparsity per batch constraint is met.

Similar to the analysis for the TopK SAE, we get the following prior over different latents per sample:

$$z_i^{\mu} \mid S_i \sim \begin{cases} U(0, \kappa) & \mu \in S_i \\ \delta_0 & \mu \notin S_i \end{cases} \text{, and} \tag{34}$$

$$z_i^{\mu_1}, \ldots, z_i^{\mu_{|S_i|}} \mid S_i \overset{\text{i.i.d.}}{\sim} U(0, \kappa) \ \text{ for } \mu_. \in S_i \tag{35}$$

The active indices $S_i$ are sampled uniformly conditioned on the number of active indices $n_i$:

$$S_i \mid n_i \sim U([M]^{n_i}) \tag{36}$$

The number of active latents $n_i$ are themselves sampled i.i.d. (since $n_i = \tilde{g}(\boldsymbol{z}_i)$ and $\{\boldsymbol{z}_i\}_i$ are i.i.d.) from a distribution whose mean is fixed:

$$n_1, \ldots, n_N \overset{\text{i.i.d.}}{\sim} P, \ \text{s.t. } \mathbb{E}[n_.] = K \tag{37}$$

This completes the derivation for the BatchTopK prior. $\qquad\square$

### I.2.4 JUMPRELU SAE

JumpReLU SAE (Rajamanoharan et al. (2024)) is trained with the $L_0$ (pseudo-)norm regularizer. This leads to the following optimization problem:

$$\underset{\boldsymbol{D},\boldsymbol{z}}{\arg\min} \sum_{i=1}^{N} \frac{1}{N} \left( \|\boldsymbol{x}_i - \boldsymbol{D}\boldsymbol{z}_i\|_2^2 + \lambda\|\boldsymbol{z}_i\|_0 \right)$$

$$\text{s.t. } \forall k, \boldsymbol{z}_k = f_{JumpReLU}(\boldsymbol{x}_k)$$

This objective is equivalent to the following prior over $\boldsymbol{z}$:

$$P(\boldsymbol{z}_1, \ldots, \boldsymbol{z}_N) \propto \prod_{i=1}^{N} \exp\left( -\eta\|\boldsymbol{z}_i\|_0 \right) \tag{38}$$

Noting the similarity with the TopK/ BatchTopK cases, we use the same analysis to derive the following conditions:

$$z_i^\mu \mid S_i \sim \begin{cases} U(0, \kappa) & \mu \in S_i \\ \delta_0 & \mu \notin S_i \end{cases}, \text{ and} \tag{39}$$

$$z_i^{\mu_1}, \ldots, z_i^{\mu_{|S_i|}} \mid S_i \overset{\text{i.i.d.}}{\sim} U(0, \kappa) \text{ for } \mu_. \in S_i \tag{40}$$

$$S_i \mid n_i \sim U([M]^{n_i}) \tag{41}$$

The number of active latents $n_i$ are again i.i.d., but there is no constraint on the mean of the distribution (unlike BatchTopK which constrained the mean of $n_i$ to equal $K$):

$$n_1, \ldots, n_N \overset{\text{i.i.d.}}{\sim} P, \tag{42}$$

which completes the analysis for JumpReLU SAE. $\qquad\square$

## J STATIONARITY MEASURES

LLM activations are empirically non-stationary across the sequence. We quantify the non-stationary nature by measuring autocorrelations and the U-statistic.

### J.1 AUTOCORRELATION

We compute autocorrelation by selecting evenly spaced tokens across the sequence and measuring the cosine similarity between each token and tokens at various lags in the past. Specifically, for tokens at position $t$, we compute similarities to tokens at $t - w$ where lag $w$ ranges from 5 to 20. This creates a heatmap where rows represent lag offsets and columns represent token positions.

For a stationary process, we expect the autocorrelation pattern to remain consistent across time—that is, the relationship between a token and its historical context should be similar regardless of position in the sequence. This would manifest as similar autocorrelation patterns repeating horizontally across token positions. In contrast, for a non-stationary process where representations evolve over time, we expect the autocorrelation patterns to vary systematically across positions, with columns showing different temporal dependency structures as the sequence progresses.

### J.2 U-STATISTIC

We measure the effective dimensionality of LLM representations using a U-statistic based on pairwise cosine similarities.

$$\text{U-stat}(t) = \frac{M^2 - M}{\|\mathbf{G}_t\|_F^2 - M} \tag{43}$$

where $\|\mathbf{G}_t\|_F^2$ is the squared Frobenius norm of the Gram matrix. This quantity estimates the effective rank $1/\text{tr}(\mathbf{C}_t^2)$, where $\mathbf{C}_t = \mathbb{E}[\hat{\boldsymbol{x}}_t\hat{\boldsymbol{x}}_t^T]$ is the second moment matrix and $\hat{\boldsymbol{x}}_t$ is the normalized activation vector at time $t$. Under stationarity, U-stat remains constant. When representations evolve over time, U-stat increases systematically as more orthogonal directions become active.

### J.3 SURROGATE

For the U-statistic and Autocorrelation metrics, we compare LLM activations to surrogate distributions that preserve certain statistical properties while removing temporal structure. We operate on representation vectors $\mathbf{X} \in \mathbb{R}^{B \times T \times d}$, where $B$ denotes batch size, $T$ denotes sequence length, and $d$ denotes the model dimension.

**U-statistic surrogate (Fig. 2 a, e):** For each sequence $i \in \{1, \ldots, B\}$, we construct the surrogate $\tilde{\mathbf{X}}_i$ by applying a random permutation $\pi_i : \{1, \ldots, T\} \to \{1, \ldots, T\}$ to the temporal positions:

$$\tilde{\mathbf{X}}_{i,t,:} = \mathbf{X}_{i,\pi_i(t),:} \quad \forall t \in \{1, \ldots, T\}$$

This preserves the marginal distribution of activations within each sequence while destroying temporal dependencies.

**Autocorrelation surrogate (Fig. 2 c, g):** Given the similarity matrix $\mathbf{S} \in \mathbb{R}^{T \times T}$ where $S_{ij} = \text{sim}(\mathbf{X}_{.,i,.}, \mathbf{X}_{.,j,.})$, we construct the surrogate similarity matrix $\tilde{\mathbf{S}}$ by replacing each diagonal with its mean:

$$\tilde{S}_{ij} = \bar{S}_k \quad \text{where } k = |i - j|, \quad \bar{S}_k = \frac{1}{T-k} \sum_{t=1}^{T-k} S_{t,t+k}$$

This preserves the average correlation structure at each lag while removing position-specific temporal patterns.

## K    Use of Large Language Models

LLMs were used in this work for the following:

- **Polish writing**: Although major parts of the writing were done by the authors themselves, LLMs (ChatGPT) were used to critique and iteratively improve the writing.
- **Research ideation**: In the initial stages of the project, conversations with LLMs (ChatGPT, Gemini) aided in refining the overall storyline of the project, as well as to get feedback on theory sections. In all cases, LLM outputs were only used by the authors to refine their ideas.

