# OpenReview forum: "Priors in time: Missing inductive biases for language model interpretability"
_ICLR.cc/2026/Conference — ICLR 2026 Poster_

### Official Review · Reviewer_mbJz · 2025-10-30

**Soundness:** 3
**Presentation:** 3
**Contribution:** 3
**Rating:** 6
**Confidence:** 4

**Summary:**

This paper examines the implicit temporal assumptions underlying Sparse Autoencoders (SAEs) widely used in mechanistic interpretability. It reveals that the conventional SAE training objective imposes an implicit independence condition on the prior it assigns to the distribution of feature activations across different token positions—an assumption inconsistent with the highly correlated and nonstationary nature of language sequences. Through empirical analyses of Llama-3.1-8B and Gemma-2-2B, the authors show that the dimensionality and autocorrelation of token representations evolve systematically with position. To address this, they introduce the Temporal Sparse Autoencoder, which decomposes each activation into a predictable component derived from past context and a novel component for the current token capturing new information, akin to the innovation concept in signal processing. The Temporal SAE uncovers temporally structured linguistic features—such as event boundaries in narratives and reanalysis in garden-path sentences—that standard SAEs fail to reveal, underscoring the need for temporal inductive biases in interpretability methods.

**Strengths:**

1. The primary advantage of this paper lies in its thoughtful incorporation of linguistic insights into the analysis, design, and evaluation of SAE methods and LLM representations. The empirical experiments in Section 4, presented in Figure 2, clearly show that LLM representations of token sequences conform to the non-stationarity of natural language, featuring continuous increments of new information regulated by underlying dependency structures. The evaluation of the proposed Temporal SAE on garden-path sentences is both novel and insightful, as the dependency parsing of such sentences is a classic topic in linguistics. The superior performance of the Temporal SAE on these examples effectively demonstrates the benefits of explicitly incorporating temporal and linguistic inductive biases into the interpretability framework.

2. The spectral analysis of SAE and LLM representations at the end of Section 6 is particularly insightful and adds an important dimension to the depth of the work. It convincingly shows that the decomposition of LLM representations by the Temporal SAE into predictable and novel components successfully captures the bipartite structure of representations—comprising a relatively slow-varying component and a more dynamic component with respect to the introduction of new tokens. This analysis reinforces the interpretability and mechanistic validity of the proposed temporal modeling approach.

**Weaknesses:**

1. One of the primary motivations (and functions) of the original SAE is to decompose the polysemous LLM representations into distinct monosemantic features that are more interpretable and less ambiguous, which can further be utilized in steering experiments. This paper, however, makes no attempt to semantically interpret the learned features in this regard—likely because the architectural design of the Temporal SAE, where the feature activations of the predictable part are obtained via an attention mechanism, complicates the interpretation of both activations and activation features.

2. The paper suffers from a lack of clarity with respect to concepts and notations in certain parts, which I list below.

- **Equation 2**: The normalization factor should be $\frac{1}{T}$.
- **Section G.2, Equation 40**: $M$ is not defined, which seems to refer to the number of tokens. For the U-statistic to correctly estimate the effective rank, it appears that the model representations must be normalized; however, this is not stated.
- **Line 291**: The authors claim $x_{n,t} \perp x_{p,t}$, but this is not structurally guaranteed by the Temporal SAE (e.g., via a projection-based approach). The authors themselves acknowledge in Line 338 that the orthogonality is only approximate. They might consider rephrasing the argument in Line 291 for precision.
- **Predictable component definition**: The authors state that they use a self-attention layer on top of a single ReLU layer to model $f$ for obtaining the predictable component. However, Figure 3 suggests that the attention layer depends on the SAE activations $v_1, ..., v_t$, and the paper does not specify how these activations are obtained.
- **Baselines in Figure 2(d) and (h)**: The authors provide a baseline for their variance preservation experiments using representations of a certain number of nearest token representations. However, the method for computing this baseline is never stated (presumably using an equal number of random directions). I also recommend adding another baseline using projections based on the first $k$ tokens’ representations in the context, which could further substantiate the authors’ claims.

**Questions:**

1. What are $W$ and $L_0$ in Table 1?

2. What is the off-the-shelf algorithm used to extract hierarchical structures from the correlation plot, and how does it work?

3. Do the “predictive codes (and novel codes, respectively)” frequently mentioned in Section 6 refer to the SAE activations $z_{p,t}$ or to $x_{p,t}$?

4. Could you provide an (at least intuitive) explanation for why the slow-moving part of the representations (and the predictive codes) exhibits a faster spectral decay?

---

> ### Author Response · Authors · 2025-11-24
> **Rebuttals (1/3)**
>
> We thank the reviewer for their thoughtful and detailed feedback! We really liked their description of our work as “thoughtful incorporation of linguistic insights into design of SAEs”, “evaluation … is novel and insightful”, “spectral analysis of representations is insightful”. We provide responses to specific questions raised in the review below.
>
> ----
>
> > **One of the primary motivations (and functions) of the original SAE is to decompose the polysemous LLM representations...**
>
> Great point! Before responding to the interpretability afforded via Temporal SAEs, we would like to clarify the primary goal our our work: we intended to demonstrate that temporal structures in language model activations cannot be reliably reflected by a linear combination of monosemantic vectors, as assumed by existing SAEs. For example, temporal structures such as event boundaries, parsing of sentences, etc., do not respect the original motivation of SAEs, which is rightly pointed out by the reviewer as being to “decompose the polysemous LLM representations into distinct monosemantic features that are more interpretable and less ambiguous”. We make this mismatch explicit in the paper, showing standard SAEs make assumptions about language model activations that empirically does not hold true. In this sense, we note our primary goal with introducing Temporal SAEs was fundamentally different from standard SAEs’ goal of identifying monosemantic latents: we defined Temporal SAEs to identify temporal structures in language models’ activations (via the predictive code), while also isolating monosemantic latents identifying via standard SAEs (via the novel code). To show that Temporal SAEs achieved this goal, we have added several new results, as summarized below.
>
> - **Predictive codes capture global structure in stories:** We have added low-dimensional visualizations of codes from different SAEs extracted using UMAP from stories (see Fig. 4 and App. G). We consistently see predictive codes from Temporal SAE cluster up tokens according to events (as highlighted in the dendrograms in Fig. 4b and ones shown in App. G). We find similar results from Temporal SAEs trained on Llama as well (see App. H). Interestingly, these results generalize to OOD domains like chat: specifically, when we use Temporal SAEs on activations extracted Gemma-2-2b-IT (a chat model), we see two manifold-like structures emerge that capture semantics of user vs. model. We in fact found these manifolds can be further decomposed into sub-manifolds that correspond to specific domains like advice on code and food recipes—these sub-manifolds are as of yet not annotated in the figure, but we will do so for the final version of the paper.
>
> - **Quantifying event boundaries suggested via predictive codes:** Motivated by the findings above, we have added a new experiment to quantify how well different SAEs identify event boundaries. Specifically, we manually wrote a set of stories with precise event boundaries (see Fig. 6a), used GPT-5 to generate 50 synthetic stories with similar structure, and then inspired by the study by Georgiou et al. [1], had GPT-5 label the event boundaries in these stories. We then measured how clustered the representations of tokens belonging to the same vs. different events are. In particular, we computed the average cosine similarity of token representations when they come from the same event versus different ones. As shown in Fig. 6c, we find Temporal SAEs’ predictive codes achieve substantially higher cosine similarity than other SAEs–as expected by clustering exhibited in similarity heatmaps in Fig. 6b. To further elaborate on this point, we took inspiration from literature on graph clustering and added noise to the input representations before computing codes and computing similarity heatmaps: this process should yield coarser clusters since high frequency signals will get overwhelmed by noise. Results are shown in Fig. 6d,e, and show a clear signature of coarser clusters emerging under increasing noise scale, demonstrating the multiscale nature of language!
>
> - **Novel codes from Temporal SAEs and codes from standard SAEs behave similarly:** UMAP visualizations of novel codes from Temporal SAEs and standard SAEs’ codes show they primarily cluster representations according to lexical information, i.e., the inputted token (see Fig. 4b and App. G, H). In fact, the code similarity heatmaps produced by novel codes match very well different SAEs’ latent codes similarity maps. Motivated by this, we ran auto-interpretability experiments with novel codes of Temporal SAEs and standard SAEs, finding features identified from standard SAEs had counterparts in novel codes of Temporal SAEs. We promise these autointerpretability experiments (with the option to perform live steering) and a new heatmap visualization tool to assess similarity maps from predictive codes will be made available via Neuronpedia if the paper is accepted.
>
> **(Continued below...)**

---

> ### Author Response · Authors · 2025-11-24
> **Rebuttals (2/3)**
>
> - **Assessing causality via SAEBench:** To address the reviewer’s comment on investigating steering ability of codes extracted using Temporal SAEs, we are running experiments on a subset of SAEBench [2] evaluations that gauge causality of assigned interpretations. We note the precise protocol for steering predictive codes is unclear, as noted by the reviewer: the apt notion of intervention likely involves developing tooling that allows one to move on a manifold-like structure isolated using predictive codes; we are actively pursuing this question, but believe it warrants its own detailed study. Thus, for now, we specifically focus on the novel codes extracted using Temporal SAEs for this experiment. In the Spurious Correlation Removal metric, temporal novel codes achieve a score of 0.16, while the TopK SAE baseline scores 0.25. Both codes significantly outperform the PCA baseline score 0.03. We are still running the Targeted Probe Perturbation (TPP) and RAVEL evaluations, which are currently bottlenecked by some code refactorization issues to run evaluations, but promise to include the numbers in the final version of the paper.
>
> [1] https://learnmem.cshlp.org/content/32/2/a054043.short
> [2] https://arxiv.org/abs/2503.09532
>
>
> ----
>
> > **Equation 2: The normalization factor should be 1/T**
>
> Thank you for highlighting this typo! We have fixed this in the new draft.
>
> ----
>
> > **Section G.2, Equation 40: M is not defined, which seems to refer to the number of tokens. For the U-statistic to correctly estimate the effective rank, it appears that the model representations must be normalized; however, this is not stated.**
>
> Apologies for this oversight! We have included the definition of M in the updated version of the paper. We indeed normalize the model representations when computing the U-statistic, and have updated the App. A to explicitly mention this. We have also significantly expanded App. A to clarify details for other experiments.
>
> ----
>
> > **Line 291: ...The authors themselves acknowledge in Line 338 that the orthogonality is only approximate. They might consider rephrasing the argument in Line 291 for precision.**
>
> We agree! The approximate orthogonality is emergent and not a strictly imposed constraint on the architecture. When we observed this orthogonality, we realized that this aligns well with our decomposition of predictive vs. novel codes and hence decided to state it in the SAE formulation. However, the reviewer’s comment is correct that this makes the architecture definition imprecise, since the architecture does not have any explicit orthogonality constraint. Accordingly, we have rephrased the discussion around line 338 (now line 311 in the updated paper) and removed the orthogonality constraint from Line 291 (now line 271 in the updated paper). Thank you for this comment!
>
> ----
>
> > **Predictable component definition**
>
> Thank you for pointing out that our description of the architecture was unclear! The inference of Temporal SAE starts by projecting language model activations $x$ into a higher dimensional latent space spanned by $D^T$, yielding $v$. The Attention Layer is applied to $v$ to output $z_{p,t}$, which is subsequently transformed into $x_{p,t}$. Crucially, we note none of these computations depend on the SAE used for computing the novel code. Instead the SAE is dependent on the Attention layer, since it is trained to reconstruct $x_{n,t} = x_{t} - x_{p,t}$. We’ve updated the description of the architecture in Section 4 for clarity—thank you for the comment!
>
> ----
>
> > **Baselines in Figure 2(d) and (h): The authors provide a baseline for their variance preservation experiments... However, the method for computing this baseline is never stated (presumably using an equal number of random directions)...**
>
> In the original submission we used the mean previous representation across the dataset as a baseline. However, your suggestion of using an equal number of random directions is more principled. We thank you for your suggestions and add both suggested baselines to Figure 2 in the updated manuscript.
>
> We observe that context representations significantly outperform random directions in explaining the context. Notably, at token position 500, the random projections span a subspace of about 20% of the number of residual stream dimensions in Gemma-2-2B, and only explain the same amount of variance as the (one-dimensional) representation at the previous token position explains! Furthermore, while the variance explained by the previous token remains constant over time, the variance explained by the first token decays over time. The first token baseline demonstrates that the immediate context is more relevant in explaining a representation than more distant representations.
>
> ----
>
> **(Continued below...)**

---

> > ### Author Response · Authors · 2025-11-24
> > **Rebuttals (3/3)**
> >
> > > **What are W  and L0 in Table 1?**
> >
> > Apologies for the confusion! W corresponds to the dictionary width, and L0 corresponds to the mean sparsity of novel codes.
> >
> > ----
> >
> > > **What is the off-the-shelf algorithm used to extract hierarchical structures from the correlation plot, and how does it work?**
> >
> > We apologize if this was not clear. We ran agglomerative clustering using the default function offered by SciPy [1] on cosine similarity maps defined using different codes, i.e., predictive / novel codes from Temporal SAEs or latent codes from standard SAEs. Specifically, we take a single input (e.g., a story), compute codes based on the protocol we're evaluating, define a cosine similarity map, and then run agglomerative clustering on it using [1]. We have added a reference to the algorithm in the main text to clarify this, and have also improved Appendix A to clarify the experimental details.
> >
> > [1] https://docs.scipy.org/doc/scipy/reference/generated/scipy.cluster.hierarchy.dendrogram.html
> >
> > ----
> >
> > > **Do the “predictive codes (and novel codes, respectively)” frequently mentioned in Section 6 refer to the SAE activations or...?**
> >
> > The predictive codes and novel codes refer to Temporal SAE's activations for the predictive and novel codes, i.e., $z_{p,t}$, $z_{n,t}$ respectively. Apologies for the confusion; we've updated the Section 5 to clarify this further.
> >
> > ----
> >
> > > **Could you provide an (at least intuitive) explanation for why the slow-moving part of the representations (and the predictive codes) exhibits a faster spectral decay?**
> >
> > Good question! For the slow-moving part of the representation, we note that this part is defined via low-frequency components extracted using a fourier transform applied to the activations; accordingly, it must by definition have a slower spectral decay. However, for the predictive codes, we note that such a slow spectral decay can be expected only if they capture the lower-order frequencies. This is exemplified in Tab. 4, where we see high kernel similarity of predictive codes with slow-moving part of the representations.
> >
> > ----
> > ----
> >
> > **Summary:** We thank the reviewer for their feedback and suggestions that helped improve the clarity of our paper! We hope our clarifications to their comments help address any remaining concerns and, if so, hope they will continue to champion our paper during the discussion period.

---

### Official Review · Reviewer_r8pv · 2025-11-02

**Soundness:** 2
**Presentation:** 3
**Contribution:** 2
**Rating:** 4
**Confidence:** 4

**Summary:**

The paper points out current SAE are built on a flawed assumption. They adopt a Bayesian view and show that standard SAEs implicitly impose a "prior" that the concepts they extract are independent and identically distributed across time. However, language is full of temporal structure and correlations. Authors introduce a new architecture, the Temporal SAE, which decomposes the activation into two parts: A predictable component and A novel component.

**Strengths:**

1. The topic of decomposes the activation into two parts is interesting.
2. The core problem and flawed assumption that standard SAE has is well introduced.
3. The proposed method is interesting.
4. The temporal SAE can successfully identify concept that standard SAE can't

**Weaknesses:**

1. The author claims their temporal SAE is better than standard SAE. However, sometimes it is really difficult to evaluate an SAE, since there is no "ground truth". Therefore, the author should evaluate their SAE on some downstream tasks such as model steering.
2. What about the interpretable concepts in the predictable component? Why are we not finding interpretable features in predictable component as well?
3. The paper never evaluates its new features on any of downstream tasks. The evaluations are purely analytical. The paper only shows that its features correlate with these temporal structures, but it never proves that these features are more useful. The big "so what?" question is unanswered.
4. This paper is built on the assumption that the novel component is orthogonal to the predictable component. However, sometimes new information doesn't just add a new, independent concept; it often refines or modifies existing, predictable concepts. Plus, I didn't see any citation prove this assumption in the paper.
5. The paper miss some important related work.

[1]: A Survey on Sparse Autoencoders: Interpreting the Internal Mechanisms of Large Language Models

**Questions:**

1. What is the difference between your method:
"input a sentence -> find the novel component at each token -> convert this novel component into human understandable concept",
and method where you just input tokens one by one, and convert them into human understandable concept?

---

> ### Author Response · Authors · 2025-11-24
> **Rebuttals (1/3)**
>
> We thank the reviewer for their feedback! We are glad the reviewer appreciated our proposed decomposition of activations into a predictable and novel part, the proposed methodology of Temporal SAE “interesting”, and our combined empirical and theoretical insights to exemplify “flawed assumptions of standard SAEs” as “well introduced”. Below, we respond to specific questions raised by the reviewer.
>
> ----
>
> > **The author claims their temporal SAE is better than standard SAE. However, sometimes it is really difficult to evaluate an SAE, since there is no "ground truth".**
>
> Great point! We completely agree that a challenge with all unsupervised methods, like SAEs, is lack of a ground truth to perform meaningful evaluations. Motivated by your comments, we decided to run new evaluations with well-defined ground truth to help make the qualitative claims in our work more quantitatively precise. Specifically, recall that our proposed protocol of Temporal SAEs has two components: a predictive one and a novel one. We report below experiments corresponding to each.
>
>  - **Quantifying event boundaries suggested via predictive codes:** In the submitted draft, we showed qualitative results highlighting predictive codes from Temporal SAEs capture event boundaries in stories. To make this claim more precise, we have added new experiments in the revised draft. Specifically, we manually wrote a set of stories with precise event boundaries (see Fig. 6a), used GPT-5 to generate 50 synthetic stories with similar structure, and then inspired by the study by Georgiou et al. [1], had GPT-5 label the event boundaries in these stories. We then measured how clustered the representations of tokens belonging to the same vs. different events are. In particular, we computed the average cosine similarity of token representations when they come from the same event versus different ones. As shown in Fig. 6c, we find Temporal SAEs’ predictive codes achieve substantially higher cosine similarity than other SAEs---as expected by clustering exhibited in similarity heatmaps in Fig. 6b. To further elaborate on this point, we took inspiration from literature on graph clustering and added noise to the input representations before computing codes and computing similarity heatmaps: this process should yield coarser clusters since high frequency signals will get overwhelmed by noise. Results are shown in Fig. 6d,e, and show a clear signature of coarser clusters emerging under increasing noise scale, demonstrating the multiscale nature of language!
>
>  - **Quantifying parse sensitivity in garden path sentences:** As noted in the submitted draft, predictive codes from Temporal SAEs show highest similarity between tokens that are necessary for making an accurate parse of the sentence. This is specifically exemplified by results on garden path sentences, where a reader is likely to misinterpret the sentence if they follow the typical interpretation of a word. While our results in the original draft were merely qualitative, we have now added a quantitative evaluation of our claim. Specifically, if our claim above is true, then predictive codes extracted from a subject phrase should be similar to the verb phrase regardless of the subject entity, since the correct parse will have the subject perform an action defined by the verb on the object phrase---results in Fig. 7 (b, c) show this empirically holds true, with standard SAEs and novel codes showing variance depending on the precise subject entity. Since computing a correct parse requires assessing the relationship between different constituents of a sentence, i.e., a temporal structure, the ability of predictive codes to score well on this task corroborates its utility as a construct capable of identifying temporal structure from model activations.
>
>  - **Predictive codes capture global structure in stories:** We have added low-dimensional visualizations of codes from different SAEs extracted using UMAP from stories (see Fig. 4 and App. G). We consistently see predictive codes from Temporal SAE cluster up tokens according to events (as highlighted in the dendrograms in Fig. 4b and ones shown in App. G). We find similar results from Temporal SAEs trained on Llama as well (see App. H). Interestingly, these results generalize to OOD domains like chat: specifically, when we use Temporal SAEs on activations extracted Gemma-2-2b-IT (a chat model), we see two manifold-like structures emerge that capture semantics of user vs. model. We in fact found these manifolds can be further decomposed into sub-manifolds that correspond to specific domains like advice on code and food recipes---these sub-manifolds are as of yet not annotated in the figure, but we will do so for the final version of the paper.
>
> **(Continued below...)**

---

> ### Author Response · Authors · 2025-11-24
> **Rebuttals (2/3)**
>
> - **Novel codes from Temporal SAEs and codes from standard SAEs behave similarly:** UMAP visualizations of novel codes from Temporal SAEs and standard SAEs’ codes show they primarily cluster representations according to lexical information, i.e., the inputted token (see Fig. 4b and App. G, H). In fact, the code similarity heatmaps produced by novel codes match very well different SAEs’ latent codes similarity maps. Motivated by this, we ran auto-interpretability experiments with novel codes of Temporal SAEs and standard SAEs, finding features identified from standard SAEs had counterparts in novel codes of Temporal SAEs. We promise these autointerpretability experiments (with the option to perform live steering) and a new heatmap visualization tool to assess heatmap visualizations from predictive codes will be made publicly available via Neuronpedia if the paper is accepted.
>
>  - **Assessing causality via SAEBench:** To address the reviewer’s comment on investigating steering ability of codes extracted using Temporal SAEs, we are running experiments on a subset of SAEBench [2] evaluations that gauge causality of assigned interpretations. We note the precise protocol for steering predictive codes is unclear, as noted by the reviewer: the apt notion of intervention likely involves developing tooling that allows one to move on a manifold-like structure isolated using predictive codes; we are actively pursuing this question, but believe it warrants its own detailed study. Thus, for now, we specifically focus on the novel codes extracted using Temporal SAEs for this experiment. In the Spurious Correlation Removal metric, temporal novel codes achieve a score of 0.16, while the TopK SAE baseline scores 0.25. Both codes significantly outperform the PCA baseline score 0.03. We are still running the Targeted Probe Perturbation (TPP) and RAVEL evaluations, which are currently bottlenecked by some code refactorization issues to run evaluations, but promise to include the numbers in the final version of the paper.
>
> [1] https://learnmem.cshlp.org/content/32/2/a054043.short
> [2] https://arxiv.org/abs/2503.09532
>
> ----
>
> > **What about the interpretable concepts in the predictable component?**
>
> Thank you for the question! As noted in our response above, when the property one is looking to explain requires analyzing the in-context, temporal structure of an input, the predictive component from Temporal SAEs offers a new affordance that prior SAEs fail to capture: understanding temporal structure via similarity maps. Similar to how max-activating examples help define explanations for a model’s behavior in standard SAEs, similarity heatmaps help predictive codes isolate temporally coherent structures from a model’s representations. This allows us to identify event boundaries in a story (Fig. 6, 4b), helps explain the temporal straightening effect postulated by recent papers [1] (Fig. 4a), and offer a hypothesis for how language models are able to parse garden-path sentences (Fig. 7).
>
> [1] https://arxiv.org/abs/2311.04930
>
> ----
>
> > **The paper never evaluates its new features on any of downstream tasks. The evaluations are purely analytical.**
>
> We refer the reviewer to our responses above that clarify the utility of predictive and novel codes extracted using Temporal SAE. We are happy to answer further questions on this point! More broadly, we emphasize that Temporal SAE is one contribution of our paper, and our main objective, as suggested by the title and introduction, is to show that existing SAEs cannot identify temporal structures of interest in language.
>
> ----
>
> **(Continued below...)**

---

> > ### Author Response · Authors · 2025-11-24
> > **Rebuttals (3/3)**
> >
> > > **This paper is built on the assumption that the novel component is orthogonal to the predictable component.**
> >
> > Thank you for this comment! We believe we made an error in writing that led to confusion around this point in the original draft. Specifically, during the development of Temporal SAE, we found the predicted and novel codes yield orthogonal errors, i.e., they capture different parts of the inputted signal. Given that this finding aligns well with our motivation of separating the predictable and novel components, we decided to add this as a constraint in our assumed generative process. However, as the reviewer pointed, this is not a necessary constraint (hence a lack of reference to this point) and in fact there is no explicit bias in the Temporal SAE architecture to induce this constraint. Instead, the orthogonal errors observed from the predictive and novel part of Temporal SAEs is an emergent property. We have fixed this misunderstanding in the paper now: we have removed the orthogonality constraint from the generative model in Eq. 3 (since it is not a necessary constraint) and improved the discussion of orthogonality observed in predictive and novel codes in paragraph “Sanity checking temporal SAEs” on page 6.
> >
> > ----
> >
> > > **The paper miss some important related work. [1]: A Survey on Sparse Autoencoders: Interpreting the Internal Mechanisms of Large Language Models**
> >
> > Thank you for pointing this out! We have added this and several more references to the updated version of the paper.
> >
> > ----
> >
> > > **What is the difference between your method: "input a sentence -> find the novel component at each token -> convert this novel component into human understandable concept", and method where you just input tokens one by one, and convert them into human understandable concept?**
> >
> > Great question! This hits at heart of the point we intend to highlight through our work: the predictive codes in Temporal SAE (our method) can capture temporal structures in language that cannot be captured by looking at individual tokens, as suggested by the reviewer’s methodology. For example, in a story, the primary semantics are not so much who the involved characters are, but instead how these characters relate to each other. That is, the in-context defined structure between different entities endows meaning to their actions, intentions, and goals. We argue in our paper that existing SAEs—which follow the token-by-token interpretation methodology suggested by the reviewer—cannot be used to characterize such temporal structure since they assume tokens are independent of each other. This is most clearly seen in Fig. 4, where we see standard SAEs cluster tokens by their lexical information and lack any useful temporal coherence expected from event structures in stories. As language models get used in tasks that involve longer horizons, e.g., agentic workflows, we believe regarding the time axis as a first-class citizen is critical in interpretability and hence methods like Temporal SAE will be eventually needed.
> >
> > ----
> > ----
> >
> > **Summary:** We thank the reviewer for their thoughtful feedback that helped improve the quality of our paper! In response to their comments, we have added several new experiments to the paper that add quantitative rigor to our claims. We hope our responses address the reviewer’s concerns and, if so, they would consider raising their score. Please let us know if you have any further questions!

---

### Official Review · Reviewer_QXh2 · 2025-11-03

**Soundness:** 3
**Presentation:** 4
**Contribution:** 3
**Rating:** 8
**Confidence:** 3

**Summary:**

The authors first provide a formal characterization of standard SAEs, arguing SAEs impose an implicit prior that assumes concepts are independent and identically distributed (i.i.d.) across time. They contrast this with an empirical analysis of LM activations, which reveals rich temporal dynamics, including non-stationarity and strong context-dependent correlations. To address this mismatch, the authors introduce the Temporal SAE, a novel architecture with a temporal inductive bias. This model decomposes activations at each timestep into two components: a predictable component ($x_p$), inferred from the context, and a novel component ($x_n$), which captures the residual information. The authors demonstrate that this decomposition allows the model to separate slow-moving, abstract information (in $x_p$) from fast-moving, novel information (in $x_n$). They provide evidence that this model is better at identifying temporal structures, such as event boundaries and correct garden-path sentence parses, which standard SAEs fail to capture.

**Strengths:**

1. *Formal Characterization of SAE Priors:* The paper provides a clear and valuable formalization of the implicit assumptions in standard SAEs. Characterizing the priors as i.i.d. across time (Proposition 3.1) 6cleanly articulates the central problem.
2. *Strong Empirical Analysis:* The claims about the data's temporal structure are well-supported by a robust empirical analysis. The use of multiple metrics, including intrinsic dimensionality (U-statistic) and autocorrelation, demonstrates the non-stationarity and context-dependence of LM activations (Figure 2) that hold back traditional SAE-based approaches.
3. *Clarity and Intuition:* The paper is very well-written and intuitive. The core problem and the proposed solution are explained clearly, making the work easy to follow.

**Weaknesses:**

1. *Evaluation Relies on Correlational Evidence:* The primary evidence for the Temporal SAE's improved interpretability is qualitative (dendrograms in Fig. 5, 7) or correlational (CKA with slow/fast Fourier signals in Table 3, Fig. 6). While standard metrics like reconstruction (Table 1) are included as sanity checks, the paper never demonstrates that the disentangled $z_p$and $z_n$ features are _causally_ more useful than standard SAE features at steering model behavior.
    - The qualitative claims about event boundaries (Fig. 5) are subjective and would be much stronger if validated quantitatively against human-annotated datasets, such as those the authors already cite (e.g., Zacks et al., 2007; Baldassano et al., 2018).
    - A stronger claim would require causal validation. For instance, are $z_p$ features more effective for patching high-level topics between texts? The paper claims $z_n$ captures "novel" information; does this imply $z_n$ features might be better targets targets for model editing? A demonstration that interventions on $z_n$ are more localized and effective than on standard (entangled) $z$ features would provide stronger causal evidence.
2. *Novelty =/= Un-predictability*: The paper's decomposition rests on an assumption that "novel" information is orthogonal to "predictable" information. This is questionable, as linguistically or cognitively "novel" information (e.g., the start of a new topic) is not always unpredictable. In fact, research on human event memory (which the authors cite) suggests that event boundaries are registered at predictable changes in context, not just at moments of high prediction error. This creates a potential mismatch: the model's "novel" component may only be capturing an unpredictable residual, rather than the more structured, and sometimes predictable, "novelty" that defines temporal structures.

**Questions:**

1. *Shared vs. Separate Dictionaries:* The model shares a single dictionary $D$ for both the predictable ($z_p$) and novel ($z_n$) components. Given that these codes are purported to capture fundamentally different types of information (slow vs. fast, abstract vs. transient), what is the justification for sharing the dictionary? Were separate dictionaries/encoders explored?
2. *Causal Utility of Features:* As mentioned in the weaknesses, the "gold standard" for interpretability is often causal intervention. Have the authors performed any experiments (e.g., activation patching, model editing) to demonstrate that the $z_p$ and $z_n$ features are more causally effective or localized than features from a standard SAE? Demonstrating this on the garden-path sentence examples would be particularly compelling.
3. *Quantitative Validation of Event Boundaries:* The hierarchical clustering results in Figure 5 are visually interesting, but can these results be validated more rigorously? For example, by computing a quantitative metric (e.g., F1-score, ARI) for event boundary detection against the human-annotated ground truth in datasets like those cited (Zacks et al., 2007; Baldassano et al., 2018)?

---

> ### Author Response · Authors · 2025-11-24
> **Rebuttals (1/2)**
>
> We thank the reviewer for their thorough and positive feedback! We are grateful the reviewer appreciated our work as “clear valuable formalization of implicit assumptions of SAEs”, with “robust empirical analysis”, and described our paper as “very well written and intuitive”. Below, we provide responses to specific questions raised by the reviewer.
>
> ----
>
> > **Qualitative claims about event boundaries**
>
> We agree with the reviewer that running experiments with human annotated boundaries would be the strongest result. Given time constraints, to address this comment, we performed an experiment motivated by Georgiou et al. [1]'s findings that event annotations for simple stories by GPT-4 (and likely GPT-5) have high alignment with human event boundary annotations. Specifically, we manually wrote a set of stories with precise event boundaries (see Fig. 6a), used GPT-5 to generate 50 synthetic stories with similar structure, and then inspired by the study by Georgiou et al. [1], had GPT-5 label the event boundaries in these stories. We then measured how clustered the representations of tokens belonging to the same vs. different events are. In particular, we computed the average cosine similarity of token representations when they come from the same event versus different ones. As shown in Fig. 6c, we find Temporal SAEs’ predictive codes achieve substantially higher cosine similarity than other SAEs---as expected by clustering exhibited in similarity heatmaps in Fig. 6b. To further elaborate on this point, we took inspiration from literature on graph clustering and added noise to the input representations before computing codes and computing similarity heatmaps: this process should yield coarser clusters since high frequency signals will get overwhelmed by noise. Results are shown in Fig. 6d,e, and show a clear signature of coarser clusters emerging under increasing noise scale, demonstrating the multiscale nature of language!
>
> [1] https://learnmem.cshlp.org/content/32/2/a054043.short
>
> > **A stronger claim would require causal validation.**
>
> We agree with the reviewer’s comments on how we can make strong claims for TemporalSAE by running experiments assessing how novel and predictive codes differ in their interpretability (e.g., whether novel codes offer more localized and predictive codes more global interpretability) and whether these interpretations are causally relevant. To this end, we have added the following experiments to the paper.
>
>    - **Predictive codes capture global structure:** We have added low-dimensional visualizations of codes from different SAEs extracted using UMAP from stories (see Fig. 4 and App. G). We consistently see predictive codes from Temporal SAE cluster up tokens according to events (as highlighted in the dendrograms in Fig. 4b and ones shown in App. G). We find similar results from Temporal SAEs trained on Llama as well (see App. H). Interestingly, we find these results generalize OOD to domains like chat: specifically, when we use Temporal SAEs on activations extracted Gemma-2-2b-IT (a chat model), we see two manifold-like structures emerge that capture semantics of user vs. model. We in fact found these manifolds can be further decomposed into sub-manifolds that correspond to specific domains like advice on code and food recipes—these sub-manifolds are as of yet not annotated in the figure, but we will do so for the final version of the paper.
>
>    - **Novel codes from Temporal SAEs and codes from standard SAEs behave similarly:** UMAP visualizations of novel codes from Temporal SAEs and standard SAEs’ codes show they primarily cluster representations according to lexical information, i.e., the inputted token (see Fig. 4b and App. G, H). In fact, the code similarity heatmaps produced by novel codes match very well different SAEs’ latent codes similarity maps. Motivated by this, we ran auto-interpretability experiments with novel codes of Temporal SAEs and standard SAEs, finding features identified from standard SAEs had counterparts in novel codes of Temporal SAEs. We promise these autointerpretability experiments (with the option to perform live steering) and a new heatmap visualization tool to assess heatmap visualizations from predictive codes will be made publicly available via Neuronpedia if the paper is accepted.
>
> **(Continued below...)**

---

> ### Author Response · Authors · 2025-11-24
> **Rebuttals (2/2)**
>
> - **Assessing causality via SAEBench:** Finally, to address the reviewer’s comment on performing an investigation of causality of codes extracted using Temporal SAEs, we are running experiments on a subset of SAEBench [1] evaluations that gauge causality of assigned interpretations. We note the precise protocol for steering predictive codes is unclear: the apt notion of intervention likely involves developing tooling that allows one to move on a manifold-like structure isolated using predictive codes; we are actively pursuing this question, but believe it warrants its own detailed study. Thus, for now, we specifically focus on the novel codes extracted using Temporal SAEs for this experiment. In the Spurious Correlation Removal metric, temporal novel codes achieve a score of 0.16, while the TopK SAE baseline scores 0.25. Both codes significantly outperform the PCA baseline score 0.03. We are still running the Targeted Probe Perturbation (TPP) and RAVEL evaluations, which are currently bottlenecked by some code refactorization issues to run evaluations, but promise to include the numbers in the final version of the paper.
>
> [1] https://arxiv.org/abs/2503.09532
>
> ----
>
> > **Novelty =/= Un-predictability**
>
> This is a fair point and we agree with the reviewer that a “novel” signal is not inherently unpredictable! However, we note the specific sense in which we use the term “novel” is to refer to a part of the representation that based on the past context was not expected or predicted, i.e., one that is surprising. Since our assumed generative model for the representations is that they can be decomposed into a sum of predictable and unpredictable (i.e., novel) parts, we argue the novel component capturing a surprising, unpredictable residual is consistent with our goals—we have added a short discussion (see para above Eq. 3) to emphasize this point better in the paper!
>
> ----
>
> > **Shared vs. Separate Dictionaries**
>
> Great question! We agree that given the disparate roles of predictive and novel codes, it would make sense to split their dictionaries into two. During the development of this project, we wanted to keep hyperparameters to a minimum; we thus chose to use a single, unified dictionary for both parts. However, motivated by your comment, we performed a posthoc analysis of a TemporalSAE trained on Gemma-2-2b and found the dictionary emergently organized into a part that is primarily used for computing the predictive code and a part that is primarily used for computing the novel code (see App. F). Specifically, approximately 20% of the dictionary is involved in predictive code computation and the rest 80% contributes to novel code’s computation.
>
> Given this emergent structure, we thus decided to further test your suggestion and hence trained Temporal SAEs where the dictionary was explicitly split into two parts—one for the predictive code and one for novel code. We found this split dictionary Temporal SAE yields essentially the same overall learning curves, with an equivalent final loss. Both predictive codes of shared dictionary and split dictionary temporal saes show significantly lower effective rank ($\sim$20-30) than novel codes ($\sim$390-420).  We believe a more detailed investigation of the benefits of shared vs. split dictionary Temporal SAEs is left to future work, but tentatively argue that given the emergent splitting of shared dictionary Temporal SAEs, downstream results recovered from the two architectures will be similar.
>
> ----
>
> > **Quantitative Validation of Event Boundaries**
>
> As noted above (see paragraph title “Event Boundaries”), we have now included additional quantitative results on detecting event boundaries with our predictive codes. We summarize the findings below for your convenience.
> We generate 42 stories containing two events, separated by a twist that happens at an annotated token position, in the spirit of the story in Fig 6a. The set of stories with a twist allows us to quantitatively evaluate. We report the mean cosine similarity of codes within an event, vs the mean similarity across events in Fig 6c. The difference of within-similarity and across-similarity is significantly higher for predictive codes of Temporal SAE than for novel codes of Temporal SAE and codes of existing SAE architectures. These results add quantitative evidence towards our observation that predictive codes detect event boundaries.
>
> ----
> ----
>
> **Summary:** We thank the reviewer for their feedback and the great suggestions that helped improve the quantitative rigor of our work! We hope our responses helped address their concerns and, if so, hope that they will consider championing our paper during the discussion period.

---

### Meta-Review · Area_Chair_w5UU · 2026-01-07

**Summary:**

This paper argues that standard Sparse Autoencoders (SAEs) used for mechanistic interpretability impose an implicit i.i.d./stationary prior, while LM activations exhibit strong temporal structure and non-stationarity. The authors propose Temporal SAE, which decomposes each timestep’s activation into a predictable component and a residual component. Reviews are overall positive-to-mixed: one strong accept (poster) and one marginal accept, with one marginal reject.
The key concerns driving my decision are (i) whether the paper provides sufficiently quantitative and/or causal validation that the extracted codes improve interpretability beyond correlational visualizations, (ii) whether the “novel = unpredictable residual” framing is conceptually aligned with linguistic/cognitive notions of novelty, and (iii) clarity/notation/baseline details. The rebuttal substantially strengthens the quantitative side and addresses several clarity issues, but causal intervention evidence remains somewhat limited. Overall, I recommend "Accept" because the conceptual diagnosis (prior mismatch) + methodological contribution (Temporal SAE) + strengthened empirical evidence constitute a solid interpretability advance.

**Reviewer Concerns:**

Likely addressed:
Need for more quantitative evaluation of temporal structure
Orthogonality assumption
Shared vs. separate dictionaries for predictable vs novel codes.
Adding more related work.
Notation issues
Further baselines

Only partially addressed:
Causal utility

**Reviewer Scores:**

Expected changes for each reviewer:

QXh2 (8 → 8): already positive; rebuttal directly addressed their questions.

r8pv (4 → 6): key complaints were “purely analytical” and orthogonality assumption + missing related work. Rebuttal added quantitative evaluations, clarifies/removes orthogonality constraint, adds related work, and starts causal evaluation via SAEBench.

mbJz (6 → 7): their concerns were mostly clarity/notation/baselines and positioning around monosemantic features.

---

### Decision · Program_Chairs · 2026-01-26

Accept (Poster)